# Towards Generalizable Multi-Policy Optimization with Self-Evolution for Job Scheduling

**Inguk Choi, Woo-Jin Shin, Sang-Hyun Cho, Hyun-Jung Kim**[*]
Manufacturing and Service Systems Lab
Dept. of Industrial and Systems Engineering
Korea Advanced Institute of Science and Technology (KAIST)
{inguk0826, wjshin, ie02002, hyunjungkim}@kaist.ac.kr

## Abstract

Reinforcement Learning (RL) has shown promising results in solving Job Scheduling Problems (JSPs), automatically deriving powerful dispatching rules from data without relying on expert knowledge. However, most RL-based methods train only a single decision-maker, which limits exploration capability and leaves significant room for performance improvement. Moreover, designing reward functions for different JSP variants remains a challenging and labor-intensive task. To address these limitations, we introduce a novel and generic learning framework that optimizes multiple policies sharing a common objective and a single neural network, while enabling each policy to learn specialized and diverse strategies. The model optimization process is fully guided by a self-labeling manner, eliminating the need for reward functions. In addition, we develop a training scheme that adaptively controls the imitation intensity to reflect the quality of self-labels. Experimental results show that our method effectively addresses the aforementioned challenges and significantly outperforms state-of-the-art RL methods across six JSP variants. Furthermore, our approach also demonstrates strong performance on other combinatorial optimization problems, highlighting its versatility beyond JSPs.

## 1 Introduction

Job Scheduling Problems (JSPs) are fundamental Combinatorial Optimization Problems (COPs) with significant practical importance across various industries such as manufacturing [1], logistics [2], and data centers [3]. Solving a JSP involves assigning jobs to machines (e.g., limited resources) and sequencing them on each machine. The goal is to find a schedule from a combinatorial solution space that minimizes (or maximizes) the objective function under problem-specific constraints. Traditionally, JSPs have been solved using exact methods or handcrafted heuristic algorithms. However, exact methods are computationally intractable for large-size problems [4], and designing effective heuristics for each JSP variant requires deep domain knowledge and significant manual effort [5].

Beyond expert-designed heuristics, Neural Combinatorial Optimization (NCO) methods, as variants of Hyper-Heuristics (HH) [6], have recently emerged to automate the heuristic design process [7, 8]. In particular, neural constructive heuristics, which sequentially build solutions from scratch using Deep Neural Networks (DNNs), have attracted significant attention due to their simplicity and flexibility [9, 10, 11, 12]. These methods leverage DNNs to model decision-making policies (traditionally represented by Priority Dispatching Rules (PDRs)) and learn state-to-action mappings from data via Supervised Learning (SL) or Reinforcement Learning (RL). However, due to the NP-hardness of most JSPs, SL approaches struggle to obtain sufficiently high-quality solutions for

---

[*]Corresponding author

39th Conference on Neural Information Processing Systems (NeurIPS 2025).

labeled data. Accordingly, RL-based policy gradient methods, which optimize policies using reward signals, have gained popularity and shown promising results [13].

However, despite its strengths, applying RL to JSPs still faces two key challenges. **(1) Exploration:** Due to the exponentially large search space of JSPs [14] and the trial-and-error nature of RL, effective exploration in both training and inference phases is essential for finding high-quality solutions. Nevertheless, most RL-based methods train only a single policy, which often suffers from insufficient exploration due to mode collapse, where the policy distribution converges toward a unimodal distribution during RL training [15, 16]. For Vehicle Routing Problems (VRPs), another well-known class of COPs, NCO methods effectively enhance search capabilities by leveraging optimality symmetries (solution symmetry [10] and problem symmetry [17, 18]). However, JSPs lack universally definable beneficial symmetries, making it difficult to enforce effective exploration. **(2) Reward shaping:** JSPs have numerous variants based on machine environments, constraints, and objective functions [19]. Thus, designing bespoke reward functions for each variant remains a complex and challenging task [20]. Although the objective value can be directly used as a true reward for policy optimization using the REINFORCE algorithm [21], this approach suffers from reward sparsity and non-trivial credit assignment problems [22], which add to the training complexity.

**Contributions.** In this paper, we propose a novel and generic learning framework to address the aforementioned challenges. Rather than training a single policy, our framework aims to learn multiple policies that share the same objective and model parameters, but solve the problem using distinct strategies. To this end, we introduce the `MP-ASIL` (**M**ulti-**P**olicy Optimization with **A**daptive **S**elf-**I**mitation **L**earning), designed to guide each policy to learn diverse and complementary problem-solving strategies in a fully self-evolutionary manner. `MP-ASIL` addresses the limitations of RL-based methods in the following ways. Firstly, multiple specialized policies can express a multimodal action distribution, alleviating the mode collapse problem in single policy approaches and improving solution quality as a natural byproduct of enhanced search capability. Secondly, `MP-ASIL` autonomously generates training labels to guide model optimization, eliminating the need for problem-specific Markov Decision Process (MDP) formulations. Beyond addressing the RL limitations, we also develop a training scheme that adaptively controls imitation intensity based on the quality of the self-teacher, mitigating fundamental drawbacks of existing Self-Imitation Learning (SIL)-based methods. Finally, `MP-ASIL` is a task- and model-agnostic learning framework, enabling easy plug-in to diverse neural solvers across various problem domains.

To validate the effectiveness of `MP-ASIL`, we evaluate it on six widely studied JSPs: Single Machine Scheduling Problem (SMSP), Unrelated Parallel Machine Scheduling Problem (UPMSP), Permutation Flow Shop scheduling Problem (PFSP), Flexible Flow Shop scheduling Problem (FFSP), Job Shop Scheduling Problem (JSSP), and Flexible Job Shop Scheduling Problem (FJSSP). The experimental results demonstrate that `MP-ASIL` successfully unlocks the potential of models for significantly improved exploration capabilities and overall performance, providing new state-of-the-art results on various synthetic and benchmark datasets. Furthermore, `MP-ASIL` also demonstrates strong performance on other COPs, underscoring its broad applicability beyond JSPs.

## 2 Related Work

**Neural Constructive Heuristics for JSPs.** Recent advances in artificial intelligence have opened new avenues for solving JSPs with Machine Learning (ML) [23]. Various learning-based approaches have been studied, including neural improvement heuristics that iteratively refine complete solutions via neural-guided local search [24, 25, 26] and hybrid methods that integrate ML into classical heuristics [27, 16, 28]. Nonetheless, most learning-based methods have primarily focused on neural constructive heuristics. L2D [11], a seminal work in this area, introduces a Graph Neural Network (GNN)-based policy for solving JSSP. It sequentially assigns operations to machines using the topological information of partial solutions represented as disjunctive graphs, outperforming traditional dispatching rules. Building on this success, several methods have been proposed for diverse JSP variants, differing in how the networks are designed (e.g., GNN-based policies [29, 30, 31] or Transformer [32]-based policies [12, 33]) and how the networks are trained (e.g., actor-critic methods [34, 35, 36, 37], REINFORCE algorithms [38, 39, 40, 41], or SL [42, 43, 44]). Despite these advancements, most existing approaches only train a single policy, limiting exploration capability and

leaving substantial room for performance improvement. Furthermore, they often require specialized MDP formulations or expert knowledge for training, thus limiting their generalizability to other JSPs.

Recently, to eliminate the need for reward function design and labeled data, SIL-based methods have emerged as self-labeling approaches for solving JSPs. In this paradigm, a policy generates multiple candidate solutions and selects the best one as the expert trajectory for SL. SLIM [20] generates candidate solutions via vanilla stochastic sampling from a single policy, which produces many duplicate candidates, thereby leaving less space for potentially better solutions. To improve the sampling process, SI GD [45] proposes a method based on drawing trajectories in multiple steps using stochastic beam search. However, this approach demands extensive search effort for each instance and careful hyperparameter tuning. Existing SIL-based methods also disregard the quality of their self-labels and exhibit low sample efficiency as they rely solely on the best solution and discard the rest. In contrast, our work addresses these limitations via the simple yet powerful `MP-ASIL` and demonstrates its effectiveness across various JSPs.

**Improving Solution Diversity in NCO.**    Many recent NCO methods for VRPs follow the POMO approach [10], improving exploration by generating multiple solutions from different starting points. However, in JSPs, initial actions often significantly impact solution quality, limiting the applicability of the POMO method to JSPs. Although LCP [46] proposes a general methodology that encourages sampling diverse solutions via entropy regularization, computing the entropy over the entire trajectory remains computationally intractable. Recently, Generative Flow Networks (GFlowNets) [47] have gained attention due to their powerful exploration capabilities [48, 49, 16, 50]; however, substantial post-search efforts are still required to achieve competitive results. For JSPs, some studies improve solution quality through beam search [51], active search [52], or look-ahead search [53], yet approaches specifically aimed at promoting solution diversity remain limited.

A promising research direction in NCO involves training multiple policies to learn different solution patterns. MDAM [15] proposes an Attention Model (AM) [9] with multiple decoders to train diverse policies for VRPs. It maximizes the Kullback-Leibler divergence between initial action distributions to encourage distinct solution patterns. Poppy [54] introduces "Winner-takes-all" strategy as a REINFORCE variant for multi-decoder training, in which only the best-performing policy is updated at each iteration. Despite their effectiveness, multi-decoder models require a separate decoder for each policy, resulting in substantial computational overhead and limiting scalability as the number of policies increases. Similar to our work, COMPASS [55] and PolyNet [56] use continuous or discrete latent spaces to represent multiple policies within a single model. However, both methods train the model using a variant of REINFORCE [54], where gradients are computed from complete trajectories. As a result, the training process suffers from high variance and instability.

## 3 Preliminaries

**Job Scheduling Problems.**    In this work, we focus on standard and static JSPs. A standard JSP instance of size $|\mathcal{J}| \times |\mathcal{M}|$ consists of a set of jobs $\mathcal{J}$, a set of machines $\mathcal{M}$, and a set of operations $\mathcal{O}$. Each job $j \in \mathcal{J}$ comprises $m_j$ operations $\{O_{ji}\}_{i=1}^{m_j} \subseteq \mathcal{O}$ that must be processed in a predefined order $O_{j1} \to \cdots \to O_{ji} \to \cdots \to O_{jm_j}$, where $O_{ji}$ denotes the $i$th operation of job $j$. Each operation $O_{ji}$ can be processed on exactly one machine from its set of eligible machines $\mathcal{M}_{ji} \subseteq \mathcal{M}$ with processing time $p_{jik} \in \mathbb{R}_{>0}$ on machine $k \in \mathcal{M}_{ji}$. Based on this formulation, each JSP is uniquely characterized by its specific machine environment, constraints, and objective functions [57] (see Appendix A for details). Given a JSP instance $s \sim \mathcal{D}$, where $\mathcal{D}$ is an instance distribution, our goal is to find a solution $\boldsymbol{\tau} \in \Omega$, where $\Omega$ is a finite solution space that satisfies all constraints while minimizing (or maximizing) a predefined objective function $f : (\boldsymbol{\tau}, s) \to \mathbb{R}$. $f$ need not be injective; due to the multimodality of the objective function in JSPs [14], distinct solutions can have the same objective value. In this paper, without loss of generality, we consider minimization problems.

**Constructing Solutions Using a Parameterized Policy.**    A JSP solution $\boldsymbol{\tau}$ can be autoregressively constructed by sequentially assigning each operation to a compatible machine according to the policy and appending it to the end of that machine's operation sequence. At this point, we define $\boldsymbol{\tau} = (\tau_1, \ldots, \tau_t, \ldots, \tau_{|\mathcal{O}|})$ as a sequence of decisions, with the policy $\pi_\theta$ modeled as a DNN parameterized by $\theta$. As illustrated in Figure 1, at each decision step $t$, the parameterized policy computes a conditional action distribution $\pi_\theta(\tau_t \mid s, \boldsymbol{\tau}_{<t})$ over the next operation $\tau_t$, where $\boldsymbol{\tau}_{<t}$

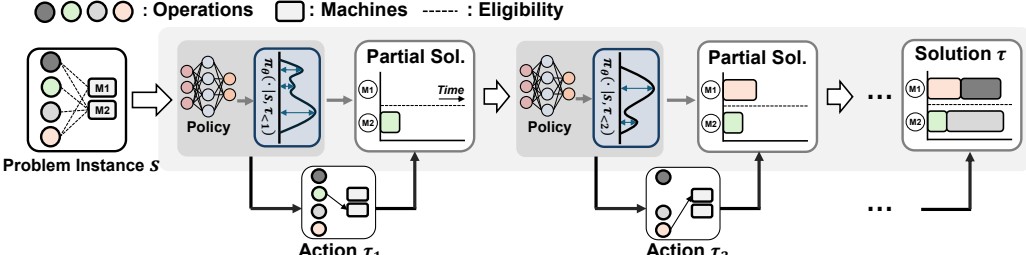

Figure 1: Illustrative example of sequential decision-making process using neural constructive heuristics to build a solution $\boldsymbol{\tau}$.

represents the partial solution until step $t$, guiding the sequential decision-making process until all operations are scheduled. Consequently, the overall policy $\pi_\theta(\boldsymbol{\tau} \mid s)$ for generating a solution $\boldsymbol{\tau}$ given an instance $s$ is factorized as:

$$\pi_\theta(\boldsymbol{\tau} \mid s) = \prod_{t=1}^{|\mathcal{O}|} \pi_\theta(\tau_t \mid s, \boldsymbol{\tau}_{<t}). \tag{1}$$

## 4 Methods

In this section, we introduce MP-ASIL, a novel and generic learning framework to address several challenges in solving JSPs with RL. An overview of MP-ASIL is illustrated in Figure 2.

### 4.1 Multi-Policy Representation: Latent Conditioned Policies

As discussed in Section 1, enforcing effective exploration in JSPs is challenging, and single policy approaches struggle to balance exploration and exploitation. To address these issues, we aim to learn multiple policies (a set of neural heuristics) that share the same objective and model parameters but can represent diverse and complementary solution patterns. In this work, we model this population by conditioning a single neural network on different latent variables, referred to as latent conditioned policies [58, 55, 56]. Formally, the latent conditioned policy is described as $\pi(\cdot \mid s, z)$, conditioned on an instance $s$ and the latent variable $z \in \mathbb{R}^{d_z}$. By sampling multiple latent variables $z^1, \ldots, z^k$ from a fixed latent distribution $\mathcal{Z}$, we can obtain a policy set $\Pi$ as follows:

$$\Pi = \left\{ \pi_\theta(\cdot \mid s, z^i) \mid z^i \sim \mathcal{Z},\ i = 1, 2, \ldots, k \right\}. \tag{2}$$

Each latent variable defines a distinct policy, enabling a single DNN to represent multiple decision-makers. These latent conditioned policies can be implemented regardless of the underlying architecture. Appendix B provides deeper motivation for using latent variables to represent the population, as well as the implementation details and the distribution for sampling the latent variables.

### 4.2 MP-ASIL: Multi-Policy Optimization with Adaptive Self-Imitation Learning

**Motivation.** Given our motivation for using multiple policies, the following question naturally arises: *How can we guide these policies to learn diverse and complementary problem-solving strategies?* This question emerges because merely representing multiple policies does not ensure that they can generate diverse solutions. To answer this, we introduce MP-ASIL, designed to guide each policy to specialize into a distinct yet powerful schedule generator. Our method is based on three principles: **(1)** we aim not merely for diversity (e.g., random policies as an extreme case), but for useful diversity that effectively helps to find better solutions [59]; **(2)** it is unnecessary for every policy in $\Pi$ to show strong performance on a given instance $s$, as inference requires selecting only the best solution among candidates; **(3)** the following population-level inference objective, defined as:

$$\mathbb{E}_{s \sim \mathcal{D}} \mathbb{E}_{z^1, \ldots, z^k \sim \mathcal{Z}} \mathbb{E}_{\boldsymbol{\tau}^1 \sim \pi_\theta(\cdot \mid s, z^1), \ldots, \boldsymbol{\tau}^k \sim \pi_\theta(\cdot \mid s, z^k)} \min\{f(\boldsymbol{\tau}^1, s), \ldots, f(\boldsymbol{\tau}^k, s)\}, \tag{3}$$

should be reflected during training. However, RL-based methods suffer from sparse learning signals, as feedback emphasizing the population objective is provided only after generating a complete

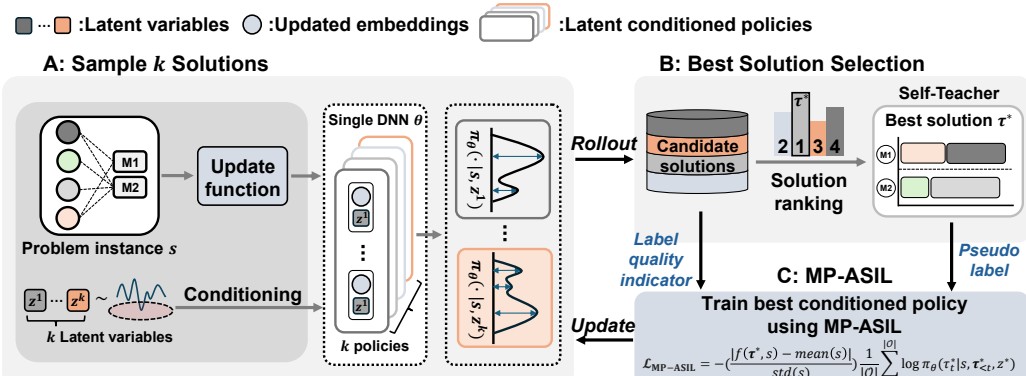

Figure 2: An overview of MP-ASIL. **Step A** (Section 4.1): We generate $k$ distinct policies by conditioning a single DNN on $k$ latent variables, and sample $k$ solutions from these policies. **Step B** (Section 4.2): Among these $k$ solutions, the one with the lowest objective value is selected as the pseudo-label $\boldsymbol{\tau}^*$. **Step C** (Section 4.2): The model is trained to imitate $\boldsymbol{\tau}^*$ using MP-ASIL.

decision trajectory. Moreover, designing dense surrogate reward functions is a complex and non-trivial task. To overcome these challenges, we reformulate the problem of learning heuristics from a return-maximization task to a classification task, in which the model autonomously generates and imitates pseudo-labels corresponding to the best current action at each constructive step.

**Training Procedure.** Specifically, for each training instance $s$, (**Step A:**) we draw $k$ latent variables from $\mathcal{Z}$ and generate $k$ policies by conditioning the policy network on the sampled variables. (**Step B:**) From these policies, we roll out $k$ candidate solutions simultaneously and select the one with the lowest objective value as the pseudo-label $\boldsymbol{\tau}^* = \arg\min_{\boldsymbol{\tau}^i \in \{\boldsymbol{\tau}^1, \ldots, \boldsymbol{\tau}^k\}} f(\boldsymbol{\tau}^i, s)$, obtained from the best-performing policy $\pi_\theta(\cdot|s, z^*)$. (**Step C:**) The model is then optimized to imitate the best decision $\boldsymbol{\tau}^* = \{\tau_1^*, \ldots, \tau_t^*, \ldots, \tau_{|\mathcal{O}|}^*\}$ at each step in an SL manner (maximizing the conditional log-likelihood). However, pseudo-labels are not guaranteed to be globally optimal, and their quality can vary according to the sampled policies. Therefore, to adaptively control imitation intensity based on pseudo-label quality, we modify the loss function (cross-entropy loss) as follows:

$$\mathcal{L}_{\text{MP-ASIL}} = -\left(\frac{|f(\boldsymbol{\tau}^*, s) - mean(s)|}{std(s)}\right) \frac{1}{|\mathcal{O}|} \sum_{t=1}^{|\mathcal{O}|} \log \pi_\theta(\tau_t^* \mid s, \boldsymbol{\tau}_{<t}^*, z^*), \tag{4}$$

where $mean(s) = \frac{1}{k} \sum_{i=1}^{k} f(\boldsymbol{\tau}^i, s)$ is the average objective value across $k$ candidate solutions and $std(s) = \sqrt{\frac{1}{k} \sum_{i=1}^{k} (f(\boldsymbol{\tau}^i, s) - mean(s))^2}$ is the standard deviation. In Equation (4), we use the normalized advantage value $\frac{|f(\boldsymbol{\tau}^*, s) - mean(s)|}{std(s)}$ as a pseudo-label quality indicator that enables adaptive SIL, where superior pseudo-labels are strongly imitated and less informative pseudo-labels are weakly imitated.

**Rationale of MP-ASIL and Summary.** Our learning framework establishes a *self-evolutionary loop* via an iterative optimization process by generating and imitating progressively stronger self-teachers. Notably, to reflect the population-level objective in Equation (3) during training, our method updates the model based solely on the performant policy, incentivizing higher probabilities for specific actions conditioned on $z^*$ and $s$. This training procedure naturally encourages the latent space to be *diverse* (guided by latent variables) and *specialized* (optimized for distinct instance sub-distributions). As a result, trained latent conditioned policies can represent a multimodal trajectory distribution without requiring diversity-enforcing mechanisms (e.g., entropy bonus), mitigating the relatively deterministic action distribution problem in single policy approaches. Our approach also resolves the sparse feedback problem in RL by removing the need for reward functions.

Beyond overcoming RL limitations, MP-ASIL improves upon existing SIL-based methods by leveraging information from all sampled solutions, improving sample efficiency and facilitating adaptive imitation intensity control to avoid over-exploitation of suboptimal solutions.

**Algorithm 1** MP-ASIL training

---

1: **Input:** Model parameters $\theta$, instance distribution $\mathcal{D}$, latent variable distribution $\mathcal{Z}$, number of epochs $E$, number of training steps $T$, batch size $B$, and number of policies $k$.
2: Initialize model parameters $\theta$.
3: **for** $epoch = 1$ to $E$ **do**
4:     **for** $step = 1$ to $T$ **do**
5:         $s_i \leftarrow \texttt{SampleInstance}(\mathcal{D}), \quad \forall i \in \{1, \ldots, B\}$
6:         $\Pi_i = \{\pi_\theta(\boldsymbol{\tau}_i^j \mid s_i, z_i^j)\}_{j=1}^k \leftarrow \texttt{SamplePolicy}(\mathcal{Z}), \quad \forall i \in \{1, \ldots, B\}$
7:         $\{\boldsymbol{\tau}_i^1, \ldots, \boldsymbol{\tau}_i^k\} \leftarrow \texttt{SampleRollout}(\Pi_i), \quad \forall i \in \{1, \ldots, B\}$
8:         $\boldsymbol{\tau}_i^* = \arg\min_{\boldsymbol{\tau}_i^j \in \{\boldsymbol{\tau}_i^1, \ldots, \boldsymbol{\tau}_i^k\}} f(\boldsymbol{\tau}_i^j, s_i), \quad \forall i \in \{1, \ldots, B\}$   ▷ Select the best solution.
9:         $\mathcal{L}_{\text{MP-ASIL}} = -\frac{1}{B} \frac{1}{|\mathcal{O}|} \sum_{i=1}^B \frac{|f(\boldsymbol{\tau}_i^*, s_i) - mean(s_i)|}{std(s_i)} \sum_{t=1}^{|\mathcal{O}|} \log \pi_\theta(\tau_{i,t}^* \mid s_i, \boldsymbol{\tau}_{i,<t}^*, z_i^*)$
10:         $\theta \leftarrow \text{Adam}(\theta, \nabla_\theta \mathcal{L}_{\text{MP-ASIL}})$   ▷ Update solely based on the performant policy.
11:     **end for**
12: **end for**
13: **Output:** Trained model parameters $\theta$.

---

Last but not least, MP-ASIL can be directly applied to existing neural constructive solvers (detailed in Appendix B.1) and JSPs without any algorithmic modifications, since our method leverages the fundamental property of JSPs that solutions generated for the same instance can be discriminated by their objective values. Therefore, we can easily obtain the MP-ASIL recipe ingredients (latent conditioned policies, pseudo-label, and label-quality indicator) for any setting. The mini-batch training of MP-ASIL is summarized in Algorithm 1.

## 5 Experiments

We evaluate MP-ASIL on six representative JSP variants: SMSP, UPMSP, PFSP, FFSP, JSSP, and FJSSP. These problems cover a wide range of JSP scenarios. Detailed definitions of each problem are provided in Appendix C. We first describe the experimental settings of MP-ASIL (Section 5.1) and then present the experimental results and detailed analysis (Section 5.2). All experiments are conducted on a 24-core Intel(R) i9-14900KS CPU and a single NVIDIA GeForce RTX 4090.

### 5.1 Experimental Settings

**Model & Training.** We implement MP-ASIL on top of the problem-specific backbone models for each task, for two reasons. First, neural solvers for JSPs have developed with specialized network architectures tailored to each problem. Second, this experimental design highlights MP-ASIL as a generic learning framework that is agnostic to the problem type and underlying model architecture. We use training hyperparameters of the backbone models from original papers whenever applicable. Detailed model architectures and training settings are presented in Appendices C and F. Additionally, Appendix D.1 provides validation scores during the training process.

**Baseline Methods.** We compare MP-ASIL with various state-of-the-art classic heuristics and NCO methods for each problem. Details of the baselines for each problem are described in Appendix C.

**Test Datasets & Inference.** We evaluate MP-ASIL on benchmark and synthetic datasets widely used in the NCO and the operations research communities (Appendix C). At inference time, we select the best solution from $k$ candidates generated by the $\Pi$. To ensure fair comparison, we match $k$ to the sample size reported in prior studies using the same architecture; otherwise, we set $k = 128$.

**Performance Metrics.** We use three metrics for evaluation: average objective value (Obj.), average performance gap (Gap), and total inference time (Time). The performance gap for each method on an instance $s$ is calculated as $100 \times (f_o^s - f_b^s)/f_b^s$, where $f_o^s$ is the objective value obtained by each method, and $f_b^s$ is the best-known objective value for $s$. Note that reported Time may not be directly comparable across methods due to differences in hardware and other experimental settings. Therefore, for clarity, results obtained from the original papers are marked with an asterisk (*).

Table 1: Experiment results on SMSP, UPMSP, and FFSP. †: Methods using the same model as `MP-ASIL`. Exact: Exact solver. Heuristics: Handcrafted heuristics. NCH: Neural constructive heuristics. Hybrid: Hybrid methods. Gray : Our (`MP-ASIL`) results. S: Sampling size. ↓: Lower is better. **Bold**: Best Obj. and Gap among the NCO methods except for Large. ●: Instance sizes unseen during `MP-ASIL` training. Time units: s (seconds), m (minutes), and h (hours).

| Method | Type | SMSP 50 | | | SMSP 100 ● | | | SMSP 500 ● | | |
|---|---|---|---|---|---|---|---|---|---|---|
| | | Obj. ↓ | Gap ↓ | Time ↓ | Obj. ↓ | Gap ↓ | Time ↓ | Obj. ↓ | Gap ↓ | Time ↓ |
| EDD | Heuristics | 0.3268 | 49.84% | (0s) | 0.3950 | 66.24% | (0s) | 0.7287 | 97.70% | (1s) |
| ACO [27] | Heuristics | 0.7787 | >100% | (1.1m) | 6.9138 | >100% | (2.4m) | 646.81 | >100% | (28.9m) |
| DeepACO [27] | Hybrid | 0.2296 | 5.27% | (1.1m) | 0.2551 | 7.36% | (2.6m) | 0.5944 | 61.30% | (29m) |
| GFACS [16] | Hybrid | 0.4202 | 92.64% | (1.5m) | 1.2153 | >100% | (3m) | 14.612 | >100% | (33.7m) |
| MP-ASIL ($k$=128) | NCH | **0.2181** | **0.00%** | (5s) | **0.2376** | **0.00%** | (17s) | **0.3691** | **0.16%** | (14.5m) |
| MP-ASIL (Large) | NCH | 0.2181 | 0.00% | (1m) | 0.2376 | 0.00% | (2.3m) | 0.3685 | 0.00% | (53.5m) |

| Method | Type | UPMSP 50×3 ● | | | UPMSP 50×6 ● | | | UPMSP 100×6 ● | | |
|---|---|---|---|---|---|---|---|---|---|---|
| | | Obj. ↓ | Gap ↓ | Time ↓ | Obj. ↓ | Gap ↓ | Time ↓ | Obj. ↓ | Gap ↓ | Time ↓ |
| EDD | Heuristics | 2836.8 | >100% | (1s) | 778.5 | >100% | (1s) | 2472.1 | >100% | (2s) |
| ATCSR_Rm [60] | Heuristics | 877.0 | 22.91% | (6.6m) | 294.5 | 14.40% | (6.9m) | 735.6 | 75.94% | (10.0m) |
| Cho et al. (S=6) † [41] | NCH | 784.4 | 9.94% | (1.8m) | 294.2 | 14.29% | (2.4m) | 502.4 | 20.16% | (10.6m) |
| MP-ASIL ($k$=6) | NCH | **751.9** | **5.37%** | (1.8m) | **275.7** | **7.09%** | (2.4m) | **458.1** | **9.57%** | (10.7m) |
| MP-ASIL (Large) | NCH | 713.5 | 0.00% | (6.1m) | 257.4 | 0.00% | (24.6m) | 418.1 | 0.00% | (1h) |

| Method | Type | FFSP 20×12 | | | FFSP 50×12 | | | FFSP 100×12 | | |
|---|---|---|---|---|---|---|---|---|---|---|
| | | Obj. ↓ | Gap↓ | Time ↓ | Obj. ↓ | Gap↓ | Time ↓ | Obj. ↓ | Gap↓ | Time ↓ |
| CPLEX (1m)* [61] | Exact | 46.4 | 81.04% | (17h) | – | – | – | – | – | – |
| CPLEX (10m)* [61] | Exact | 36.6 | 42.80% | (167h) | – | – | – | – | – | – |
| SPT* [12] | Heuristics | 31.3 | 22.12% | (40s) | 57.0 | 14.22% | (1m) | 99.3 | 10.71% | (2m) |
| GA* [62] | Heuristics | 30.6 | 19.39% | (7h) | 56.4 | 13.03% | (16h) | 98.7 | 10.04% | (29h) |
| PSO* [63] | Heuristics | 29.1 | 13.54% | (13h) | 55.1 | 10.42% | (26h) | 97.3 | 8.48% | (48h) |
| MatNet (S=24) † [12] | NCH | 27.3 | 6.51% | (8s) | 51.5 | 3.21% | (13s) | 91.5 | 2.02% | (26s) |
| PolyNet ($k$=24) † [56] | NCH | 26.9 | 5.11% | (8s) | 51.2 | 2.56% | (13s) | 91.1 | 1.59% | (27s) |
| MP-ASIL ($k$=24) | NCH | **26.9** | **4.88%** | (8s) | **51.1** | **2.40%** | (13s) | **90.9** | **1.32%** | (27s) |
| MP-ASIL (Large) | NCH | 25.6 | 0.00% | (26s) | 49.9 | 0.00% | (1.1m) | 89.7 | 0.00% | (3.2m) |

Table 2: Experiment results on PFSP using the TA benchmark. G: Greedy action selection. The Time metric is reported in Appendix D.2.

| Method | Type | 20×5 | 20×10 | 50×5 ● | 50×10 ● | 100×5 ● | 100×10 ● | 200×10 ● | **Avg.** |
|---|---|---|---|---|---|---|---|---|---|
| | | Gap ↓ | Gap ↓ | Gap ↓ | Gap ↓ | Gap ↓ | Gap ↓ | Gap ↓ | Gap ↓ |
| ILS [64] | Heuristics | 6.81% | 9.45% | 4.14% | 11.69% | 3.44% | 9.57% | 6.63% | 7.39% |
| IGA [65] | Heuristics | 3.36% | 10.56% | 1.97% | 7.53% | 1.03% | 5.73% | 3.43% | 4.80% |
| NEH [66] | Heuristics | 2.40% | 4.45% | 0.66% | 4.69% | 0.41% | 2.04% | 1.28% | 2.28% |
| IL (G) [42] | NCH | 18.20% | 26.96% | 12.30% | 26.76% | 10.13% | 19.03% | 15.25% | 18.38% |
| PFSPNet (G)* [38] | NCH | – | 14.78% | – | 11.95% | – | 8.21% | – | 11.65% |
| Q-Learning (S=5)* [37] | NCH | 9.90% | 13.41% | 6.24% | 15.43% | 4.87% | 11.64% | 8.74% | 10.03% |
| MP-ASIL ($k$=128) | NCH | **0.37%** | **3.32%** | **0.22%** | **4.05%** | **0.16%** | **2.15%** | **1.51%** | **1.68%** |

## 5.2 Experimental Results

**Benchmark Results.** We first evaluate the performance of `MP-ASIL` on synthetic datasets for SMSP, UPMSP, and FFSP. The test datasets contain 100, 500, and 1,000 instances per problem size for SMSP, UPMSP, and FFSP, respectively. Additionally, we report results for `MP-ASIL` (Large), which generates $k \times 16$ policies, serving as an anchor for computing the Gap. As shown in Table 1, `MP-ASIL` significantly outperforms all baselines, achieving state-of-the-art results across all problem types and problem sizes. Specifically, for **SMSP**, DeepACO [27] and GFACS [16] retrain models for each problem size. In contrast, `MP-ASIL` trains solely on small-size instances ($|\mathcal{J}|$=50) yet demonstrates remarkable cross-size generalization, achieving 0.00% Gap for SMSP 50 and SMSP 100, and 0.16% Gap for SMSP 500. For **UPMSP**, we use the same network architecture (except for the multi-policy implementation) and training settings as Cho et al. [41], differing only in the policy optimization manner (REINFORCE algorithm with a shared baseline [3, 17] vs. `MP-ASIL`). Table 1 shows that `MP-ASIL` strongly outperforms Cho et al. with the advantage of `MP-ASIL` becoming more pronounced as problem sizes increase. Note that the ATCSR_Rm [60] results are obtained through a greedy

Table 3: Experiment results on JSSP using the TA benchmark. NIH: Neural improvement heuristics. The Time metric is reported in Appendix D.2.

| Method | Type | 15×15 Gap ↓ | 20×15 Gap ↓ | 20×20 Gap ↓ | 30×15 ● Gap ↓ | 30×20 ● Gap ↓ | 50×15 ● Gap ↓ | 50×20 ● Gap ↓ | 100×20 ● Gap ↓ | **Avg.** Gap ↓ |
|---|---|---|---|---|---|---|---|---|---|---|
| Gurobi (3600s)* [20] | Exact | 0.1% | 3.2% | 2.9% | 10.7% | 13.2% | 12.2% | 13.6% | 11.0% | 8.4% |
| OR-Tools (3600s)* [67] | Exact | 0.1% | 0.2% | 0.7% | 2.1% | 2.8% | 3.0% | 2.8% | 3.9% | 2.0% |
| L2D (G)* [11] | NCH | 26.0% | 30.0% | 31.6% | 33.0% | 33.6% | 22.4% | 26.5% | 13.6% | 27.1% |
| L2D (S=128)* [11] | NCH | 17.1% | 23.7% | 22.6% | 24.4% | 28.4% | 17.1% | 20.4% | 10.3% | 20.5% |
| SN (G)* [29] | NCH | 15.3% | 19.4% | 17.2% | 19.1% | 23.7% | 13.9% | 13.5% | 6.7% | 16.1% |
| RASCL (G)* [36] | NCH | 14.3% | 16.5% | 17.3% | 18.5% | 21.5% | 12.2% | 13.2% | 5.9% | 14.9% |
| RS (G)* [39] | NCH | 14.8% | 16.5% | 16.9% | 14.4% | 17.7% | 6.7% | 10.0% | 2.6% | 12.5% |
| SI GD (G)* [45] | NCH | 9.6% | 9.9% | 11.1% | 9.5% | 13.8% | **2.7%** | **6.7%** | 1.7% | 8.4% |
| SLIM (S=512)*† [20] | NCH | **6.5%** | 8.8% | 9.0% | 10.6% | **12.7%** | 4.9% | 7.6% | 2.1% | 7.8% |
| L2S-500* [25] | NIH | 9.3% | 11.6% | 12.4% | 14.7% | 17.5% | 11.0% | 13.0% | 7.9% | 12.2% |
| TBGAT-500* [26] | NIH | 8.0% | 9.9% | 10.0% | 13.3% | 16.4% | 9.6% | 11.9% | 6.4% | 10.7% |
| MP-ASIL (k=512) | NCH | 6.8% | **8.5%** | **8.7%** | **10.4%** | 12.8% | 4.2% | 7.0% | **1.0%** | **7.4%** |

search over 3,146 heuristic parameter configurations for each instance, following the original paper. More detailed results for UPMSP are provided in Appendix D.3. For **FFSP**, we implement `MP-ASIL` on top of the trained MatNet [12] and show remarkably better performance than all baselines.

Table 2 compares `MP-ASIL` with baseline methods on **PFSP** using the well-known Taillard (TA) benchmark [68].[2] We apply `MP-ASIL` to the MatNet-based model. From the table, we observe that previous NCO methods cannot beat classical heuristics; however, `MP-ASIL` considerably surpasses all neural solvers and even exceeds traditional approaches in terms of average Gap and Time. Although direct comparisons are limited by the lack of reported inference times from some neural solvers, `MP-ASIL` solves all instances for each problem size within one second (see Appendix D.2). Appendix D.4 provides experiment results on synthetic PFSP datasets. Table 3 compares `MP-ASIL` with other methods on **JSSP** using the TA benchmark. As shown in the table, SLIM [20], which differs from `MP-ASIL` in learning strategy but uses the same backbone model, already outperforms all learning-based methods in terms of Time and average Gap. Nevertheless, `MP-ASIL` achieves a relative performance improvement of about 5.1% in average Gap. Experimental results on additional JSSP benchmarks are provided in Appendices D.5 and D.6. Due to space limitations, experimental results on deterministic and stochastic **FJSSP** can be found in Appendix D.7.

**Ablation Studies.** Recall that `MP-ASIL` consists of three key components: **(1)** multiple policies represented by latent variables, **(2)** SIL to optimize multiple policies, and **(3)** an advantage weight to control imitation intensity. To validate the contribution of each component to enhanced performance, we conduct ablation studies by progressively removing individual components. Table 4 clearly shows that our full version consistently surpasses all ablation versions by a large margin, highlighting the critical role of each component. We provide ablation results on all test instances in Appendix D.8.

Table 4: Result of ablation studies. AdvW: Advantage weight. MP: Multiple policies. ↑: Performance drop relative to `MP-ASIL`. In the MP-ablation version, latent conditioned policies are replaced by a single policy. In the SIL-ablation version, SIL is substituted with RL approaches: the REINFORCE algorithm with a shared baseline [3, 17] for a single policy or Poppy method [54] for multiple policies. [✗ ✗ ✓] is equivalent to the SLIM [20].

| AdvW | MP | SIL | SMSP 100 | UPMSP 100×6 | PFSP 100×10 | FFSP 100×12 | JSSP 100×20 |
|---|---|---|---|---|---|---|---|
| ✓ | ✓ | ✓ | **0.00%** | **9.57%** | **2.15%** | **1.32%** | **0.96%** |
| ✗ | ✓ | ✓ | 0.67% (0.67% ↑) | 10.30% (0.73% ↑) | 3.56% (1.41% ↑) | 1.37% (0.05% ↑) | 1.75% (0.79% ↑) |
| ✗ | ✗ | ✓ | 3.37% (3.37% ↑) | 14.33% (4.76% ↑) | 3.57% (1.42% ↑) | 1.40% (0.08% ↑) | 2.10% (1.14% ↑) |
| ✗ | ✓ | ✗ | 35.77% (35.77% ↑) | 13.50% (3.93% ↑) | 2.16% (0.01% ↑) | 1.59% (0.27% ↑) | 2.47% (1.51% ↑) |
| ✗ | ✗ | ✗ | 6.69% (6.69% ↑) | 20.16% (10.59% ↑) | 3.78% (1.63% ↑) | 2.02% (0.70% ↑) | 3.96% (3.00% ↑) |

**The Effect of $k$.** We analyze the effect of $k$ on model performance during training and inference. We train our model using different values of $k$, with $k \in \{32, 64, 128\}$ for PFSP and $k \in \{64, 128, 256\}$ for JSSP. We then evaluate the trained models across various inference settings with $k \in \{32, 64, 128, 256, 512\}$. Figure 3 presents the analysis results on the TA benchmark.

---

[2]The best-known results for PFSP and JSSP are obtained from `http://mistic.heig-vd.ch/taillard/`.

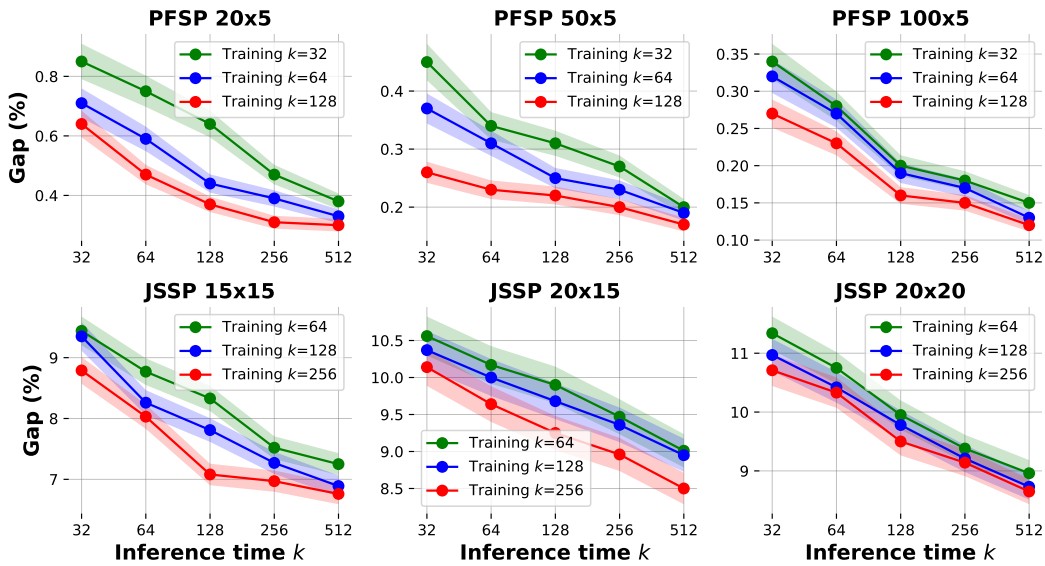

Figure 3: The effect of $k$ on model performance. The results are averaged over ten runs.

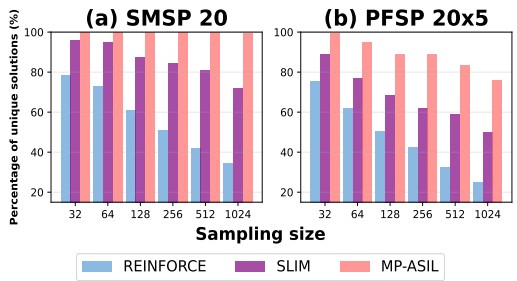

Figure 4: Average percentage of unique solutions. The $x$-axis denotes the number of candidate solutions per instance.

Table 5: Experiment results on TSP 100 and CVRP 100. Poppy uses 16 decoders for TSP 100 and 32 decoders for CVRP 100. d: Days. Other symbols follow definitions provided in Table 1.

| Method | TSP 100 | | CVRP 100 | |
|---|---|---|---|---|
| | Gap ↓ | Time ↓ | Gap ↓ | Time ↓ |
| LKH3* | 0.000% | (8h) | 0.00% | (6d) |
| POMO *† | 0.146% | (1m) | 0.76% | (2m) |
| Sym-NCO *† | 0.180% | (1m) | 0.89% | (2m) |
| Poppy* | 0.07% | (1m) | 0.51% | (5m) |
| MP-ASIL | **0.000%** | (1m) | **0.28%** | (2m) |

From the figure, we can observe that **(1)** training with larger $k$ generates stronger models, and **(2)** increasing $k$ at inference time consistently enhances performance. These findings align with our hypothesis that larger $k$ produces more specialized decision-makers, enabling more extensive solution space exploration and increasing the chance of finding better solutions, albeit with increased memory and computational requirements.

**Exploration Capability.** In this part, we validate the capability of MP-ASIL to generate diverse solutions. For evaluation, we use 1,000 instances for both SMSP 20 and PFSP 20×5. We intentionally choose small-size problems, which represent a challenging scenario for generating diverse solutions [69]. The solution diversity is calculated as the average percentage of unique solutions among $k$ candidates generated per instance, which is a widely used population-level diversity metric [69, 56]. For comparison, we also report the solution diversity of representative single policy approaches, such as REINFORCE with a shared baseline and SLIM. Figure 4 demonstrates that MP-ASIL achieves significantly higher average solution diversity than baselines across all scenarios. Notably, MP-ASIL shows a diversity of 99.4% for SMSP 20 even at $k$=1024, emphasizing its remarkable ability to generate diverse solution patterns. Additionally, we can observe that REINFORCE generates many duplicate solutions at the sampling stage, as pointed out in many recent studies [15, 45, 16].

**Policy Specialization.** We verify that each policy specializes in distinct instance sub-distributions by analyzing how the best-performing latent variable $z$ varies across instances. Specifically, we randomly sample 16 policies, evaluate their performance on batches of instances, and count the

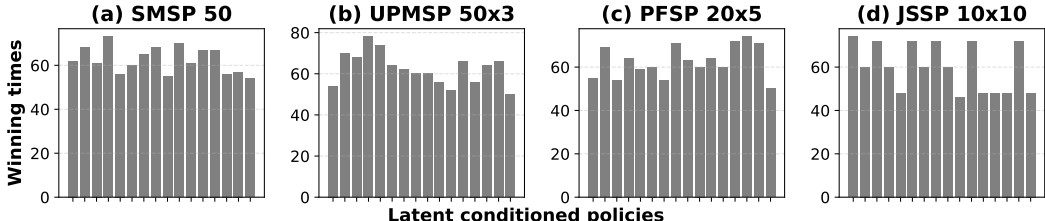

Figure 5: Number of instances where each policy performs best (winning times). 16 policies on the $x$-axis are ordered arbitrarily.

number of instances each policy solves best. From Figure 5, we find that each policy achieves top performance across different instances. These results demonstrate that different policies specialize in producing high-quality solutions for distinct instance sub-distributions, leading to significantly improved overall performance and robustness. To aid better understanding, visualizations of the performance landscape in the policy latent space are provided in Appendix D.9. Additionally, we can search the policy latent space to find promising latent variables for each instance at test time. Details on this approach can be found in Appendix D.10.

**Experiments with Routing.** Finally, to demonstrate the versatility of `MP-ASIL`, we apply it to other COPs, specifically the Traveling Salesman Problem (TSP) and Capacitated Vehicle Routing Problem (CVRP) with 100 nodes (denoted as TSP 100 and CVRP 100), which are extensively studied in the NCO literature. We implement `MP-ASIL` on top of POMO [10], training it on $n = 100$ node instances uniformly distributed in $[0, 1]^2$. We follow the original POMO training hyperparameters (see Appendix F). We compare `MP-ASIL` with state-of-the-art neural solvers, including POMO, Sym-NCO [17] and Poppy [54], using synthetic datasets from [10]. At inference time, `MP-ASIL`, POMO, and Sym-NCO generate $8 \times n(= k)$ solutions for each instance of size $n$ nodes, where eight represents the instance augmentation proposed by [12]. Poppy samples $d$ (number of decoders) $\times n$ solutions for each instance. Unlike baseline methods that enforce distinct initial actions, `MP-ASIL` does not impose different starting points during training and inference. This rollout strategy enables multiple behaviors to freely explore the search space and is universally applicable across all COPs.

Table 5 reports the Gap relative to LKH3 [70] and Time. From the table, we can see that `MP-ASIL` significantly outperforms neural methods on both TSP 100 and CVRP 100. Surprisingly, `MP-ASIL` finds practically optimal solutions for TSP 100 in less than one minute. These results show that `MP-ASIL` can be effective across other COPs. Results for various out-of-distribution VRP scenarios (cross-size, cross-distribution, and cross-problem generalization) are presented in Appendix E.

## 6 Conclusion

In this work, we propose `MP-ASIL`, a generic learning framework for job scheduling. `MP-ASIL` addresses several limitations in RL-based policy gradient methods by enabling multiple policies to autonomously learn diverse and specialized problem-solving strategies without external supervision. We also develop a training scheme to mitigate the suboptimality of self-teaching labels, a fundamental drawback of SIL, and enhance sample efficiency. Last but not least, `MP-ASIL` is agnostic to both network architectures and scheduling problems, allowing its benefits to be realized universally across various problem settings. Extensive experiments demonstrate that `MP-ASIL` achieves new state-of-the-art results on six scheduling problems and significantly outperforms previous neural solvers on routing tasks, highlighting its versatility and broadening the scope of current NCO methods.

## Acknowledgments and Disclosure of Funding

This work was supported in part by the National Research Foundation of Korea (NRF) grant funded by the Korea government (MSIT) (RS-2024-00334171, RS-2025-02216640) and in part by the IITP (Institute of Information Communications Technology Planning Evaluation)-ITRC (Information Technology Research Center) grant funded by the Korea government (Ministry of Science and ICT) (IITP-2025-RS-2024-00437268).

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

# Appendix

# Contents

# A  Job Scheduling Problems

A JSP can be described by a three-field notation $\alpha|\beta|\gamma$ [57]. The $\alpha$ field specifies the machine environment, the $\beta$ field represents constraints, and the $\gamma$ field defines the objective function. Here, we provide representative examples of each field considered in our work. Figures 6, 7, and 8 provide illustrative examples of each field.

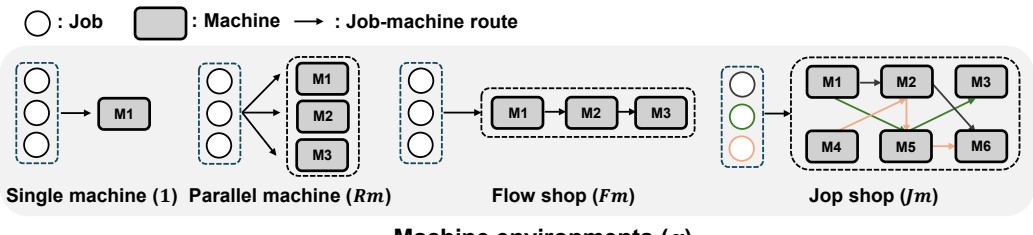

Figure 6: Illustrative examples of machine environments ($\alpha$).

## A.1  Machine Environments $\alpha$

**Single Machine** (1).   There is a single machine, representing the simplest case of machine environments. In this environment, each job $j$ consists of a single operation.

**Parallel Machine** ($Pm$) **/ Unrelated Parallel Machine** ($Rm$).   There are $|\mathcal{M}|$ parallel machines, and each job $j$ is processed by exactly one of these machines. If the processing time of job $j$ varies across different machines, this environment is known as the Unrelated Parallel Machine ($Rm$). Each job $j$ consists of a single operation.

**Flow Shop** ($Fm$).   There are $|\mathcal{M}|$ machines in series, and each job must be processed sequentially on all machines following the same fixed route from machine 1 to machine $|\mathcal{M}|$. If there are $|\mathcal{Q}|$ sequential stages, where $\mathcal{Q}$ is a set of stages, each equipped with parallel machines, the resulting environment is known as the Flexible Flow Shop ($FFc$). In this case, jobs must sequentially pass through each stage, from stage 1 to stage $|\mathcal{Q}|$, being processed by exactly one machine per stage. Each processing step at a machine or stage represents an operation.

**Job Shop** ($Jm$).   There are $|\mathcal{M}|$ machines, and each job $j$ consists of a sequence of operations that must be processed in a predefined, job-specific order. Each operation requires processing on one machine, and when an operation can be processed on one of several alternative machines, the problem setting is referred to as the Flexible Job Shop ($FJc$).

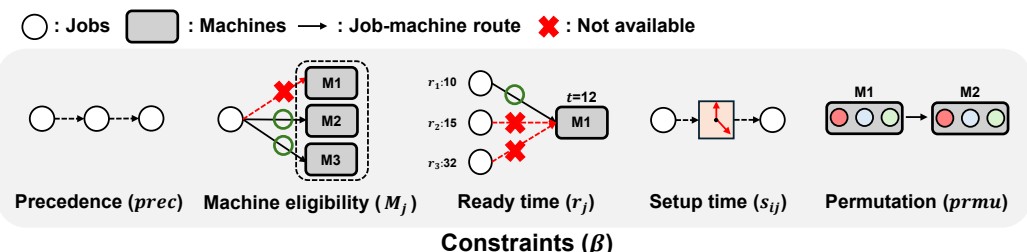

Figure 7: Illustrative examples of constraints ($\beta$).

## A.2  Constraints $\beta$

**Precedence** ($prec$).   The precedence constraints enforce that one or more operations (or jobs) must finish before another operation (or job) can start.

**Machine Eligibility ($M_j$).** Only a subset of the machines $\mathcal{M}_j \subseteq \mathcal{M}$ can process job $j$.

**Ready Time ($r_j$).** Each job $j$ cannot start processing before its ready time $r_j$.

**Sequence Dependent Setup Time ($s_{ij}$).** Switching from job $i$ to job $j$ incurs a setup time. If the setup time between jobs $i$ and $j$ depends on machine $k$, we denote it as $s_{ijk}$.

**Permutation ($prmu$).** The permutation constraint requires that the job processing order determined at the first machine remains unchanged throughout all machines in a flow shop environment.

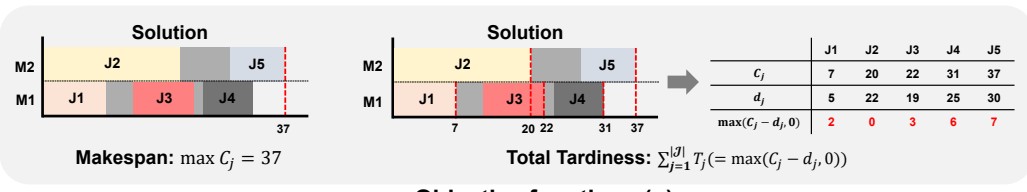

Figure 8: Illustrative examples of objective functions ($\gamma$).

## A.3 Objective Functions ($\gamma$)

**Makespan ($C_{max}$).** The makespan, defined as $\max(C_1, \ldots, C_n)$, where $C_j$ is the completion time of job $j$, denotes the completion time of the last job processed.

**Total Tardiness ($\sum T_j$).** The total tardiness, defined as $\sum T_j = \sum \max(C_j - d_j, 0)$, denotes the sum of job completion delays relative to their due dates $d_j$. When each job $j$ has a weight $w_j$, the objective function becomes $\sum w_j T_j$.

## B    Multi-Policy Representation

### B.1    Implementation of Latent Conditioned Policies

**Overview.** Neural dispatcher architectures generally fall into two main categories: Heavy Encoder Light Decoder (HELD) [9, 10, 12], and Light Encoder Heavy Decoder (LEHD) [11, 71, 72]. HELD-based models employ a computationally expensive encoder once to generate static hidden embeddings, subsequently constructing solutions sequentially using a lightweight decoder. In contrast, LEHD-based models dynamically recompute hidden embeddings at each decision step using multiple decoder layers.

Figure 9 illustrates how latent conditioned policies are applicable to the architectures above. Following previous work [55], HELD-based models concatenate the latent variables with the query, key, and value inputs of the Multi-Head Attention (MHA) layer. LEHD-based models concatenate the latent variables with the final hidden embeddings before computing action probabilities through a Multi-Layer Perceptron (MLP). This means that, regardless of the underlying architecture, multiple policies can be easily implemented by conditioning the decision-making layer's input embeddings on latent variables. The latent variables are sampled once at the start of the solution rollout and remain unchanged during the solution construction. Notably, within each policy, an identical latent variable is concatenated to all hidden embeddings. In this work, we use LEHD-based models for SMSP, UPMSP, JSSP, and FJSSP. In contrast, PFSP and FFSP utilize HELD-based models. In the following sections, we detail how latent conditioned policies are implemented within each architecture.

**HELD.** The encoder generates hidden embeddings for the input instance $s$. Initially, raw operation features $X = \{x_i | i \in \{1, \ldots, |\mathcal{O}|\}\} \in \mathbb{R}^{|\mathcal{O}| \times d_x}$ are projected into $d_h$-dimensional embeddings via a linear layer. These embeddings $H^{(0)} = \{h_i^{(0)} | i \in \{1, \ldots, |\mathcal{O}|\}\} \in \mathbb{R}^{|\mathcal{O}| \times d_h}$ are iteratively updated $L$ times by the update function $\mathcal{F}$ to generate a set of final hidden embeddings $H^{(L)} =$

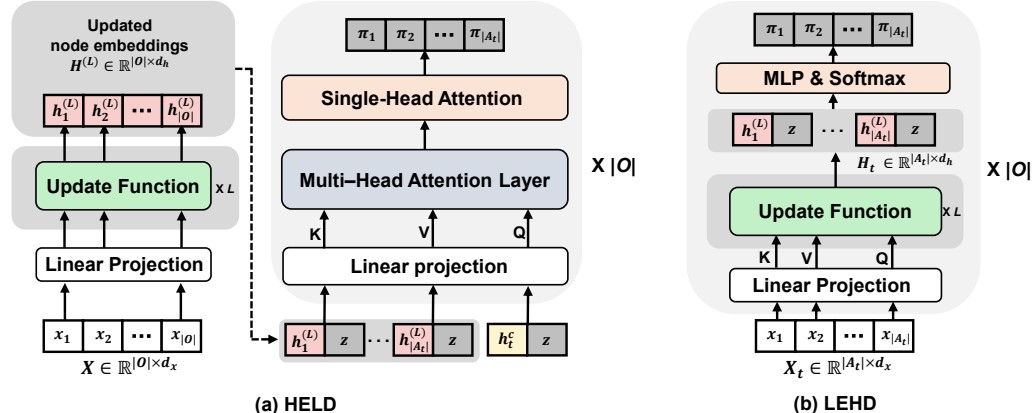

Figure 9: Illustrations of applying the latent conditioned policies into (a) HELD and (b) LEHD.

$\{h_i^{(L)}|i \in \{1, \ldots, |\mathcal{O}|\}\} \in \mathbb{R}^{|\mathcal{O}| \times d_h}$. $\mathcal{F}$ can be designed using any neural network model (e.g., GNN or Transformer).

The decoder sequentially constructs a solution by leveraging the final hidden embeddings, which are the output of the encoder. At each decoding step $t$, the decoder receives hidden embeddings $H_t = \{h_i^{(L)}|i \in \mathcal{A}_t\} \in \mathbb{R}^{|\mathcal{A}_t| \times d_h}$, where $\mathcal{A}_t$ represents the set of feasible operations at step $t$, alongside a context vector $h_t^c$ indicating the current state. To integrate hidden embedding information into the context vector, HELD employs MHA to update the context vector. In this process, latent variables are concatenated with the query, key, and value inputs of the MHA module. The context vector is updated as follows:

$$q_t = W_q \text{Concat}(h_t^c, z), \quad K_t = W_k \text{Concat}(H_t, z), \quad V_t = W_v \text{Concat}(H_t, z), \tag{5}$$

$$h_t^{c'} = \text{MHA}\left(q_t, K_t, V_t\right), \tag{6}$$

where $W_q, W_k, W_v \in \mathbb{R}^{(d_h + d_z) \times d_h}$ are learnable model weights. At the final stage, single-head attention computes a conditional action distribution at step $t$. The action probability $\pi_i$ for operation $i \in \mathcal{A}_t$ is computed by:

$$\pi_i = \text{Softmax}\left(C \cdot \tanh\left(\frac{(h_t^{c'})^T K_t}{\sqrt{d_h}}\right)\right)_i, \tag{7}$$

where $C$ is the clipping parameter and $\tanh$ is the hyperbolic tangent function.

**LEHD.** The LEHD-based models recompute hidden embeddings at every step $t$, effectively capturing dynamic relationships. At step $t$, the raw operation features $X_t = \{x_i|i \in \mathcal{A}_t\} \in \mathbb{R}^{|\mathcal{A}_t| \times d_x}$ are projected into $d_h$-dimensional hidden embeddings via a linear layer. These embeddings $H_t^{(0)} = \{h_i^{(0)}|i \in \mathcal{A}_t\} \in \mathbb{R}^{|\mathcal{A}_t| \times d_h}$ are iteratively updated $L$ times by the update function $\mathcal{F}$ to generate a set of final hidden embeddings $H_t^{(L)}$.

At step $t$, a conditional action distribution is computed via an MLP using the final hidden embeddings $H_t^{(L)}$. In this process, the latent variables are concatenated to the final node embeddings. The action probability $\pi_i$ for operation $i \in \mathcal{A}_t$ is computed by:

$$\pi_i = \text{Softmax}\left(\text{MLP}(\text{Concat}(H_t^{(L)}, z))\right)_i. \tag{8}$$

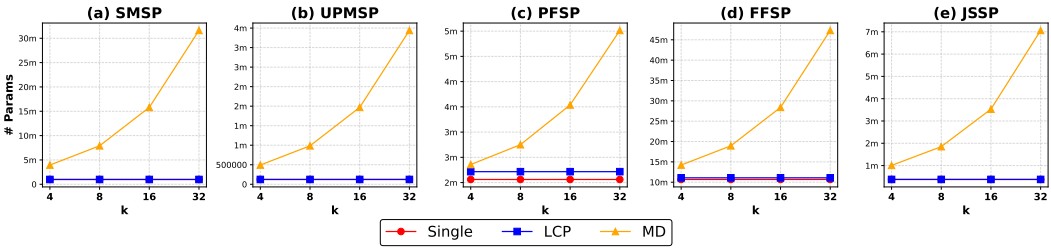

Figure 10: Number of parameters for each strategy. Here, Single denotes a single policy, LCP indicates latent conditioned policies, MD represents a multi-decoder architecture, and $k$ denotes the number of policies.

## B.2 Why We Use Latent Variables to Represent Multiple Policies

Early approaches for representing multiple policies often rely on multi-decoder models without parameter sharing [15, 54]. However, as illustrated in Figure 10, this design leads to substantial parameter growth with the number of policies, limiting scalability and flexibility. In contrast, latent conditioned policies add a fixed number of parameters, regardless of the number of policies. This property enables efficient and scalable representation of multiple policies, which provides the rationale for our design choice.

## B.3 The Effect of Latent Distributions

The latent variable $z$, indexing different policies, can affect model performance based on its distribution $\mathcal{Z}$. To identify an effective prior, we evaluate models trained under three latent distributions: **(1)** $\mathcal{Z}_1 = \mathcal{U}(-1, 1)^{16}$; **(2)** $\mathcal{Z}_2$, a joint distribution combining $\mathcal{U}(-1, 1)^8$ with an 8-dimensional categorical (one-hot) distribution; **(3)** $\mathcal{Z}_3$, a joint distribution combining $\mathcal{U}(-1, 1)^4$ with a 12-dimensional categorical (one-hot) distribution. We do not consider a 16-dimensional categorical (one-hot) distribution because it can only generate 16 policies.

Figure 11 compares performances across these prior distributions on the TA benchmarks. For reference, we also include the results of SLIM and the REINFORCE with a shared baseline as representative single policy approaches. From the figure, we can observe that $\mathcal{Z}_3$ shows robust and strong zero-shot performance. Therefore, we adopt $\mathcal{Z}_3$ as the default prior for all problems except UPMSP, which uses lower-dimensional hidden embedding ($d_h = 64$). For UPMSP, we adopt a joint prior combining $\mathcal{U}(-1, 1)^2$ with a 6-dimensional categorical distribution. However, all prior distributions consistently outperform single policy approaches. This finding underscores the robustness and generalizability of our approach, with its effectiveness stemming from the fundamental model optimization mechanism rather than fine-tuning the latent space.

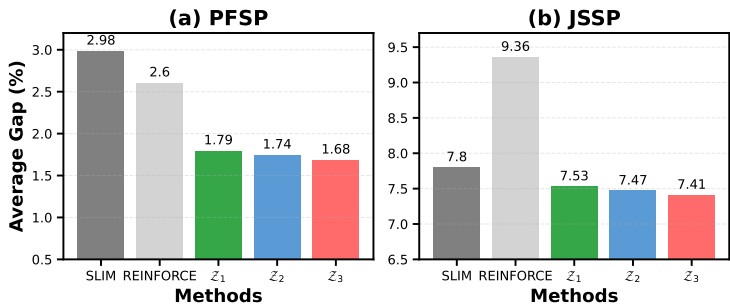

Figure 11: The effect of latent distributions.

# C Benchmark Problems

In this section, we formally define six benchmark problems, following the notations introduced in Sections 3 and Appendix A. We also provide details on the training and inference settings. As mentioned in Section 5, learning-based methods for JSPs have evolved with specialized network architectures and algorithms tailored to each problem variant. Therefore, due to the current absence of foundation models, we implement `MP-ASIL` using different backbone models for each problem type. Naturally, comparisons are conducted against distinct baselines specific to each problem.

## C.1 Single Machine Scheduling Problem ($1||\Sigma w_j T_j$)

**Definition.** The Single Machine Scheduling Problem (SMSP) is one of the most fundamental scheduling problems in which a set of jobs $\mathcal{J}$ must be processed on a single machine. Each job $j \in \mathcal{J}$ has processing time $p_j$, due date $d_j$, and weight $w_j$. The optimization objective is to find a schedule that minimizes the total weighted tardiness $\Sigma w_j T_j$.

**Model Architecture.** Due to the absence of existing neural constructive solvers, we develop a transformer-based solver following the BQ-NCO architecture [72]. In our implementation, nodes represent jobs defined by three features: $p_j$, $w_j$, and remaining deadline slack ($d_j - e_t$), where $e_t$ is the elapsed time at step $t$. The processes for updating hidden embeddings and making decisions follow the approach described in Appendix B.1.

**Training Instances.** Following previous work [27], we randomly generate training instances as follows: $p_j \sim \mathcal{U}(0, 1)$, $w_j \sim \mathcal{U}(0, 1)$, and $d_j \sim \mathcal{U}(0, |\mathcal{J}|)$, where $\mathcal{U}$ represents a continuous uniform distribution. We train our model using instances with $|\mathcal{J}| = 50$.

**Test Instances.** For evaluation, we use 100 SMSP instances for each problem size $|\mathcal{J}| = 50, 100$, and 500 from [27]. The test instances follow the same distribution used for generating the training instances.

**Baseline Algorithms.** We compare `MP-ASIL` with **(1) Handcrafted Heuristics:** Earliest Due Date (EDD) and Ant Colony Optimization (ACO) [27]; **(2) Hybrid Methods:** DeepACO [27] and GFACS [16].

## C.2 Unrelated Parallel Machine Scheduling Problem ($Rm|s_{ijk}, M_j, r_j|\Sigma w_j T_j$)

**Definition.** The Unrelated Parallel Machine Scheduling Problem (UPMSP) is a generalized parallel machine scheduling problem, requiring the assignment and sequencing of a set of jobs $\mathcal{J}$ onto a set of machines $\mathcal{M}$. Each job $j \in \mathcal{J}$ must be processed on exactly one machine selected from its eligible machine set $\mathcal{M}_j \subseteq \mathcal{M}$ with processing time $p_{jk}$ on machine $k \in \mathcal{M}_j$. Each job $j$ has weight $w_j$, due date $d_j$, and ready time $r_j$. Additionally, each machine $k \in \mathcal{M}$ requires setup time $s_{ijk}$. The optimization objective is to find a schedule that minimizes the total weighted tardiness $\Sigma w_j T_j$.

**Model Architecture.** We use the LEHD-based architecture introduced by Cho et al. [41]. We refer readers to the original paper for detailed architectural specifications and features.

**Training Instances.** Following previous work [41], we randomly generate training instances as follows: $|\mathcal{M}_j| \sim \text{Uniform}(\{1, 2, \ldots, |\mathcal{M}|\})$, $p_{jk} \sim \text{Uniform}(\{1, 2, \ldots, 99\})$, $w_j \sim \mathcal{U}(0, 1)$, $s_{ijk} \sim \text{Uniform}(\{0, 1, \ldots, 10\})$, and $r_j \sim \text{Uniform}(\{0, 1, \ldots, \lfloor \hat{p}/2 \rfloor\})$, where $\hat{p} = \frac{1}{|\mathcal{M}|^2} \sum_{k=1}^{|\mathcal{M}|} \sum_{j=1}^{|\mathcal{J}|} p_{jk}$ and Uniform() denotes a discrete uniform distribution. Due dates are sampled as $d_j \sim r_j + \text{Uniform}\left(\{\max(0, \lfloor (\hat{p} - r_j) \times (1 - T - R/2) \rfloor), \ldots, \max(0, \lfloor (\hat{p} - r_j) \times (1 - T + R/2) \rfloor)\}\right)$ using tightness $T$ and range $R$ parameters, which range from 0.2 to 1.0 in steps of 0.2. During training, we uniformly sample $T$ and $R$. We train our model using instances with $|\mathcal{J}| \times |\mathcal{M}| = 25 \times 3$.

**Test Instances.** For evaluation, we use 500 UPMSP instances for each problem size $|\mathcal{J}| \times |\mathcal{M}| = 50 \times 3, 50 \times 6$, and $100 \times 6$ from [41]. The test instances follow the same distribution used for generating the training instances.

**Baseline Algorithms.** We compare `MP-ASIL` with **(1) Handcrafted Heuristics:** EDD and ATCSR_RM [60]; **(2) Neural Constructive Heuristic:** Cho et al. [41].

## C.3 Permutation Flow Shop Scheduling Problem ($Fm|prmu|C_{max}$)

**Definition.** The Permutation Flow shop Scheduling Problem (PFSP) is a widely studied variant of the flow shop scheduling problem, where a set of jobs $\mathcal{J}$ is processed on a sequence of machines. The processing order determined at the first machine remains unchanged across all subsequent machines. Each job $j \in \mathcal{J}$ has a processing time $p_{jk}$ on each machine $k \in \mathcal{M}$. The optimization objective is to find a schedule that minimizes the makespan $C_{max}$.

**Model Architecture.** We use the HELD-based architecture based on MatNet [12]. The model takes as input a $\mathbf{P}_{|\mathcal{J}| \times |\mathcal{M}|}$ matrix, with each element representing processing time $p_{jk}$. In our implementation, the context vector $h_t^c$ at step $t$ is defined as:

$$h_t^c = W_c \text{Concat} \left( h_{t-1}, h_t^u, h^m \right), \tag{9}$$

where $W_c \in \mathbb{R}^{3d_h \times d_h}$ is the learnable model weight, $h_{t-1}$ represents the embedding of the last selected job, $h_t^u$ denotes the mean embedding of unselected jobs up to step $t$, and $h^m$ is obtained by concatenating all machine embeddings and projecting to a $d_h$-dimensional vector. At step $t = 1$, we use a learnable start-token embedding as $h_{t-1}$.

**Training Instances.** We randomly sample $p_{jk}$ from Uniform($\{1, 2, \ldots, 99\}$). We train separate models for two problem sizes: instances with $|\mathcal{J}| \times |\mathcal{M}| = 20 \times 5$ and $20 \times 10$.

**Test Instances.** For evaluation, we use the TA benchmark [68]. Each problem size consists of 10 instances. The benchmark instances follow the same distribution used for generating the training instances. For problems with five machines, we employ the model trained on $20 \times 5$ instances, while for problems with ten machines, we use the model trained on $20 \times 10$ instances.

**Beseline Algorithms.** We compare `MP-ASIL` with **(1) Handcrafted Heuristics:** Iterated Local Search (ILS) [64], Iterated Greedy Algorithm (IGA) [65], and Nawaz-Enscore-Ham (NEH) algorithm [66]; **(2) Neural Constructive Heuristics:** Q-Learning [37], IL [42], and PFSPNet [38].

## C.4 Flexible Flow Shop Scheduling Problem ($FFc||C_{max}$)

**Definition.** The Flexible Flow shop Scheduling Problem (FFSP) involves scheduling a set of jobs $\mathcal{J}$ through multiple sequential stages $\mathcal{Q}$. Each stage $q \in \mathcal{Q}$ comprises unrelated parallel machines $\mathcal{M}_q \subseteq \mathcal{M}$, and each job must be processed by exactly one machine at every stage. Each job $j \in \mathcal{J}$ has a processing time $p_{jqk}$ on machine $k \in M_q$ at stage $q \in \mathcal{Q}$. The optimization objective is to find a schedule that minimizes the makespan $C_{max}$.

**Model Architecture.** We use the HELD-based architecture introduced by PolyNet [56]. We refer readers to the original paper for detailed architectural specifications and features.

**Training Instances.** Following prior work [12], we set $|\mathcal{Q}| = 3$, where each stage consists of $|\mathcal{M}_q| = 4$ machine. Processing times are sampled as $p_{jqk} \sim$ Uniform($\{2, 3, \ldots, 9\}$). We train separate models for three problem sizes: instances with $|\mathcal{J}| \times |\mathcal{M}| = 20 \times 12$, $50 \times 12$, and $100 \times 12$.

**Test Instances.** For evaluation, we use 1,000 FFSP instances for each problem size $|\mathcal{J}| \times |\mathcal{M}| = 20 \times 12$, $50 \times 12$, and $100 \times 12$ from [12]. The test instances follow the same distribution used for generating the training instances.

**Baseline Algorithms.** We compare `MP-ASIL` with **(1) Exact Solver:** CPLEX [61]; **(2) Hand-crafted Heuristics:** Shortest Processing Time (SPT), Genetic Algorithm (GA) [62], and Particle Swarm Optimization (PSO) [63]; **(3) Neural Constructive Heuristics:** MatNet [12] and PolyNet [56].

## C.5 Job Shop Scheduling Problem ($Jm||C_{max}$)

**Definition.** The Job Shop Scheduling Problem (JSSP) is a well-known JSP that has attracted considerable attention from the NCO community. A JSSP instance consists of a set of jobs $\mathcal{J}$ and a set of machines $\mathcal{M}$. Each job $j \in \mathcal{J}$ comprises $m_j$ operations that must be processed in a predefined order. Each operation $O_{ji}$ ($1 \le i \le m_j$) can only be processed on a specific machine with processing time $p_{ji}$. The optimization objective is to find a schedule that minimizes the makespan $C_{max}$.

**Model Architecture.** We use the LEHD-based architecture introduced by SLIM [20]. We refer readers to the original paper for detailed architectural specifications and features.

**Training Instances.** Following [20], we use 5,000 problem instances for each problem size: $|\mathcal{J}| \times |\mathcal{M}|$=10×10, 15×10, 15×15, 20×10, 20×15, and 20×20. Processing times are sampled as $p_{ji} \sim \text{Uniform}(\{1, 2, \dots, 99\})$.

**Test Instances.** For evaluation, we use the TA benchmark [68]. Each problem size consists of 10 instances. The benchmark instances follow the same distribution used for generating the training instances.

**Baseline Algorithms.** We compare `MP-ASIL` with **(1) Exact Solvers:** Gurobi and OR-Tools; **(2) Neural Constructive Heuristics:** L2D [11], SN [29], RASCL [36], SI GD [45], and SLIM [45]; **(3) Neural Improvement Heuristics:** L2S [25] and TBGAT [26].

## C.6 Flexible Job Shop Scheduling Problem ($FJc||C_{max}$)

**Definition.** The Flexible Job Shop Scheduling Problem (FJSSP) is a generalized version of the JSSP, where each operation can be processed on one of the several compatible machines $\mathcal{M}_{ji} \subseteq \mathcal{M}$ with processing time $p_{jik}$ on machine $k \in \mathcal{M}_{ji}$. The optimization objective is to find a schedule that minimizes the makespan $C_{max}$.

**Model Architecture.** We use the LEHD-based architecture introduced by DANIEL [31]. We refer readers to the original paper for detailed architectural specifications and features.

**Training Instances.** Following [31], we randomly sample $|\mathcal{M}_{ji}|$ from $\text{Uniform}(\{1, 2, \dots, |\mathcal{M}|)$ and $\bar{p}_{ji}$ from $\text{Uniform}(\{1, 2, \dots, 20\})$. $p_{ijk}$ is sampled from $\text{Uniform}(\{\max(1, \lfloor 0.8 \times \bar{p}_{ji} \rfloor), \dots, \min(20, \lfloor 1.2 \times \bar{p}_{ji} \rfloor)\})$. Each job has $m_j \sim \text{Uniform}(\{4, 5, 6\})$ operations. We train our model using instances with $|\mathcal{J}| \times |\mathcal{M}| = 10 \times 5$.

**Test Instances.** For evaluation, we use 100 instances for each problem size $|\mathcal{J}| \times |\mathcal{M}| = 10 \times 5$, 20×5, 15×10, 20×10, 30×10, and 40×10 from [31].

**Baseline Algorithms.** We compare `MP-ASIL` with **(1) Exact Solver:** OR-Tools; **(2) Handcrafted Heuristic:** First in First Out (FIFO), Most Operations Remaining (MOR), SPT, and MWKR; **(3) Neural Constructive Heuristics:** HGNN [73], MCGA [74], RS [39], and DANIEL [31].

# D Additional Experiments

In this section, we present supplementary experimental results and analyses. Since `MP-ASIL` can generate different results based on sampled policies, we report the averaged results of three trials.

## D.1 Validation Curves

Figure 12 shows the validation curves for `MP-ASIL`. For comparison, we also report validation results from state-of-the-art methods, including the REINFORCE with a shared baseline [17], SLIM [20], and Poppy [54]. The validation sets consist of 100 instances each for SMSP, UPMSP, PFSP, and FFSP, and 600 instances for JSSP, all sampled from the same distribution as the training instances. Validation is performed at every epoch for JSSP and every 10 epochs for other problems. We train each model from scratch except for FFSP, for which we use publicly available MatNet [12]

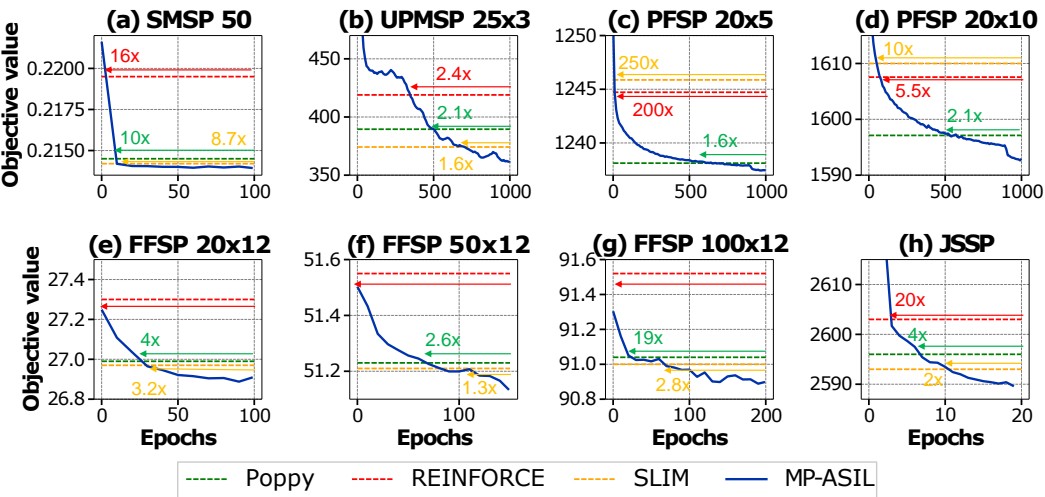

Figure 12: Validation curves.

checkpoints to initialize our policy network.[3] From the figure, we can see that MP-ASIL achieves performance comparable to all baselines using approximately $1.3\times$ to $250\times$ fewer epochs.

## D.2 Computation Time for PFSP and JSSP

Due to space limitations, we report only the Gap in Tables 2 and 3. In this section, we additionally provide Time in Tables 6 and 7. Time results for PFSP are reported only when available from the original papers or when the authors made code available.

Table 6: Inference time for PFSP. Symbols follow definitions provided in Table 1.

| Method | Type | 20×5 Time ↓ | 20×10 Time ↓ | 50×5 ● Time ↓ | 50×10 ● Time ↓ | 100×5 ● Time ↓ | 100×10 ● Time ↓ | 200×10 ● Time ↓ |
|---|---|---|---|---|---|---|---|---|
| ILS [64] | Heuristics | (0.1s) | (0.2s) | (0.2s) | (0.4s) | (0.3s) | (0.9s) | (1.9s) |
| IGA [65] | Heuristics | (0.6s) | (1.1s) | (3.7s) | (7.0s) | (14.9s) | (28.6s) | (1.9m) |
| NEH [66] | Heuristics | (0.1s) | (0.1s) | (0.6s) | (1.1s) | (4.4s) | (8.6s) | (1.1m) |
| IL (G) [42] | NCH | (0.3s) | (0.3s) | (0.4s) | (0.4s) | (0.7s) | (0.7s) | (1.2s) |
| MP-ASIL ($k$=128) | NCH | (0.2s) | (0.2s) | (0.2s) | (0.2s) | (0.3s) | (0.3s) | (0.8s) |

Table 7: Inference time for JSSP. Symbols follow definitions provided in Table 1.

| Method | Type | 15×15 Time ↓ | 20×15 Time ↓ | 20×20 Time ↓ | 30×15 ● Time ↓ | 30×20 ● Time ↓ | 50×15 ● Time ↓ | 50×20 ● Time ↓ | 100×20 ● Time ↓ |
|---|---|---|---|---|---|---|---|---|---|
| Gurobi (3600s)* [20] | Exact | (10h) | (10h) | (10h) | (10h) | (10h) | (10h) | (10h) | (10h) |
| OR-Tools (3600s)* [67] | Exact | (77m) | (8h) | (10h) | (10h) | (10h) | (10h) | (10h) | (10h) |
| L2D (G)* [11] | NCH | (4s) | (6s) | (7s) | (10s) | (14s) | (18s) | (30s) | (1.6m) |
| SN (G)* [29] | NCH | (35s) | (1.1m) | (1.8m) | (2.9m) | (4.7m) | (8.8m) | (16.0m) | (1.2h) |
| RASCL (G)* [36] | NCH | (8s) | (11s) | (14s) | (15s) | (17s) | (28s) | (39s) | (1.6m) |
| RS (G)* [39] | NCH | (5s) | (8s) | (9s) | (19s) | (22s) | (53s) | (1.2m) | (4.6m) |
| L2D (S=128)* [11] | NCH | (6.8m) | (9.0m) | (13.1m) | (15.6m) | (26.3m) | (35.9m) | (1.0h) | (3.4h) |
| SI GD (G)* [45] | NCH | (10s) | (11s) | (11s) | (12s) | (15s) | (24s) | (39s) | (5.0m) |
| SLIM (S=512)*† [20] | NCH | (3s) | (5s) | (7s) | (9s) | (12s) | (22s) | (29s) | (1.5m) |
| L2S-500* [25] | NIH | (1.6m) | (1.7m) | (1.8m) | (2.1m) | (2.3m) | (2.7m) | (3.8m) | (8.4m) |
| TBGAT-500* [26] | NIH | (2.1m) | (2.4m) | (2.9m) | (2.9m) | (3.2m) | (4.0m) | (4.1m) | (7.0m) |
| MP-ASIL ($k$=512) | NCH | (3s) | (5s) | (7s) | (9s) | (12s) | (22s) | (29s) | (1.5m) |

---

[3]https://github.com/yd-kwon/MatNet

## D.3 Detailed UPMSP Results

As detailed in Appendix C.2, the distribution of $d_j$ in UPMSP instances depends on the parameters $T$ and $R$. While the main paper presents average results due to space limitations, here we report detailed results for each parameter combination, evaluating 20 instances per configuration.

Table 8: Experiment results on UPMSP across all parameter combinations. **Bold**: Best Obj. Other symbols follow definitions provided in Table 1.

| $|\mathcal{J}|$ | $|\mathcal{M}|$ | $T$ | $R$ | || | EDD | ATCSR_Rm [60] | Cho et al. (S=6) † [41] | MP-ASIL ($k$=6) |
|---|---|---|---|---|---|---|---|---|
| 50 | 3 | 0.2 | 0.2 | || | 917.72 | **37.35** | 117.92 | 54.76 |
| 50 | 3 | 0.2 | 0.4 | || | 856.02 | **12.92** | 65.1 | 26.82 |
| 50 | 3 | 0.2 | 0.6 | || | 917.8 | **19.85** | 47.96 | 29.59 |
| 50 | 3 | 0.2 | 0.8 | || | 672.76 | 16.19 | 30.76 | **15.14** |
| 50 | 3 | 0.2 | 1.0 | || | 384.16 | 22.18 | 7.14 | **4.72** |
| 50 | 3 | 0.4 | 0.2 | || | 1233.29 | 228.01 | 254.75 | **186.66** |
| 50 | 3 | 0.4 | 0.4 | || | 1506.45 | **134.38** | 209.42 | 179.57 |
| 50 | 3 | 0.4 | 0.6 | || | 1489.27 | 194.58 | 157.27 | **110.32** |
| 50 | 3 | 0.4 | 0.8 | || | 1680.08 | 151.21 | 186.57 | **140.40** |
| 50 | 3 | 0.4 | 1.0 | || | 1283.9 | 164.41 | 160.6 | **137.93** |
| 50 | 3 | 0.6 | 0.2 | || | 2405.19 | 620.78 | 607.09 | **584.25** |
| 50 | 3 | 0.6 | 0.4 | || | 2669.6 | 593.82 | 528.3 | **491.25** |
| 50 | 3 | 0.6 | 0.6 | || | 3154.25 | 759.75 | 621.97 | **616.45** |
| 50 | 3 | 0.6 | 0.8 | || | 2963.12 | 668.87 | **560.29** | 579.01 |
| 50 | 3 | 0.6 | 1.0 | || | 2502.97 | 542.00 | 484.42 | **459.23** |
| 50 | 3 | 0.8 | 0.2 | || | 4078.04 | 1456.9 | 1434.37 | **1356.45** |
| 50 | 3 | 0.8 | 0.4 | || | 4640.32 | 1608.95 | 1499.61 | **1458.93** |
| 50 | 3 | 0.8 | 0.6 | || | 4683.02 | 1726.59 | 1412.5 | **1410.99** |
| 50 | 3 | 0.8 | 0.8 | || | 3750.98 | 1179.13 | 1011.52 | **946.71** |
| 50 | 3 | 0.8 | 1.0 | || | 3381.26 | 985.23 | 875.89 | **842.48** |
| 50 | 3 | 1.0 | 0.2 | || | 5871.35 | 2766.78 | 2600.2 | **2517.76** |
| 50 | 3 | 1.0 | 0.4 | || | 5326.36 | 2494.93 | 2047.97 | **2019.69** |
| 50 | 3 | 1.0 | 0.6 | || | 5049.34 | 2239.92 | 1850.28 | **1805.2** |
| 50 | 3 | 1.0 | 0.8 | || | 4893.06 | 1980.81 | **1500.76** | 1539.71 |
| 50 | 3 | 1.0 | 1.0 | || | 4609.76 | 1325.51 | 1338.26 | **1283.15** |
| **Average** | | | | || | 2836.80 | 876.97 | 784.43 | **751.89** |

| $|\mathcal{J}|$ | $|\mathcal{M}|$ | $T$ | $R$ | || | EDD | ATCSR_Rm [60] | Cho et al. (S=6) † [41] | MP-ASIL ($k$=6) |
|---|---|---|---|---|---|---|---|---|
| 50 | 6 | 0.2 | 0.2 | || | 96.49 | **0.63** | 9.44 | 3.65 |
| 50 | 6 | 0.2 | 0.4 | || | 56.54 | **0.69** | 5.14 | 1.33 |
| 50 | 6 | 0.2 | 0.6 | || | 104.96 | **1.08** | 8.35 | 2.15 |
| 50 | 6 | 0.2 | 0.8 | || | 26.89 | **0.82** | 1.83 | 1.65 |
| 50 | 6 | 0.2 | 1.0 | || | 13.61 | **2.90** | 5.43 | 5.66 |
| 50 | 6 | 0.4 | 0.2 | || | 207.71 | **23.08** | 45.72 | 34.29 |
| 50 | 6 | 0.4 | 0.4 | || | 214.54 | **17.03** | 34.18 | 18.41 |
| 50 | 6 | 0.4 | 0.6 | || | 251.38 | **30.32** | 40.36 | 32.23 |
| 50 | 6 | 0.4 | 0.8 | || | 237.65 | 41.40 | 52.60 | **34.29** |
| 50 | 6 | 0.4 | 1.0 | || | 133.83 | 58.33 | 69.88 | **69.59** |
| 50 | 6 | 0.6 | 0.2 | || | 560.17 | **132.46** | 160.3 | 140.24 |
| 50 | 6 | 0.6 | 0.4 | || | 610.25 | 157.71 | 159.78 | **148.74** |
| 50 | 6 | 0.6 | 0.6 | || | 710.42 | **145.43** | 170.82 | 147.21 |
| 50 | 6 | 0.6 | 0.8 | || | 752.29 | 236.75 | 247.67 | **225.23** |
| 50 | 6 | 0.6 | 1.0 | || | 649.65 | 192.54 | 219.31 | **184.92** |
| 50 | 6 | 0.8 | 0.2 | || | 1358.95 | 507.04 | 474.49 | **462.67** |
| 50 | 6 | 0.8 | 0.4 | || | 1513.27 | 621.03 | 597.84 | **571.43** |
| 50 | 6 | 0.8 | 0.6 | || | 1377.82 | 519.09 | 504.68 | **484.47** |
| 50 | 6 | 0.8 | 0.8 | || | 1288.62 | 468.83 | 447.89 | **418.04** |
| 50 | 6 | 0.8 | 1.0 | || | 1047.54 | 326.61 | 320.17 | **297.40** |
| 50 | 6 | 1.0 | 0.2 | || | 1902.52 | 1175.66 | 1160.46 | **1118.15** |
| 50 | 6 | 1.0 | 0.4 | || | 1839.51 | 912.1 | 930.03 | **888.82** |
| 50 | 6 | 1.0 | 0.6 | || | 1678.61 | 722.82 | 678.23 | **681.27** |
| 50 | 6 | 1.0 | 0.8 | || | 1455.71 | 602.00 | 558.43 | **520.20** |
| 50 | 6 | 1.0 | 1.0 | || | 1372.49 | 466.88 | 451.87 | **464.67** |
| **Average** | | | | || | 778.46 | 294.50 | 294.23 | **275.70** |

| $|\mathcal{J}|$ | $|\mathcal{M}|$ | $T$ | $R$ | || | EDD | ATCSR_Rm [60] | Cho et al. (S=6) † [41] | MP-ASIL ($k$=6) |
|---|---|---|---|---|---|---|---|---|
| 100 | 6 | 0.2 | 0.2 | || | 255.49 | 0.25 | 1.39 | **0.04** |
| 100 | 6 | 0.2 | 0.4 | || | 89.55 | **0.00** | **0.00** | **0.00** |
| 100 | 6 | 0.2 | 0.6 | || | 43.55 | **0.00** | **0.00** | **0.00** |
| 100 | 6 | 0.2 | 0.8 | || | 17.77 | 0.02 | **0.00** | **0.00** |
| 100 | 6 | 0.2 | 1.0 | || | 11.04 | 2.62 | **0.00** | **0.00** |
| 100 | 6 | 0.4 | 0.2 | || | 430.48 | 32.65 | 22.94 | **10.17** |
| 100 | 6 | 0.4 | 0.4 | || | 881.5 | 48.43 | 15.84 | **9.99** |
| 100 | 6 | 0.4 | 0.6 | || | 563.9 | 25.30 | 7.29 | **0.79** |
| 100 | 6 | 0.4 | 0.8 | || | 517.33 | 27.44 | 12.92 | **8.76** |
| 100 | 6 | 0.4 | 1.0 | || | 306.99 | 49.9 | 20.56 | **12.20** |
| 100 | 6 | 0.6 | 0.2 | || | 1497.45 | 327.21 | 150.37 | **148.92** |
| 100 | 6 | 0.6 | 0.4 | || | 1898.12 | 314.28 | 129.59 | **110.07** |
| 100 | 6 | 0.6 | 0.6 | || | 2163.50 | 167.17 | 138.09 | **122.61** |
| 100 | 6 | 0.6 | 0.8 | || | 2738.14 | 561.34 | 301.71 | **272.11** |
| 100 | 6 | 0.6 | 1.0 | || | 1560.70 | 284.26 | 206.50 | **177.32** |
| 100 | 6 | 0.8 | 0.2 | || | 3880.05 | 1152.10 | 722.26 | **636.25** |
| 100 | 6 | 0.8 | 0.4 | || | 5125.19 | 1361.08 | 1002.46 | **915.62** |
| 100 | 6 | 0.8 | 0.6 | || | 4449.80 | 912.5 | 863.04 | **770.21** |
| 100 | 6 | 0.8 | 0.8 | || | 3896.58 | 901.39 | 644.09 | **539.94** |
| 100 | 6 | 0.8 | 1.0 | || | 2409.26 | 452.19 | 348.51 | **313.65** |
| 100 | 6 | 1.0 | 0.2 | || | 7073.31 | 3718.23 | 2655.76 | **2507.50** |
| 100 | 6 | 1.0 | 0.4 | || | 6527.61 | 3209.26 | 2040.39 | **1941.05** |
| 100 | 6 | 1.0 | 0.6 | || | 5800.50 | 2089.76 | 1377.43 | **1258.89** |
| 100 | 6 | 1.0 | 0.8 | || | 4949.04 | 1314.94 | 1057.43 | **977.17** |
| 100 | 6 | 1.0 | 1.0 | || | 4716.41 | 1436.42 | 840.95 | **719.93** |
| **Average** | | | | || | 2472.13 | 735.55 | 502.37 | **458.11** |

## D.4 Experiment Results on PFSP Using Synthetic Datasets

In the PFSP benchmark results presented in Table 2, IL [42] is trained on a distribution different from that of the benchmark. Following the IL paper setting, we compare `MP-ASIL` with IL on synthetic datasets sampled from a Gamma distribution ($\alpha = 1$, $\theta = 2$).

**Setup.** IL is trained on instances with $|\mathcal{J}| \times |\mathcal{M}| = 20 \times 5$ to imitate solutions generated by NEH [66], a highly specialized PFSP solver. We also train `MP-ASIL` on small-size instances ($|\mathcal{J}| \times |\mathcal{M}| = 20 \times 5$) sampled from the Gamma distribution, using the same hyperparameters detailed in Table 23. For evaluation, we use 1,000 instances each for $20 \times 5$, $50 \times 5$, and $100 \times 5$, and 100 instances each for $200 \times 5$, $500 \times 5$, and $1000 \times 5$ from [42]. We also include classical heuristics (Iterated Local Search (ILS) [64], Iterated Greedy Algorithm (IGA) [65], and NEH [66]) as baselines.

**Remark.** IL does not directly provide datasets in its code repository but offers code for dataset generation.[4] Thus, we generate datasets and reproduce the evaluation results by running the trained IL model provided by the authors.

**Results.** Table 9 shows that `MP-ASIL` achieves new state-of-the-art results on these datasets. Surprisingly, `MP-ASIL`, trained exclusively on $20 \times 5$ instances, demonstrates strong cross-size generalization, outperforming NEH on all problem sizes except $100 \times 5$, while achieving computational speedups of $4 \times$ to $360 \times$. This superior performance of `MP-ASIL` can be attributed to its self-evolutionary approach. While IL faces a performance ceiling due to imitating suboptimal solutions, `MP-ASIL` autonomously generates and learns from self-teaching labels. Thus, `MP-ASIL` can break the fundamental limitation of classic SL methods, which strongly depend on the performance of their teacher algorithms.

Table 9: Experiment results on PFSP using synthetic datasets. Gap: Performance gap relative to NEH. Other symbols follow definitions provided in Table 1.

| Method | PFSP 20×5 | | | PFSP 50×5 ● | | | PFSP 100×5 ● | | |
|---|---|---|---|---|---|---|---|---|---|
| | Obj. ↓ | Gap ↓ | Time ↓ | Obj. ↓ | Gap ↓ | Time ↓ | Obj. ↓ | Gap ↓ | Time ↓ |
| ILS | 29.96 | 2.74% | (9s) | 65.33 | 4.38% | (18s) | 121.49 | 3.63% | (36s) |
| IGA | 29.05 | -0.37% | (1m) | 63.19 | 0.96% | (5m) | 118.08 | 0.72% | (25m) |
| NEH | 29.16 | 0.00% | (4s) | 62.59 | 0.00% | (56s) | 117.24 | 0.00% | (7m) |
| IL | 31.93 | 9.50% | (0s) | 68.05 | 8.72% | (1s) | 125.34 | 6.91% | (2s) |
| `MP-ASIL` ($k = 128$) | **28.43** | **-2.50%** | (1s) | **62.48** | **-0.18%** | (2s) | **117.37** | **0.11%** | (8s) |

| Method | PFSP 200×5 ● | | | PFSP 500×5 ● | | | PFSP 1000×5 ● | | |
|---|---|---|---|---|---|---|---|---|---|
| | Obj. ↓ | Gap ↓ | Time ↓ | Obj. ↓ | Gap ↓ | Time ↓ | Obj. ↓ | Gap ↓ | Time ↓ |
| ILS | 230.12 | 2.92% | (7s) | 551.77 | 2.36% | (19s) | 1076.35 | 2.07% | (39s) |
| IGA | 224.85 | 0.56% | (10m) | 540.86 | 0.34% | (1h) | 1056.55 | 0.19% | (4.2h) |
| NEH | 223.60 | 0.00% | (9m) | 539.06 | 0.00% | (1.5h) | 1054.50 | 0.00% | (12h) |
| IL | 234.87 | 5.04% | (1s) | 566.96 | 5.18% | (4s) | 1115.22 | 5.76% | (21s) |
| `MP-ASIL` ($k = 128$) | **223.23** | **-0.15%** | (3s) | **537.31** | **-0.32%** | (20s) | **1049.17** | **-0.51%** | (2m) |

## D.5 Experiment Results on JSSP Using Lawrence's Benchmark

**Setup.** Lawrence's (LA) benchmark [75], widely used for evaluating generalization capability in JSSP research papers [26, 20], consists of five instances for each of the eight different problem sizes. To evaluate the generalization performance of `MP-ASIL`, we compare it with the baseline methods listed in Table 3, excluding approaches that do not provide results on the LA benchmark.

**Results.** Table 10 demonstrates that `MP-ASIL` achieves the best average Gap (Avg.) among neural constructive heuristics, finding optimal solutions for LA 15×5, LA 20×5, and LA 30×10. Moreover, `MP-ASIL` solves all instances within one second per problem size, clearly outperforming L2S-500 and showing comparable performance to TBGAT-500, both known to require significantly longer

---
[4]https://github.com/lokali/PFSS-IL

computation times as reported in [26] (e.g., `MP-ASIL` requires 0.8 seconds for LA 15×15 instances, whereas L2S-500 and TBGAT-500 demand approximately one minute).

Table 10: Experiment results on JSSP using the LA benchmark. Symbols follow definitions provided in Table 1.

| Method | LA 10×5 • | LA 10×10 | LA 15×5 • | LA 15×10 | LA 15×15 | LA 20×5 • | LA 20×10 | LA 30×10 • | Avg. |
|---|---|---|---|---|---|---|---|---|---|
| Gurobi (3600s)* [20] | 0.0% | 0.0% | 0.0% | 0.0% | 0.0% | 0.0% | 0.0% | 0.0% | 0.0% |
| OR-Tools (3600s)* [67] | 0.0% | 0.0% | 0.0% | 0.0% | 0.0% | 0.0% | 0.0% | 0.0% | 0.0% |
| MWKR* [20] | 16.0% | 12.2% | 5.5% | 17.8% | 18.2% | 5.2% | 17.2% | 8.6% | 12.6% |
| L2D (G)* [11] | 14.3% | 23.7% | 7.8% | 27.2% | 27.1% | 6.3% | 24.6% | 8.4% | 17.4% |
| SN (G)* [29] | 12.1% | 11.9% | 2.7% | 14.6% | 16.1% | 3.6% | 15.7% | 3.1% | 10.0% |
| L2D (S=128)* [11] | 8.8% | 10.4% | 2.8% | 16.2% | 17.4% | 3.1% | 18.3% | 6.8% | 10.6% |
| SLIM (S=512)*† [20] | **1.1%** | 2.5% | **0.0%** | 5.0% | 5.6% | **0.0%** | 5.6% | **0.0%** | 2.5% |
| L2S-500* [25] | 2.1% | 4.4% | **0.0%** | 6.4% | 7.3% | **0.0%** | 7.0% | 0.2% | 3.4% |
| TBGAT-500* [26] | 2.1% | **1.8%** | **0.0%** | **3.6%** | **5.5%** | **0.0%** | 5.0% | **0.0%** | **2.3%** |
| `MP-ASIL` ($k = 512$) | 1.3% | 2.1% | **0.0%** | 4.3% | **5.5%** | **0.0%** | 4.9% | **0.0%** | **2.3%** |

## D.6 Experiment Results on JSSP Using Synthetic Datasets

For JSSP, we also compare `MP-ASIL` with search-based NCO methods using synthetic JSSP datasets commonly used in the NCO literature.

**Setup.** Our test uses three sets of 100 instances each for 10×10, 15×15, and 20×15 from [52]. Baseline methods include L2D [11], Poppy [54], EAS [52], COMPASS [55], and TBGAT-500 [26]. L2D and Poppy use naive stochastic sampling, whereas EAS fine-tunes its policy individually for each test instance. COMPASS improves solution quality through CMA-based policy space search (detailed in Section D.10), and TBGAT-500 employs a GNN-based local search operator to iteratively update solutions for 500 iterations.

**Results.** Table 11 presents the average makespan (Obj.), Gap relative to OR-Tools [67], and Time. From the table, we can find that `MP-ASIL` significantly outperforms all search-based methods in solution quality and computational speed, with total inference times under one minute per test set. Notably, compared to COMPASS, which also learns latent conditioned policies but conducts extensive inference time search, `MP-ASIL` reduces the Gap using one-shot inference alone, from 4.7% to **2.7%** on JSSP 10×10, from 8.0% to **6.1%** on JSSP 15×15, and from 10.4% to **8.2%** on JSSP 20×15.

Table 11: Experiment results on JSSP using synthetic datasets. $d$ denotes the number of decoders. Other symbols follow definitions provided in Table 1.

| Method | JSSP 10×10 | | | JSSP 15×15 | | | JSSP 20×15 | | |
|---|---|---|---|---|---|---|---|---|---|
| | Obj. ↓ | Gap ↓ | Time ↓ | Obj. ↓ | Gap ↓ | Time ↓ | Obj. ↓ | Gap ↓ | Time ↓ |
| OR-Tools* | 807.6 | 0.0% | (37s) | 1188.0 | 0.0% | (3h) | 1345.5 | 0.0% | (80h) |
| L2D* | 871.7 | 8.0% | (8h) | 1378.3 | 16.0% | (25h) | 1624.6 | 20.8% | (40h) |
| Poppy ($d = 16$)* | 849.7 | 5.2% | (3h) | 1290.4 | 8.6% | (5h) | 1495.7 | 11.2% | (8h) |
| EAS* | 858.4 | 6.3% | (5h) | 1295.2 | 9.0% | (9h) | 1498.0 | 11.3% | (11h) |
| COMPASS* | 845.5 | 4.7% | (3h) | 1282.8 | 8.0% | (5h) | 1485.6 | 10.4% | (8h) |
| TBGAT-500* | – | 2.7% | (16m) | – | 6.7% | (21m) | – | 9.3% | (23m) |
| `MP-ASIL` ($k$=512) | **829.7** | **2.7%** | (12s) | **1260.9** | **6.1%** | (28s) | **1456.6** | **8.2%** | (40s) |

## D.7 Experiment results on FJSSP

**Deterministic FJSSP.** For FJSSP, one of the most complex scheduling problems, we apply `MP-ASIL` on top of DANIEL [31], a state-of-the-art neural FJSSP solver. For comparison, we also include **(1) Exact Solver:** OR-Tools (1800s); **(2) Handcrafted Heuristic:** First in First Out (FIFO), Most Operations Remaining (MOR), SPT, and MWKR; **(3) Neural Constructive Heuristics:** HGNN [73], MCGA [74], RS [39], and DANIEL [31] as baselines. Table 12 reports the Gap relative to OR-Tools (1800s) and Time. From the table, we can see that `MP-ASIL` significantly enhances the performance of DANIEL and outperforms all other methods. Remarkably, on large-scale instances unseen during

training ($20\times10$, $30\times10$, and $40\times10$), `MP-ASIL` even surpasses OR-Tools (1800s) while requiring $79\times$ to $291\times$ shorter computation times.

Table 12: Experiment results on FJSSP. Symbols follow the definitions provided in Table 1.

| Method | Type | $10\times5$ Gap ↓ | Time ↓ | $20\times5\bullet$ Gap ↓ | Time ↓ | $15\times10\bullet$ Gap ↓ | Time ↓ | $20\times10\bullet$ Gap ↓ | Time ↓ | $30\times10\bullet$ Gap ↓ | Time ↓ | $40\times10\bullet$ Gap ↓ | Time ↓ |
|---|---|---|---|---|---|---|---|---|---|---|---|---|---|
| OR-Tools (1800s)* | Exact | 0.00% | (50h) | 0.00% | (50h) | 0.00% | (50h) | 0.00% | (50h) | 0.00% | (50h) | 0.00% | (50h) |
| FIFO | Heuristics | 24.06% | (16s) | 14.87% | (32s) | 28.65% | (51s) | 19.22% | (1.2m) | 19.50% | (1.8m) | 16.67% | (2.5m) |
| MOR | Heuristics | 19.87% | (16s) | 13.85% | (32s) | 20.68% | (51s) | 12.20% | (1.2m) | 15.57% | (1.8m) | 15.13% | (2.5m) |
| SPT | Heuristics | 34.76% | (16s) | 22.56% | (32s) | 38.22% | (51s) | 30.25% | (1.2m) | 27.47% | (1.8m) | 21.66% | (2.5m) |
| MWKR | Heuristics | 17.58% | (16s) | 11.51% | (32s) | 19.41% | (51s) | 10.30% | (1.2m) | 13.96% | (1.8m) | 13.37% | (2.5m) |
| HGNN (S=100)* | NCH | 9.66% | (1.9m) | 10.31% | (3.9m) | 12.13% | (6.6m) | 9.64% | (10.7m) | 12.36% | (21.3m) | 12.26% | (40.9m) |
| MCGA (S=100)* | NCH | 9.01% | – | 8.36% | – | 11.77% | – | 7.70% | – | 12.44% | – | 12.50% | – |
| RS (S=100)* | NCH | 7.26% | – | 7.22% | – | 9.59% | – | 6.06% | – | 11.14% | – | 11.29% | – |
| DANIEL (S=100)*† | NCH | 5.57% | (1.2m) | 2.46% | (3.1m) | 6.79% | (6.5m) | -1.03% | (10.2m) | 4.43% | (20.6m) | 3.77% | (37.6m) |
| MP-ASIL (k=100) | NCH | **3.00%** | (1.2m) | **0.67%** | (3.1m) | **4.61%** | (6.5m) | **-3.00%** | (10.3m) | **-0.15%** | (20.7m) | **-0.59%** | (37.8m) |

Additionally, we evaluate our approach on well-known FJSSP benchmarks, including Brandimarte [76] and Hurink [77] datasets. The Hurink benchmark consists of edata, rdata, and vdata instances, with operations assignable to 1-2 machines, 1-3 machines, and $1\text{-}|\mathcal{M}|$ machines, respectively. We also include Genetic Programming (GP) [78], a representative HH, as a baseline. GP is a widely used methodology among dispatching rule generation HH, evolving populations of individual tree structures to automatically discover effective problem-solving strategies. This approach shares the same spirit as our method from the perspective of *heuristics to generate heuristics*. As shown in Table 13, `MP-ASIL` substantially surpasses all baselines, demonstrating robust and superior performance in out-of-distribution scenarios.

Table 13: Experiment results on FJSSP using four benchmark datasets. Symbols follow the definitions provided in Table 1.

| Method | Type | Brandimarte Gap ↓ | Hurink (edata) Gap ↓ | Hurink (vdata) Gap ↓ | Hurink (rdata) Gap ↓ |
|---|---|---|---|---|---|
| MWKR | Heuristics | 28.91% | 18.60% | 4.25% | 13.86% |
| GP | HH | 12.13% | – | – | – |
| HGNN (S=100)* | NCH | 18.56% | 8.71% | 1.32% | 5.57% |
| MCGA (S=100)* | NCH | 18.67% | 8.38% | 1.40% | 5.71% |
| RS (S=100)* | NCH | 15.40% | 7.90% | 0.70% | 4.72% |
| DANIEL (S=100)*† | NCH | 9.53% | 9.08% | 0.69% | 4.95% |
| MP-ASIL (k=100) | NCH | **7.32%** | **7.24%** | **0.50%** | **4.66%** |

**Stochastic FJSSP.** `MP-ASIL` is also applicable to stochastic scheduling problems. For stochastic problems, we can evaluate solution quality through expected makespan $\mathbb{E}(C_{max})$ or Value-at-Risk $VaR_\alpha(C_{max})$ metrics by applying the model's decisions identically to multiple scenarios sampled from probability distributions. Thus, `MP-ASIL` can evaluate each policy's decisions through scenario sets, enabling easy application in the same way as deterministic problems. To validate this approach, we perform experiments on the stochastic FJSSP, where processing times are random variables. We apply `MP-ASIL` to the Scenario Processing Module (SPM)-DAN [79], which introduces an attention-based SPM to solve stochastic FJSSP. Specifically, we implement latent conditioned policies by concatenating latent variables to the decision-making layer (MLP) input as detailed in Appendix B.1. We employ the identical `MP-ASIL` training procedure described in the main text. Our training objective is to minimize $VaR_\alpha(C_{max})$. Training hyperparameters and experimental settings follow the original paper, with training conducted on only $10\times5$ instances. Baseline methods include four dispatching rules (FIFO, MOR, SPT, and MWKR) and SPM-DAN.

Tables 14 and 15 present experiment results on various stochastic FJSSP datasets. Each problem size comprises 100 instances from SPM-DAN. From Table 14, we observe that `MP-ASIL` demonstrates clear performance improvement over SPM-DAN and significantly outperforms all dispatching rules, but requires slightly more computation time. Moreover, for different objective functions $\mathbb{E}(C_{max})$, as shown in Table 15, `MP-ASIL` consistently achieves the best performance without retraining.

Through these experimental results, we verify that `MP-ASIL` can be effective for stochastic scheduling problems.

Table 14: Experiment results on stochastic FJSSP. The objective function is $VaR_\alpha(C_{max})$. We set $\alpha = 95\%$, following the original paper. Symbols follow the definitions provided in Table 1.

| Method | 10x5 | | | 20x5 ● | | | 15x10 ● | | | 20x10 ● | | |
|---|---|---|---|---|---|---|---|---|---|---|---|---|
| | Obj. ↓ | Gap ↓ | Time ↓ | Obj. ↓ | Gap ↓ | Time ↓ | Obj. ↓ | Gap ↓ | Time ↓ | Obj. ↓ | Gap ↓ | Time ↓ |
| FIFO | 757.19 | 13.00% | (40s) | 1308.89 | 6.98% | (1.3m) | 1215.23 | 14.46% | (2.2m) | 1448.87 | 11.41% | (2.8m) |
| MOR | 753.22 | 12.41% | (40s) | 1326.69 | 8.43% | (1.3m) | 1182.71 | 11.40% | (2.2m) | 1435.38 | 10.37% | (2.8m) |
| SPT | 820.38 | 22.43% | (40s) | 1427.94 | 16.71% | (1.3m) | 1309.28 | 23.32% | (2.2m) | 1485.49 | 14.27% | (2.8m) |
| MWKR | 741.49 | 10.66% | (40s) | 1317.16 | 7.66% | (1.3m) | 1155.40 | 8.82% | (2.2m) | 1419.13 | 9.12% | (2.7m) |
| SPM-DAN | 670.08 | 0.00% | (1.8m) | 1223.49 | 0.00% | (6.2m) | 1061.71 | 0.00% | (14.7m) | 1300.53 | 0.00% | (25.3m) |
| MP-ASIL | **654.34** | **-2.35%** | (1.8m) | **1194.25** | **-2.39%** | (6.2m) | **1026.87** | **-3.28%** | (15m) | **1230.77** | **-5.07%** | (25.9m) |

Table 15: Experiment results on stochastic FJSSP. The objective function is $\mathbb{E}(C_{max})$. Symbols follow the definitions provided in Table 1.

| Method | 10x5 | | | 20x5 ● | | | 15x10 ● | | | 20x10 ● | | |
|---|---|---|---|---|---|---|---|---|---|---|---|---|
| | Obj. ↓ | Gap ↓ | Time ↓ | Obj. ↓ | Gap ↓ | Time ↓ | Obj. ↓ | Gap ↓ | Time ↓ | Obj. ↓ | Gap ↓ | Time ↓ |
| FIFO | 645.35 | 10.18% | (40s) | 1171.82 | 4.98% | (1.3m) | 1074.24 | 12.75% | (2.2m) | 1301.00 | 9.16% | (2.7m) |
| MOR | 641.50 | 9.52% | (40s) | 1186.68 | 6.32% | (1.3m) | 1044.47 | 9.63% | (2.2m) | 1288.04 | 8.07% | (2.7m) |
| SPT | 705.50 | 20.45% | (40s) | 1280.55 | 12.76% | (1.3m) | 1167.27 | 22.52% | (2.2m) | 1485.49 | 14.27% | (2.7m) |
| MWKR | 632.44 | 7.98% | (40s) | 1179.92 | 5.71% | (1.3m) | 1020.88 | 7.15% | (2.2m) | 1275.73 | 7.04% | (2.7m) |
| SPM-DAN | 585.74 | 0.00% | (1.8m) | 1116.19 | 0.00% | (6.3m) | 952.73 | 0.00% | (14.7m) | 1191.87 | 0.00% | (25.4m) |
| MP-ASIL | **570.54** | **-2.60%** | (1.8m) | **1091.14** | **-2.24%** | (6.3m) | **926.65** | **-2.74%** | (14.9m) | **1139.56** | **-4.39%** | (25.8m) |

## D.8 Ablation Studies

In this section, we present comprehensive ablation results for all problems evaluated in this study, using the ablation variants defined in Section 5.2. As shown in Figure 13, `MP-ASIL` performs best in 22 out of 24 benchmark datasets. Specifically, `MP-ASIL` significantly outperforms Poppy (AdvW, SIL ablation) across all benchmark problems, highlighting the effectiveness of `MP-ASIL` in optimizing multiple policies compared to RL. Furthermore, `MP-ASIL` shows superior performance compared to SLIM (AdvW, MP ablation), the state-of-the-art SIL-based method, on all benchmark datasets except JSSP 15×15 and JSSP 30×30. Additionally, removing Advantage Weight (AdvW ablation) substantially drops performance across all datasets, underscoring the importance of our proposed training scheme that dynamically adjusts imitation intensity for self-labels.

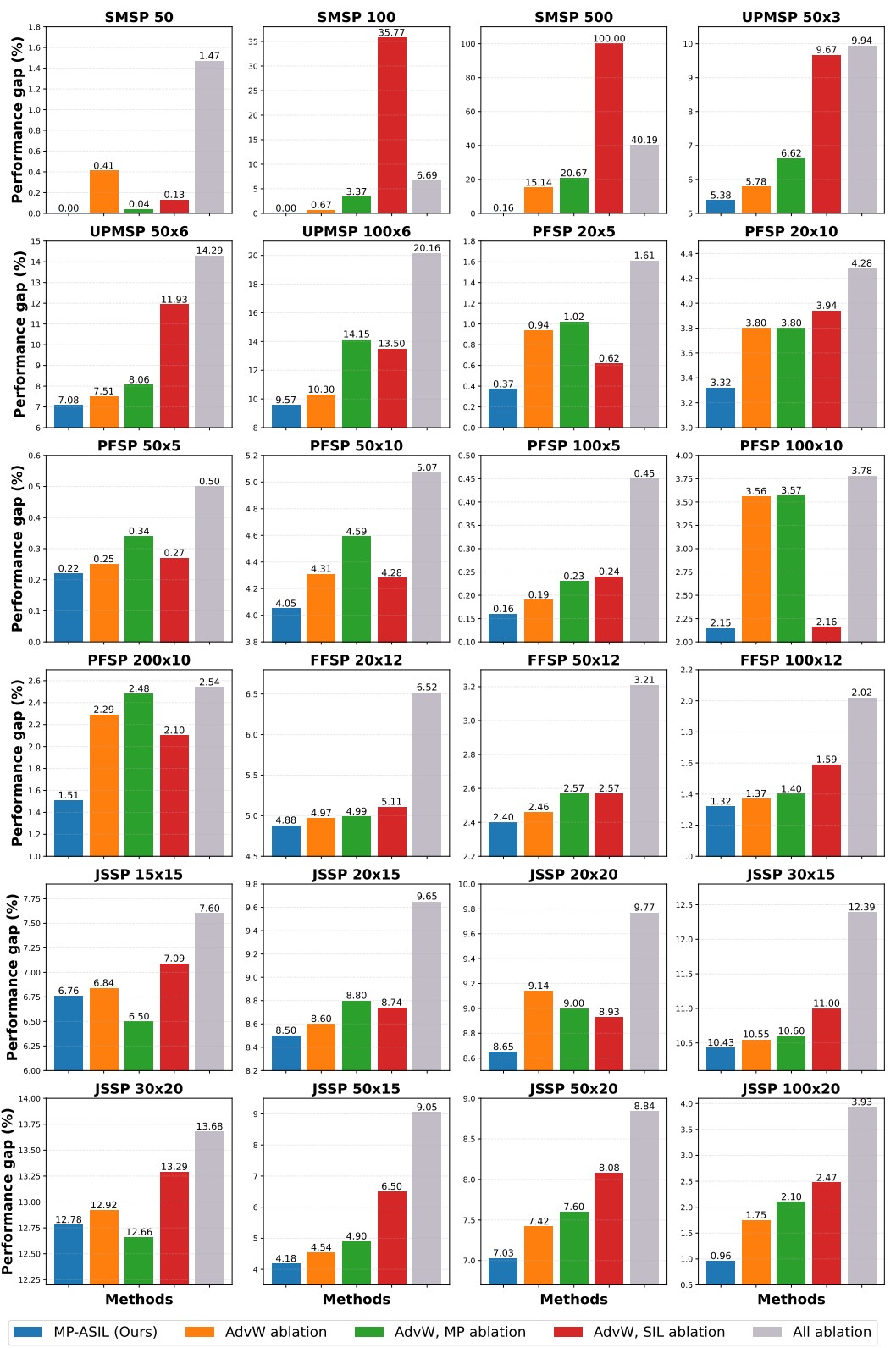

Figure 13: Results of ablation studies. Gaps greater than 100% are truncated to 100%.

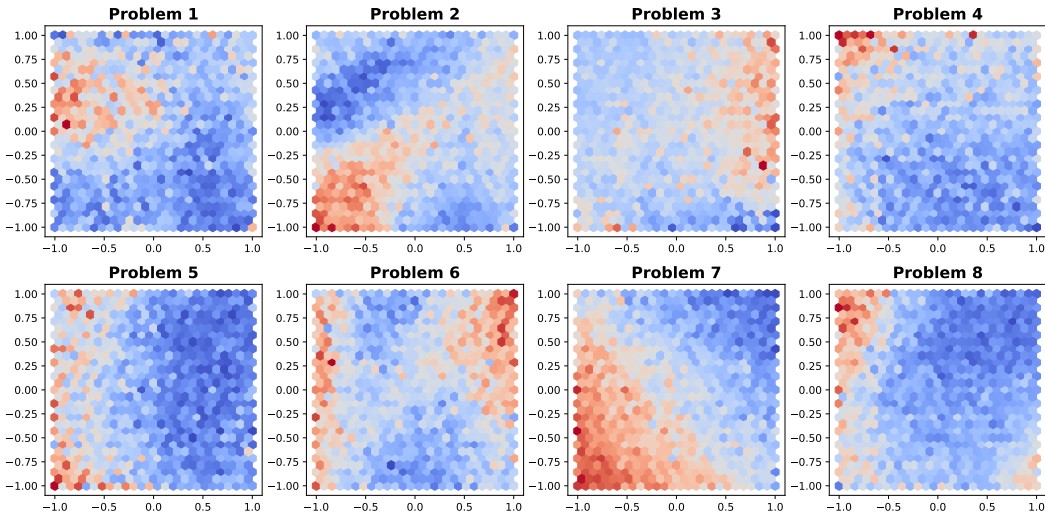

Figure 14: Policy latent space heat maps on 8 problem instances. Red-colored regions correspond to low-performing latent regions, whereas blue-colored regions show high-performing areas.

## D.9   Performance Landscape Visualization

In this section, we visualize the performance landscape of a two-dimensional policy latent space. To this end, we train the model with a two-dimensional latent space bounded in $[-1, 1]$. We then evaluate 32,000 randomly sampled latent vectors on eight TA PFSP instances of size $20 \times 5$ and report the results in Figure 14. From the figure, we observe that: **1)** the performance landscape is instance-dependent and **2)** high-performing regions differ across various instances. These findings confirm that different policy regions specialize in generating superior solutions for distinct instance subsets. These conclusions also motivate the application of search techniques to find high-performing latent space regions during inference, with the results presented in the subsequent section.

## D.10   Policy Latent Space Search at Inference Time

Randomly sampling latent variables at inference time does not guarantee that promising latent variables will be included (although increasing the number of samples can improve the likelihood). However, at inference time, we can apply a principled search procedure to find high-performing latent variables. Recently, COMPASS [55] introduced a policy latent space search method using the Covariance Matrix Adaptation (CMA) [80] evolutionary algorithm. In this section, we use CMA at test time to search for promising latent areas on a per-instance basis.

Following the COMPASS paper, we employ three independent CMA components in parallel with 1,600 search attempts and report the results in Table 16. For comparison, we present results that apply CMA to latent-conditioned policies trained via Poppy [54] instead of `MP-ASIL`, which exactly matches the original COMPASS training setup. Poppy is an RL-based method. In addition, we report sampling-based results (`MP-ASIL + SAM`), where the number of samples is set equal to the number of search attempts.

Table 16: Performance evaluation results combined with CMA on the TA benchmark. ↓: Lower is better.

| Method | PFSP 20×10 Gap ↓ | JSSP 20×15 Gap ↓ |
|---|---|---|
| Poppy + CMA | 2.27% | 7.35% |
| `MP-ASIL + SAM` | 2.04% | 6.98% |
| `MP-ASIL + CMA` | **1.63%** | **6.37%** |

From the table, we can see that `MP-ASIL + SAM` already outperforms Poppy + CMA without the search method. Furthermore, `MP-ASIL + CMA` achieves significant relative performance improvements of 25.1% on PFSP 20×10 and 9.6% on JSSP 20×15 compared to `MP-ASIL + SAM`. These results highlight the effectiveness of `MP-ASIL` in optimizing multi-policy compared to the RL-based method and show that performance can be significantly improved through search procedures to identify promising regions in the latent space.

# E    Vehicle Routing Problems

In this section, we present performance evaluation results for various out-of-distribution scenarios in VRPs (cross-size, cross-distribution, and cross-problem generalization). These experiments aim to demonstrate that `MP-ASIL` can show good generalization capabilities across diverse COPs.

## E.1    Cross-size Generalization

This part presents the cross-size generalization performance of `MP-ASIL` on TSP and CVRP.

**Setup.**    For evaluation, we use three datasets of 1,000 instances each with $n = 125$, 150, and 200 from [52] and compare `MP-ASIL` with the baseline methods listed in Table 5. All methods generate solutions following the procedure detailed in Section 5.2, employing models trained on instances with $n = 100$. As before, `MP-ASIL` does not enforce distinct initial actions.

**Results.**    Table 17 summarizes the experimental results for TSP and CVRP instances of various sizes. From the table, we can observe that `MP-ASIL` significantly outperforms state-of-the-art NCO methods across all sizes, with the performance advantage increasing as instance size grows for both problems. These results confirm that `MP-ASIL` demonstrates remarkable cross-size generalization capabilities across various COPs.

Table 17: Experiment results on TSP and CVRP. Symbols follow definitions provided in Table 1.

|  | Method | $n = 125$ ● | | | $n = 150$ ● | | | $n = 200$ ● | | |
|---|---|---|---|---|---|---|---|---|---|---|
|  |  | Obj. ↓ | Gap ↓ | Time ↓ | Obj. ↓ | Gap ↓ | Time ↓ | Obj. ↓ | Gap↓ | Time ↓ |
| TSP | LKH3* | 8.583 | 0.000% | (73m) | 9.346 | 0.000% | (99m) | 10.687 | 0.000% | (3h) |
|  | POMO* †[10] | 8.607 | 0.278% | (<1m) | 9.397 | 0.542% | (<1m) | 10.843 | 1.457% | (1m) |
|  | Sym-NCO †[17] | 8.619 | 0.413% | (<1m) | 9.402 | 0.599% | (<1m) | 10.849 | 1.516% | (1m) |
|  | Poppy ($d$=16)* [54] | 8.594 | 0.14% | (<1m) | 9.372 | 0.27% | (<1m) | – | – | – |
|  | MP-ASIL | **8.585** | **0.028%** | (<1m) | **9.359** | **0.143%** | (<1m) | **10.798** | **1.041%** | (1m) |
| CVRP | LKH3* | 17.50 | 0.00% | (19h) | 19.22 | 0.00% | (20h) | 22.00 | 0.00% | (25h) |
|  | POMO*†[10] | 17.73 | 1.29% | (<1m) | 19.64 | 2.18% | (1m) | 22.90 | 4.12% | (1m) |
|  | Sym-NCO†[17] | 17.72 | 1.23% | (<1m) | 19.61 | 2.03% | (<1m) | 22.78 | 3.54% | (1m) |
|  | Poppy ($d$=32)* [54] | 17.63 | 0.74% | (1m) | 19.50 | 1.46% | (1m) | – | – | – |
|  | MP-ASIL | **17.62** | **0.70%** | (<1m) | **19.46** | **1.23%** | (<1m) | **22.57** | **2.58%** | (1m) |

Furthermore, to validate performance on larger-scale TSP datasets (TSP 100, TSP 200, TSP 500, and TSP 1000 from [45]), we implement `MP-ASIL` with BQ-NCO [72] and perform evaluations. Baseline methods include BQ-NCO trained via SL and BQ-NCO trained through SI GD [45], another SIL-based approach. The results are presented in Table 18. The table shows that `MP-ASIL` does not surpass the baseline approaches on TSP 100, TSP 200, and TSP 1000. However, `MP-ASIL` remains easy to implement, without requiring labeled data for SL or extensive search and hyperparameter tuning needed for SI GD (as mentioned in Section 2). To improve performance, a discussion on this topic is provided in the Limitations section (Appendix G).

Table 18: Experiment results on large-size TSP. beam: Beam search. Other symbols follow definitions provided in Table 1.

| Method | $n = 100$ | $n = 200$ ● | $n = 500$ ● | $n = 1000$ ● |
|---|---|---|---|---|
|  | Gap ↓ | Gap ↓ | Gap ↓ | Gap ↓ |
| BQ-NCO beam16 (SL)*† | **0.02%** | **0.09%** | 0.43% | **0.91%** |
| BQ-NCO beam16 (SI GD)*† | **0.02%** | 0.10% | 0.46% | 1.01% |
| MP-ASIL ($k = 16$) | **0.02%** | 0.14% | **0.41%** | 1.19% |

## E.2    Cross-distribution Generalization

This part discusses the cross-distribution generalizability of `MP-ASIL` on TSP.

**Setup.** For evaluation, we use four TSP 100 datasets with different distributions, including uniform, clustered, explosion, and implosion from INViT [81]. Each dataset contains 2,000 instances. As baseline methods, we include **RL-based approaches** (POMO [10], PointerFormer [18], Omni-TSL [82], ELG [83], and INViT [81]) and **SL-based methods** (LEHD [71] and BQ-NCO [72]), following the baseline methods used in INViT. In our experiments, we employ the trained POMO models used in Section 5.2. At test time, we use $k = 100$ policies for each instance to generate solutions and also apply data augmentation to enhance overall performance.

**Results.** Table 19 shows the performance on TSP 100 datasets from four different distributions. The Gap is computed relative to Gurobi. The table demonstrates that MP-ASIL substantially surpasses all NCO methods across all distributions while maintaining efficient runtime. Particularly, while the second-best and third-best methods vary across datasets, MP-ASIL consistently maintains top performance. We attribute this robustness to MP-ASIL's ability to learn multiple specialized behaviors. Even when some policies perform poorly on specific distributions, others can achieve superior performance and provide complementary strengths, thereby demonstrating excellent and robust performance under distribution shift scenarios.

Table 19: Experiment results on four TSP 100 datasets. Symbols follow definitions provided in Table 1.

| Method | Uniform | | Clustered ● | | Explosion ● | | Implosion ● | |
|---|---|---|---|---|---|---|---|---|
| | Gap ↓ | Time ↓ | Gap ↓ | Time ↓ | Gap ↓ | Time ↓ | Gap ↓ | Time ↓ |
| Gurobi* | 0.00% | (23.8m) | 0.00% | (34.4m) | 0.00% | (28.3m) | 0.00% | (28.7m) |
| POMO*† | 1.29% | (2.0m ) | 3.89% | (1.7m) | 1.42% | (1.7m) | 1.44% | (1.7m) |
| PointerFormer* | 0.43% | (1.7m) | 3.96% | (1.7m) | 0.87% | (1.7m) | 0.71% | (1.7m) |
| Omni-TSL* | 2.55% | (2.0m) | 3.62% | (2.0m) | 3.21% | (2.0m) | 2.67% | (2.0m) |
| ELG* | 0.51% | (3.2m) | 3.69% | (2.3m) | 0.93% | (3.5m) | 0.85% | (3.2m) |
| INViT-2V* | 1.65% | (3.0m) | 3.12% | (2.9m) | 1.85% | (3.1m) | 1.95% | (2.9m) |
| INViT-3V* | 0.95% | (4.2m) | 2.47% | (4.0m) | 1.12% | (4.3m) | 1.21% | (4.0m) |
| LEHD* | 0.57% | (11.5m) | 4.51% | (14.9m) | 0.68% | (11.1m) | 1.17% | (18.3m) |
| BQ-NCO* | 5.90% | (16.6m) | 8.86% | (17.4m) | 6.41% | (18.0m) | 6.40% | (16.8m) |
| MP-ASIL ($k = 100$) | **0.00%** | (1.0m) | **1.05%** | (1.0m) | **0.06%** | (1.0m) | **0.21%** | (1.0m) |

### E.3 Cross-problem Generalization

In real-world industrial applications, cross-problem generalization is as important as cross-size and cross-distribution generalization since practical problems typically involve various attributes and constraints. In this part, we verify the ability of MP-ASIL to address this critical requirement.

**Setup.** Figure 15 summarizes the attributes considered in our experiments. Each problem variant is derived by incorporating one or more additional attributes into the standard CVRP formulation (e.g., CVRP + Time Window (TW) = CVRPTW, CVRP + Open route (O) = OVRP, and CVRP + TW + O + Backhaul (B) = OVRPBTW).

- **Time Windows (TW)**: The vehicle must visit node $i$ within the time window from $e_i$ to $l_i$, and each node has a service time $s_i$. If the vehicle arrives at node $i$ before $e_i$, it must wait until $e_i$ to begin service.
- **Open Routes (O)**: The vehicle is not required to return to the depot after completing visits to all nodes.
- **Backhaul (B)**: The classical CVRP assumes that demand is positive (linehaul) for delivery. In practice, however, customers can have negative demand (backhaul) for pickup. The linehaul and backhaul customers may coexist on the same route.
- **Duration Limit (L)**: The total length of each route must not exceed a predetermined threshold.

We build MP-ASIL upon POMO-MTL [84], a transformer-based multi-task VRP solver. Except for the policy optimization strategy (a single policy with POMO vs. MP-ASIL) and the rollout strategy,

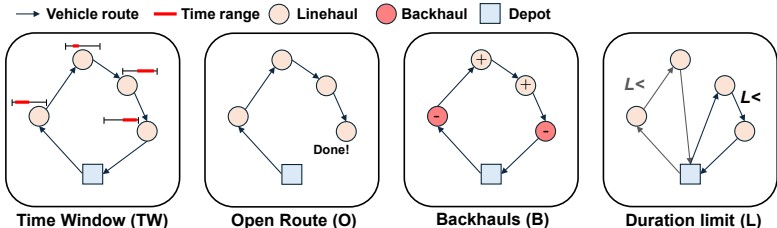

Figure 15: Illustrations of various attributes of VRPs.

we follow the same training setup, evaluation procedure, and test datasets as in POMO-MTL. During both training and inference, we do not enforce distinct initial actions. This strategy is particularly critical in complex VRPs, where initial actions can significantly influence solution quality, rendering rollouts from every node less suitable; however, this aspect is frequently overlooked in current studies [56].

**Results.** Table 20 shows the performance evaluation results for five trained VRPs and five unseen VRPs. Performance gaps (Gap) are computed relative to HGS [85], a state-of-the-art handcrafted VRP solver. For trained VRPs, the table demonstrates that MP-ASIL significantly outperforms POMO-MTL across all VRP variants, reducing the Gap from 1.71% to **1.48%** on CVRP 100, from 3.81% to **3.70%** on VRPTW 100, from 4.48% to **4.01%** on OVRP 100, from 3.58% to **2.80%** on VRPB 100, and from 1.66% to **1.23%** on VRPL 100.

For unseen VRPs, MP-ASIL shows remarkable cross-problem generalization, surpassing POMO-MTL across nearly all instances. Specifically, MP-ASIL achieves performance improvements by reducing the Gap from 4.50% to **3.77%** on VRPBL 100, from 3.05% to **3.01%** on VRPBTW 100, and from 11.50% to **10.86%** on OVRPBLTW 100, while performing slightly worse only on OVRPL 100 (4.57% vs. 4.90%). These results strongly support the capability of MP-ASIL to effectively generalize across diverse out-of-distribution scenarios, underscoring its effectiveness in both single-task and multi-task settings.

Table 20: Experiment results on ten VRP variants using synthetic datasets. Each problem includes 5,000 test instances. Symbols follow definitions provided in Table 1.

(a) Trained VRPs

| Method | CVRP 100 | | | VRPTW 100 | | | OVRP 100 | | |
|---|---|---|---|---|---|---|---|---|---|
| | Obj. ↓ | Gap ↓ | Time ↓ | Obj. ↓ | Gap ↓ | Time ↓ | Obj. ↓ | Gap ↓ | Time ↓ |
| HGS* | 15.54 | 0.00% | 14h | 26.14 | 0.00% | 14h | 9.71 | 0.00% | 14h |
| POMO-MTL*† | 15.80 | 1.71% | 35s | 27.13 | 3.81% | 35s | 10.14 | 4.48% | 35s |
| MP-ASIL | **15.77** | **1.48%** | 36s | **27.11** | **3.70%** | 36s | **10.10** | **4.01%** | 36s |

| Method | VRPB 100 | | | VRPL 100 | | | AVG. | | |
|---|---|---|---|---|---|---|---|---|---|
| | Obj. ↓ | Gap ↓ | Time ↓ | Obj. ↓ | Gap ↓ | Time ↓ | Obj. ↓ | Gap ↓ | Time ↓ |
| HGS* | 11.13 | 0.00% | (14h) | 15.54 | 0.00% | (14h) | 15.612 | 0.00% | (14h) |
| POMO-MTL*† | 11.53 | 3.58% | (35s) | 15.80 | 1.66% | (35s) | 16.27 | 3.05% | (35s) |
| MP-ASIL | **11.44** | **2.80%** | (36s) | **15.73** | **1.23%** | (36s) | **16.03** | **2.68%** | (36s) |

(b) Unseen VRPs

| Method | VRPBL 100 ● | | | OVRPL 100 ● | | | VRPBTW 100 ● | | |
|---|---|---|---|---|---|---|---|---|---|
| | Obj. ↓ | Gap ↓ | Time ↓ | Obj. ↓ | Gap ↓ | Time ↓ | Obj. ↓ | Gap ↓ | Time ↓ |
| HGS* | 11.15 | 0.00% | (14h) | 9.71 | 0.00% | (14h) | 26.31 | 0.00% | (14h) |
| POMO-MTL*† | 11.65 | 4.50% | (35s) | **10.15** | **4.57%** | (35s) | 27.11 | 3.05% | (35s) |
| MP-ASIL | **11.57** | **3.77%** | (36s) | 10.19 | 4.90% | (36s) | **27.10** | **3.01%** | (36s) |

| Method | OVRPLTW 100 ● | | | OVRPBTW 100 ● | | | AVG. ● | | |
|---|---|---|---|---|---|---|---|---|---|
| | Obj. ↓ | Gap ↓ | Time ↓ | Obj. ↓ | Gap ↓ | Time ↓ | Obj. ↓ | Gap ↓ | Time ↓ |
| HGS* | 17.35 | 0.00% | (14h) | 17.31 | 0.00% | (14h) | 16.36 | 0.00% | (14h) |
| POMO-MTL*† | 19.34 | 11.50% | (35s) | 19.32 | 11.61% | (35s) | 17.51 | 7.03% | (35s) |
| MP-ASIL | **19.23** | **10.86%** | (36s) | **19.29** | **11.44%** | (36s) | **17.47** | **6.78%** | (36s) |

# F   Training Hyperparameters

In this section, we provide training hyperparameters, closely following the original papers. Our experimental results in this paper show that simply applying `MP-ASIL` without hyperparameter tuning achieves significantly improved performance while maintaining comparable training and inference time. This demonstrates the practical value and ease of adoption of our method, enabling practitioners to seamlessly replace policy gradient methods without requiring extensive hyperparameter calibration or algorithmic redesign.

**SMSP.**   Due to the lack of neural constructive solver for SMSP, we develop the model based on BQ-NCO [72].

Table 21: Hyperparameter setting for SMSP. h: Hours.

|                                        | SMSP        |
| -------------------------------------- | ----------- |
| Learning Rate (LR)                     | 1e-4        |
| Weight decay                           | 1e-6        |
| The number of encoder layers           | 1           |
| The number of decoder layers           | 5           |
| The number of attention heads          | 8           |
| Hidden embedding dimension ($d_h$)     | 128         |
| Batch-size                             | 50          |
| Continuous latent variable dimension   | 4           |
| Categorical latent variable dimension  | 12          |
| The number of policies (Training)      | 128         |
| Epochs                                 | 100         |
| Optimizer                              | Adam        |
| LR scheduler                           | MultiStepLR |
| LR milestones                          | [90,100]    |
| LR gamma                               | 0.1         |
| Epoch size                             | 1,000       |
| Training time                          | ~7h         |

**UPMSP.**   We follow the training settings from [41].

Table 22: Hyperparameter setting for UPMSP. d: Days.

|                                        | UPMSP    |
| -------------------------------------- | -------- |
| Learning Rate (LR)                     | 5e-4     |
| Weight decay                           | –        |
| The number of encoder layers           | 1        |
| The number of decoder layers           | 3        |
| The number of attention heads          | 8        |
| Hidden embedding dimension ($d_h$)     | 64       |
| Batch-size                             | 32       |
| Continuous latent variable dimension   | 2        |
| Categorical latent variable dimension  | 6        |
| The number of policies (Training)      | 32       |
| Epochs                                 | 1,000    |
| Optimizer                              | Adam     |
| LR scheduler                           | Constant |
| Epoch size                             | 256      |
| Training time                          | ~3d      |

**PFSP.** We follow the training settings from [12].

Table 23: Hyperparameter setting for PFSP.

|  | **PFSP** |
|---|---|
| Learning Rate (LR) | 1e-4 |
| Weight decay | 1e-6 |
| The number of encoder layers | 6 |
| The number of decoder layers | 1 |
| The number of attention heads | 8 |
| Hidden embedding dimension ($d_h$) | 128 |
| Batch-size | 200 |
| Continuous latent variable dimension | 4 |
| Categorical latent variable dimension | 12 |
| The number of policies (Training) | 128 |
| Epochs | 1,000 |
| Optimizer | Adam |
| LR scheduler | MultiStepLR |
| LR milestones | [900, 950] |
| LR gamma | 0.1 |
| Epoch size | 100,000 |
| Training time | $\sim$1.2d |

**FFSP.** We follow the training settings from [12].

Table 24: Hyperparameter setting for FFSP 20$\times$12, 50$\times$12, and 100$\times$12.

|  | **FFSP 20$\times$12** | **FFSP 50$\times$12** | **FFSP 100$\times$12** |
|---|---|---|---|
| Learning Rate (LR) | 1e-5 | 1e-5 | 1e-5 |
| Weight decay | 1e-6 | 1e-6 | 1e-6 |
| The number of encoder layers | 3 | 3 | 3 |
| The number of decoder layers | 1 | 1 | 1 |
| The number of attention heads | 16 | 16 | 16 |
| Hidden embedding dimension ($d_h$) | 256 | 256 | 256 |
| Batch-size | 50 | 50 | 50 |
| Continuous latent variable dimension | 4 | 4 | 4 |
| Categorical latent variable dimension | 12 | 12 | 12 |
| The number of policies (Training) | 24 | 24 | 24 |
| Epochs | 100 | 150 | 200 |
| Optimizer | Adam | Adam | Adam |
| LR scheduler | MultiStepLR | MultiStepLR | MultiStepLR |
| LR milestones | [80,90] | [130,140] | [170,190] |
| LR gamma | 0.1 | 0.1 | 0.1 |
| Epoch size | 1,000 | 1,000 | 1,000 |
| Training time | $\sim$2h | $\sim$5h | $\sim$1d |

**JSSP.** We follow the training settings from [20].

Table 25: Hyperparameter setting for JSSP.

|  | **JSSP** |
| --- | --- |
| Learning Rate (LR) | 1e-4 |
| Weight decay | – |
| The number of encoder layers | 2 |
| The number of decoder layers | 2 |
| The number of attention heads | 3 |
| Hidden embedding dimension ($d_h$) | 128 |
| Batch-size | 16 |
| Continuous latent variable dimension | 4 |
| Categorical latent variable dimension | 12 |
| The number of policies (Training) | 256 |
| Epochs | 20 |
| Optimizer | Adam |
| LR scheduler | Constant |
| Epoch size | 30,000 |
| Training time | ∼4d |

**FJSSP.** We follow the training settings from [31].

Table 26: Hyperparameter setting for FJSSP.

|  | **FJSSP** |
| --- | --- |
| Learning Rate (LR) | 3e-4 |
| Weight decay | – |
| The number of encoder layers | 1 |
| The number of decoder layers | 2 |
| The number of attention heads | 4 |
| Hidden embedding dimension ($d_h$) | 64 |
| Batch-size | 16 |
| Continuous latent variable dimension | 4 |
| Categorical latent variable dimension | 12 |
| The number of policies (Training) | 128 |
| Epochs | 40 |
| Optimizer | Adam |
| LR scheduler | Constant |
| Epoch size | 160 |
| Training time | ∼5h |

**TSP and CVRP.** We follow the training settings from [10].

Table 27: Hyperparameter setting for TSP and CVRP.

|  | TSP | CVRP |
|---|---|---|
| Learning rate | 1e-4 | |
| Weight decay | 1e-6 | |
| The number of encoder layers | 6 | |
| The number of decoder layers | 1 | |
| The number of attention heads | 8 | |
| Hidden embedding dimension ($d_h$) | 128 | |
| Batch-size | 50 | |
| Continuous latent variable dimension | 4 | |
| Discrete latent variable dimension | 12 | |
| The number of policies (Training) | 100 | |
| Epochs | 2,000 | 8,000 |
| Optimizer | Adam | Adam |
| LR scheduler | MultiStepLR | MultiStepLR |
| LR milestones | [1900,1950] | [7900, 7950] |
| LR gamma | 0.1 | 0.1 |
| Epoch size | 100,000 | 10,000 |
| Training time | ∼12d | ∼4d |

## G   Limitation and Future Work

**Algorithm.** Despite the demonstrated effectiveness of MP-ASIL, our approach has several potential limitations. First, MP-ASIL samples latent variables from a fixed prior distribution $\mathcal{Z}$ across all instances, potentially overlooking optimal instance-specific priors. Therefore, learning adaptive, instance-dependent distributions is a promising future direction. Second, while increasing the number of policies ($k$) can make stronger models during training (detailed in Section 5.2), it also increases memory usage and training time. Therefore, an important avenue for future research is to design a practical yet effective sampling framework that can generate stronger self-teachers from fewer samples drawn from promising policy subspaces.

**Applications.** In this work, we primarily focus on deterministic and static JSPs. For future research, we plan to apply MP-ASIL to more realistic scenarios, including communication latency [86] and dynamic environments, as well as multi-objective JSPs. Additionally, we aim to demonstrate MP-ASIL's effectiveness across a broader range of COPs.

## H   Broader Impact

This paper introduces a new learning paradigm for scheduling problems. MP-ASIL addresses diverse decision-making tasks in manufacturing and logistics via end-to-end learning, potentially reducing human reliance on effective heuristic algorithm design. However, unlike simple PDRs, deep learning lack interpretability, raising trust concerns for AI-driven decision-making systems. Therefore, advancing explainable AI methods to elucidate and justify decision-making processes remains a critical avenue for future research.

## I   Licenses

The licenses for code repositories and datasets used in this work are summarized in Table 28.

Table 28: List of licenses for code repositories and datasets used in this work.

| Resource | Type | Link | License |
|---|---|---|---|
| BQ-NCO | Code | https://github.com/naver/bq-nco | CC BY-NC-SA 4.0 |
| DeepACO and ACO | Code& dataset | https://github.com/henry-yeh/DeepACO | MIT License |
| GFACS | Code | https://github.com/ai4co/gfacs | MIT License |
| POMO | Code | https://github.com/yd-kwon/POMO | MIT License |
| MatNet | Code& dataset | https://github.com/yd-kwon/MatNet | MIT License |
| IL, ILS, IGA, and NEH | Code | https://github.com/lokali/PFSS-IL | Available online |
| SLIM | Code& dataset | https://github.com/AndreaCorsini1/SelfLabelingJobShop | Available online |
| DANIEL | Code& dataset | https://github.com/wrqccc/FJSP-DRL | Available online |
| SPM-DAN | Code | https://github.com/ai-for-decision-making-tue/NCO-for-Stochastic-FJSP | Available online |
| Sym-NCO | Code | https://github.com/alstn12088/Sym-NCO | Available online |
| EAS | Dataset | https://github.com/ahottung/EAS | Available online |
| SI GD | Dataset | https://github.com/grimmlab/gumbeldore/tree/main | Available online |
| INViT | Dataset | https://github.com/Kasumigaoka-Utaha/INViT | Available online |
| POMO-MTL | Code& Dataset | https://github.com/FeiLiu36/MTNCO | Available online |

