# OpenReview forum: "Towards Generalizable Multi-Policy Optimization with Self-Evolution for Job Scheduling"
_NeurIPS.cc/2025/Conference — NeurIPS 2025 poster_

### Official Review · Reviewer_9EdW · 2025-06-30

**Clarity:** 3
**Significance:** 2
**Originality:** 2
**Rating:** 4
**Confidence:** 4

**Summary:**

This paper proposes a novel self-labeling approach for combinatorial optimization problems (COPs) aimed at enhancing solution diversity. The authors tackle the challenge that a limited set of solutions can lead to suboptimal training, either due to convergence to a suboptimal policy or poor sample efficiency. To address this, they introduce MP-ASIL, which employs a multi-policy framework by conditioning in the latent space, enabling the use of a single deep neural network (DNN) to represent multiple policies. Furthermore, the authors design a loss function that weights the quality of each label based on normalized advantage weighting. MP-ASIL is evaluated on five scheduling problems and two routing problems, demonstrating superior performance over existing baselines in most cases.

**Questions:**

1. Would your approach also work for more complex scheduling problems, like the flexible job scheduling problem?
2. The latent space approach seems interesting. Are there any other benefits of using it, such as adapting to other COP problems more easily?
3. In the paper, the authors also discuss more complex objectives; however, for JSP, using the current lower bound makespan of a partial solution seems to work quite well for RL. Could the authors give a detailed example of an objective that RL could not handle but self-labeling could?

**Ethical Concerns:**

["NO or VERY MINOR ethics concerns only"]

**Final Justification:**

The authors addressed all the issues raised in my previous comments and conducted the requested additional experiments. Accordingly, I have raised my score from 3 to 4.

**Limitations:**

Most limitations are discussed; however, I would have liked a discussion about the enormous training requirements of sometimes more than a week, and the different hyperparameters used for each problem

**Quality:**

3

**Strengths And Weaknesses:**

Strengths:
- The paper is well written and accessible to understand.
- The introduced concepts, such as using the latent space for multiple policies and the weighted loss function, are, to my understanding, novel for a self-labeling approach.
- The introduced method, MP-ASIL, is well tested on multiple scheduling and routing problems and outperforms the baselines it compares to.

Weaknesses:
A major weakness of the paper is the lack of mention and comparison to other self-labeling approaches, except for SLIM. Since SLIM, other approaches were proposed, such as (Pirnay & Grimm, 2024), which also tackle the issue of solution diversity. Moreover, (Pirnay & Grimm, 2024) achieves significantly better results on JSP compared to MP-ASIL. For TSP and CVRP, it seems that MP-ASIL achieves better results, but this was hard to compare due to the limited experimentation of the authors. The authors also experimented on quite a lot of scheduling problems; however, this limits the diversity of their experiments. In the paper, the authors mention broad applicability to other CO problems, but mainly test on scheduling problems and only include limited experiments for routing problems. I would rather have seen a more diverse set of CO problems being tested, instead of five scheduling problems.
The authors did not include essential literature in their work, which tackles similar issues and seems to outperform them significantly. I also find the experiments lack diversity. I strongly believe the paper would significantly improve with more diverse experiments on other CO problems. Lastly, the authors also claim their approach is more sample-efficient, but I cannot find any results that support this claim; rather, I see that their training time for JSP is almost identical to SLIM, with both taking around 4 days, which disproves this. Below, I have listed more specific issues I found with the paper, and which I hope can help the authors improve their paper:

- On lines 36-38, it is stated that RL struggles with effective exploration due to the use of single policies. However, the authors’ claim is only supported by limited work. I am not convinced that this is really such a big issue for RL, given that we can include an entropy bonus. I would either remove this statement or add more relevant literature that explores this problem.
- A similar issue is found for the reward shaping claim on lines 44-48, whereby the claim is only supported by severely outdated references from before the year 2000.
- On lines 86-88, I read that supervised learning is used for CO problems, whereby they train on small-sized instances. However, the authors also state that generating these instances is challenging. I am not convinced by this statement, because for small-scale instances it can be trivial to generate optimal solutions, depending on the problem and the scale.
- In Eq. 5, it seems that you take the absolute value. Would this also not reward worse examples? I am not sure if this is correct, but this seems to require more explaination.


Pirnay, J., & Grimm, D. G. (2024). Self-Improvement for Neural Combinatorial Optimization: Sample Without Replacement, but Improvement. Transactions on Machine Learning Research.

---

> ### Author Rebuttal · Authors · 2025-07-30
>
> We thank the reviewer for their comments and positive feedback! Please find our responses to the main concerns and questions below:
>
> >**W1**: Other self-labeling approaches
>
> We sincerely apologize for the lack of mention and comparison to other self-labeling approaches, such as Pirnay&Grimm 2024 (SI GD) [1]. We will revise the manuscript to discuss and include this comparison. However, we respectfully disagree that SI GD achieves significantly better results on job shop scheduling problems compared to MP-ASIL. SI GD employs a deep transformer network, whereas we use a simpler neural network architecture [2], and this additional model complexity may contribute their improved solution quality, limiting direct comparison. Nevertheless, on the Taillard  benchmark, MP ASIL reduced the average performance gap from 8.1% to 7.4% compared to SI GD (greedy), while delivering twice the inference speed. Similarly, against SI GD (beam 16), MP ASIL closed the gap from 7.8% to 7.4% with roughly a 10× faster runtime. SI GD with more sophisticated search surpass our performance, it incurs over 59× the computational cost.
>
> >**W2**: Diversity of experiments
>
> We acknowledge that our experiments are mainly focused on scheduling and routing problems. However, our main contribution centers on job scheduling problems (JSPs) and we show superior performance on five JSP variants. We also demonstrated broader applicability by evaluating the TSP and ten CVRP variants, which are standard benchmarks in Neural Combinatorial Optimization (NCO) literature. Compared to previous studies, we believe that we do not cover fewer CO problems at all.
>
> Despite our extensive experimental evaluation, we fully appreciate the reviewer’s concern regarding the applicability of our method to other CO problems. The strength of our method lies in its simplicity and universality. As described in Appendix B, given any existing architecture for CO problems, one can easily apply latent conditioning and MP-ASIL to improve performance. The model-agnostic nature of MP-ASIL makes it applicable to a wide range of CO problems. For future work, we plan to apply our approach to other CO problems such as the Knapsack problem to further validate its universal applicability.
>
> >**W3**: Sample efficiency
>
> SLIM faces the challenge of low sample efficiency, as all sampled solutions except the best one are discarded during training. However, MP-ASIL leverages information from all sampled solutions through adaptive imitation intensity control, significantly improving sample efficiency. Additionally, MP-ASIL achieves comparable—or even superior—validation performance to SLIM using only 1.2× to 10× fewer training epochs. (due to the NeurIPS 2025 policy, we cannot upload images during the rebuttal phase. Therefore, we will incorporate in our revised manuscript to provide validation performance of SLIM). These empirical results demonstrate that our approach is more sample-efficient.
>
> >**W4**: On lines 36-38
>
> Thank you for your informative comment. Computing the entropy over the entire trajectory remains computationally intractable, which limits the practical applicability of entropy bonus methods. Also, there are many literature that have pointed out that single policy approaches struggle with sufficient exploration [3,4,5]. We will add relevant literature as references on line 39.
>
> >**W5**: On lines 44-48
>
> Thank you for your valuable feedback. We will add [2] as reference on line 46, where they stated "RL approaches require the careful definition of a meaningful reward function, which is often a complex and challenging task".
>
> >**W6**: On lines 86-88
>
> We sincerely apologize for any confusion. For clarity, we believe it would be better to remove that content.
>
> [**Removed**] Line 39: However, as mentioned earlier, obtaining expert annotations for imitation is challenging.
>
> >**W7**: Eq. 5
>
> We assume that we solve a minimization problem. So $f(\boldsymbol{\tau^*}, s)-mean(s) \leq 0$ always holds. To use this difference as a non‐negative weight, we take its absolute value. However, Equation 5 can be rewritten in alternative form as follows:
>
> - $$\mathcal{L}_{ASIL}= \left( \frac{f(\boldsymbol{\tau}^*, s) - \textit{mean}(s)}{\textit{std}(s)} \right)\frac{1}{\mathcal{|O|}}\sum\_{t=1}^{\mathcal{|O|}}\log\pi\_{\theta}\left(\tau^\*\_t | s, \boldsymbol{\tau}^\*\_{\<t}, z^\* \right)$$
>
> - $$\mathcal{L}_{ASIL}= -\left( \frac{\textit{mean}(s)-f(\boldsymbol{\tau}^*, s)}{\textit{std}(s)} \right)\frac{1}{\mathcal{|O|}}\sum\_{t=1}^{\mathcal{|O|}}\log\pi\_{\theta}\left(\tau^\*\_t | s, \boldsymbol{\tau}^\*\_{\<t}, z^\* \right)$$
>
> If you think either of the two equations is clearer than Eq. 5, we would appreciate your recommendation.
>
> > **Q1**: Flexible job shop
>
> Thank you for your valuable suggestion. We conduct additional experiments by applying MP-ASIL to the DANIEL [6], a state of the art Flexible Job Shop Scheduling Problem (FJSSP) architecture and report the results in Table 1 (below). From the table, we can observe that MP-ASIL significantly outperforms all neural solvers, including DANIEL. Furthermore, on large scale instances, MP-ASIL even surpasses OR-Tools. We believe these findings further demonstrate the universal applicability and robustness of MP-ASIL across diverse CO domains.
>
> Table 1. Experiment results on FJSSP
>
> | Method | Type | 10×5 |  | 20×5 |  | 15×10 |  | 20×10 |  | 30×5 |  | 40×10 |  |
> |--------|------|------|------|------|------|-------|------|-------|------|------|------|-------|------|
> |        |      | Gap ↓ | Time ↓ | Gap ↓ | Time ↓ | Gap ↓ | Time ↓ | Gap ↓ | Time ↓ | Gap ↓ | Time ↓ | Gap ↓ | Time ↓ |
> | OR-Tools | Exact | 0.00% | (50h) | 0.00% | (50h) | 0.00% | (50h) | 0.00% | (50h) | 0.00% | (50h) | 0.00% | (50h) |
> | HGNN (S=100) | NCH | 9.66% | (1.9m) | 10.31% | (3.9m) | 12.13% | (6.6m) | 9.64% | (10.7m) | 12.36% | (21.3m) | 12.26% | (40.9m) |
> | MCGN (S=100) | NCH | 9.01% | - | 8.36% | - | 11.77% | - | 7.70% | - | 12.44% | - | 12.50% | - |
> | RS (S=100) | NCH | 7.26% | - | 7.22% | - | 9.59% | - | 6.06% | - | 11.14% | - | 11.29% | - |
> | DANIEL (S=100) | NCH | 3.57% | (1.2m) | 2.46% | (3.1m) | 6.79% | (6.5m) | 1.59% | (10.2m) | 4.43% | (20.6m) | 3.77% | (37.6m) |
> | MP-ASIL (k=100) | NCH | **3.00%** | (1.2m) | **0.67%** | (3.1m) | **4.61%** | (6.5m) | **-3.00%** | (10.3m) | **-0.15%** | (20.7m) | **-0.59%** | (37.8m) |
>
> >**Q2**: Latent space approach
>
> We appreciate your interest in our approach. Latent conditioned policies have the advantage of easily representing multiple policies with minimal model parameter increase. Specifically, as mentioned in Appendix B, when adapting to different CO problems, latent conditioned policies can be easily implemented by concatenating latent variables to the input (hidden embeddings) of the decision-making layer, regardless of problem-specific architectures. Additionally, as described in Appendix E.1, latent conditioned policies add a fixed number of model parameters, regardless of the number of policies. This property enables flexible, efficient, easy, and scalable multiple policy representation.
>
> > **Q3**: More complex objectives
>
> For JSSP with makespan objective, the difference between the makespan lowerboud before and after making a decision as a reward is widely used. However, this reward function only considers the precedence relationships of operations to calculate the lower bound, leading to estimation error accumulation, which hinders accurate guidance for the scheduling process. Moreover, for problems without precedence relationships, it is impossible to estimate tight lower bound using this approach.
> When considering total total tardiness objectives, another popular performance measure, the influence of an early scheduling decision is delayed. Intuitively, one might set the immediate reward to be the negative tardiness of the job scheduled. However, even if that job has zero tardiness it is scheduled, we cannot account for how this choice will affect the tardiness of all subsequent jobs—yet the reward remains zero. This zero‐reward signal provides no differentiation between actions that will later incur large penalties and those that will not, making accurate credit assignment impossible.
>
> > **L**: Training requirements and the different hyperparameters
>
> For each problem, we use the same training hyperparameters from the original papers. Our experimental results demonstrate that simply applying MP-ASIL without hyperparameter tuning achieves significantly improved performance with approximately the same training time, which we respectfully believe is a strength of our approach. This demonstrates the practical value and ease of adoption of our method - practitioners can integrate MP-ASIL into existing frameworks without extensive hyperparameter optimization or architectural modifications.
>
> [1] Pirnay, Jonathan, and Dominik G. Grimm. "Self-improvement for neural combinatorial optimization: Sample without replacement, but improvement." arXiv preprint arXiv:2403.15180 (2024).
>
> [2] Corsini, Andrea, et al. "Self-labeling the job shop scheduling problem." Advances in Neural Information Processing Systems 37 (2024): 105528-105551.
>
> [3] Grinsztajn, Nathan, et al. "Winner takes it all: Training performant rl populations for combinatorial optimization." Advances in Neural Information Processing Systems 36 (2023): 48485-48509.
>
> [4] Chalumeau, Felix, et al. "Combinatorial optimization with policy adaptation using latent space search." Advances in Neural Information Processing Systems 36 (2023): 7947-7959.
>
> [5] Hottung, André, Mridul Mahajan, and Kevin Tierney. "PolyNet: Learning diverse solution strategies for neural combinatorial optimization." arXiv preprint arXiv:2402.14048 (2024).
>
> [6] Wang, Runqing, et al. "Flexible job shop scheduling via dual attention network-based reinforcement learning." IEEE Transactions on Neural Networks and Learning Systems 35.3 (2023): 3091-3102.

---

> ### Comment · Reviewer_9EdW · 2025-08-03
>
> I acknowledge that the authors addressed many of my concerns. Nevertheless, I am still unsure about a few points in their rebuttal, which I list below.
>
> ---
> ### W1: Other self-labeling approaches
> Based on your rebuttal, I agree that the method of Pirnay et al. [1] might be difficult to compare with yours, as you stated. However, I am still concerned about the novelty of your work when your justification hinges on the fact that Pirnay et al. use a different network architecture and evaluation strategy. Both you and Pirnay et al. aim to solve a similar problem, which requires a rigorous comparison to isolate which method performs better by eliminating other factors, such as evaluation strategy and network architecture.
>
> For example, would MP-ASIL outperform Pirnay et al. if the same network architecture and evaluation strategy were used? It might be the case that Pirnay et al. would perform better with your network architecture.
>
> In short, your argumentation is not yet sufficient to fully convince me on this topic.
>
> ---
> ### W5: On lines 44‚Äì48 and Q3: More complex objectives
> Corsine et al. [2] make this claim again without justifying it. However, I appreciate your answer to Q3, as it explains why designing a reward function might not always be feasible, and could serve as a better justification than simply citing other work.
>
> To add to this, would it be possible to support multiple objectives with MP-ASIL, rather than a singular objective?
>
> ---
> ### W7: Eq. 5
>
> I believe version 1 is a significant improvement; however, both versions will not yield the same result. For example, if $\text{mean}(s) = 5$ and if $f(\tau*, s)$ is either 7 or 3, the old version would produce 2 in both cases, whereas the new version would produce 2 and -2, respectively. Moreover, in your old version, both $f(\tau*, s) = 7$ and $f(\tau*, s) = 3$ would yield the same imitation control. Could you maybe explain a bit further what is happening here?
>
> ---
> ### Q1: FJSP
>
> Thank you for your additional experiments. Your results show that MP-ASIL outperforms DANIEL [3]. Could you elaborate a bit more on how you adapted MP-ASIL to FJSP? I also noticed that Wang et al. [3] reported significantly different results from yours in Table 1. Could you explain this discrepancy, since your reported runtime is significantly higher, and the reported Gap is also different than Wang et al? Could you also evaluate on the Brandimarte and Hurink benchmark sets?
>
> ---
> ### References
>
> [1] Pirnay, Jonathan, and Dominik G. Grimm. "Self-improvement for neural combinatorial optimization: Sample without replacement, but improvement." arXiv preprint arXiv:2403.15180 (2024)
>
> [2] Corsini, Andrea, et al. "Self-labeling the job shop scheduling problem." Advances in Neural Information Processing Systems 37 (2024): 105528-105551.
>
> [3] Wang, Runqing, et al. "Flexible job shop scheduling via dual attention network-based reinforcement learning." IEEE Transactions on Neural Networks and Learning Systems 35.3 (2023): 3091-3102.

---

> ### Author Response · Authors · 2025-08-03
>
> Thank you for your thoughtful comment. Please find our response to your new question below.
>
> > **W1**:
>
> Our main contribution is developing an efficient and effective Multi-Policy (MP) optimization method to address limitations in RL-based MP optimization approaches. To this end, we introduced MP-ASIL, the first SIL-based MP optimization method.
> In this context, our research differs from existing SIL methods that focus on single-policy optimization. Nevertheless, when compared to SI GD:
> - SI GD performs sample-without-replacement using a stochastic beam search-based method from a single policy to sample diverse solutions. This process requires an extensive search for each instance and also demands significant hyperparameter tuning. In contrast, our framework allows multiple behaviors to freely explore the solution space. Therefore, no additional search process is required, and it can be very easily implemented.
> - Through our novel mechanism, we can obtain not only diverse but also expert policies specialized for instance sub-distributions. This is a powerful advantage that can be achieved because our objective is to optimize MP.
> - We introduce a novel mechanism that dynamically adjusts imitation intensity by considering the relative quality of pseudo-labels. This addresses an aspect that has been overlooked in existing SIL methods.
>
> Nevertheless, as you correctly pointed out, we acknowledge that experimentation is necessary to analyze which method is more effective when MP-ASIL is applied to the same backbone as SI GD. To this end, we are currently running experiments applying MP-ASIL to the BQ backbone that SI GD used for TSP. We will include these results in the final version of the paper.
> >**W5.1**:
>
> Thank you for this valuable feedback. You're right. We should provide more concrete justification rather than simply citing other work. We will ensure to add this content to the main paper instead of merely providing citations.
>
> >**W5.2**:
>
> Yes, MP-ASIL can be applied to Multi-Objective Scheduling Problems (MOSP). We can achieve this through two approaches:
> - **Weighted Sum Approach**:
> The simplest approach for solving MOSP is the weighted sum approach ($f=λ_1 f_1 + ... $ $+λ_n f_n$). Therefore, since this problem is identical to a single-objective problem, the MP-ASIL introduced in the main text can be applied without any modifications.
> - **Pareto-based Approach**:
> Another way to solve MOSP is decomposition-based methods. In this paradigm, for each weight vector $λ$, MP-ASIL can generate candidate solutions through multiple latent-conditioned policies $π(·|s, λ, z)$. Therefore, MP-ASIL approach can be applied with any algorithm modification.
>
> >**W7**:
>
> We assume that we solve a **minimization problem**. So, if $\textit{mean}(s) = 5$ , then $f(\boldsymbol{\tau}^\*,s)$ can never be 7, because $f(\boldsymbol{\tau}^\*,s)$ is the best objective value (minimum value). Also, since we only update the model for the best-performing policy, there is no case where we update on trajectories that are worse than the average.
> Additionally, Version 1 and Version 2 are essentially the same equation. The only difference is whether the negative sign is outside the parentheses or not. Therefore, the old version and Versions 1 and 2 all yield the same loss values.
>
> >**Q1**:
>
> - We implemented MP by concatenating latent variables to the input of DANIEL's decision-making layer (MLP), and applied the same MP-ASIL introduced in the main text for model training. Training hyperparameters followed the original paper, and we conducted training on 10×5 size instances.
> - We sincerely apologize. There were some typos in our table. Table 1's values are identical to the results from the original paper. Additionally, regarding computation time, while the DANIEL paper reports the time taken to solve a single instance, our table reports the time to solve all test instances, following the definition used in our main paper.
>
> **Table 1.**
> | Method | Type | 10×5 |  |  20×5 |  | 15×10 |  | 20×10 |  | 30×10 |  | 40×10 |  |
> |--------|------|------|------|-------|------|-------|------|-------|------|------|------|-------|------|
> |        |      | Gap↓ | Time↓ | Gap↓ | Time↓ | Gap↓ | Time↓ | Gap↓ | Time↓ | Gap↓ | Time↓ | Gap↓ | Time↓ |
> | DANIEL (S=100) | NCH | 5.57% | (1.2m) | 2.46% | (3.1m) | 6.79% | (6.5m) | -1.03% | (10.2m) | 4.43% | (20.6m) | 3.77% | (37.6m) |
> | MP-ASIL (k=100) | NCH | 3.00% | (1.2m) | 0.67% | (3.1m) | 4.61% | (6.5m) | -3.00% | (10.3m) | -0.15% | (20.7m) | -0.59% | (37.8m) |
> - We evaluate MP-ASIL on the Brandimarte and Hurink benchmark sets. From the Table 2, we can observe that MP-ASIL consistently outperforms DANIEL.
>
> **Table 2.**
> | Method | Brandimarte | Hurink_e | Hurink_v | Hurink_r |
> |--------|-------------|----------|----------|----------|
> | DANIEL (S=100) | 9.53% | 9.08% | 0.69% | 4.95% |
> | MP-ASIL (k=100) | 7.32% | 7.24% | 0.50% | 4.74% |

---

> > ### Comment · Reviewer_9EdW · 2025-08-05
> >
> > Thank you for addressing all the issues raised in my previous comments and for conducting the requested additional experiments. I am also pleased that you plan to include further experiments in the camera-ready version, in addition to the new FJSP results. Accordingly, I have raised my score from 3 to 4.

---

> > > ### Author Response · Authors · 2025-08-06
> > >
> > > Thank you again for your time and effort in reviewing our work.

---

### Official Review · Reviewer_NWcr · 2025-07-02

**Clarity:** 3
**Significance:** 3
**Originality:** 2
**Rating:** 4
**Confidence:** 4

**Summary:**

The paper proposes MP-ASIL (Multi-Policy with Adaptive Self-Imitation Learning), a novel framework that addresses key challenges in combinatorial optimization. Instead of using a single policy, MP-ASIL employs multiple policies sharing the same objective but developing diverse solution strategies. The framework introduces Adaptive Self-Imitation Learning to guide these policies using self-generated teaching signals efficiently. MP-ASIL overcomes previous reinforcement learning limitations by: (1) using multiple specialized policies to better represent multimodal distributions, reducing mode collapse and enhancing exploration; (2) autonomously generating training labels without requiring MDP formulations; and (3) adaptively controlling imitation intensity based on label quality. The model-agnostic approach integrates easily with existing architectures. Extensive experiments across five job scheduling problems demonstrate that MP-ASIL enhances exploration capabilities and overall performance, achieving state-of-the-art results on various datasets and outperforming leading neural solvers on other combinatorial optimization problems.

**Questions:**

- Page 3, line 135, Why each decision is an index from 1 to the number of total operations?
- How Eq. (5) compares to the loss function used in GRPO [1]?
- How is the exploration-exploitation controlled during training?

[1] DeepSeekMath: Pushing the Limits of Mathematical Reasoning in Open Language Models

**Ethical Concerns:**

["NO or VERY MINOR ethics concerns only"]

**Final Justification:**

The rebuttal addresses my main concern about inference, and I agree that the latent variable can capture various solutions across different spaces. During inference, search algorithms are typically used to find a suitable latent variable for a given instance. However, I remain concerned about the computational cost during both training and inference. Particularly challenging is developing efficient search algorithms for instances with stochasticity, despite the authors' promising results. Therefore, I believe a score of 4 is fair.

**Limitations:**

No. The authors should introduce more ablation studies to evaluate the importance of each component and hyper-parameter, such as sampling size and sampling strategies of $z$.

**Quality:**

3

**Strengths And Weaknesses:**

**Strengthen**

- Introduces latent variables to improve solution diversity, potentially reducing mode collapse and enhancing exploration efficiency.
- Proposes a self-imitation learning framework that relies on autonomously generated training labels.
- Extensive experiments and ablation studies demonstrate the effectiveness of the approach.

**Weakness**

- Theoretically, in Algorithm 1, a set of latent variables is randomly sampled for each instance at each step. If an instance is sampled multiple times, it will encounter a different set of latent variables each time. During extensive training—where each instance is sampled and trained multiple times—the model will likely disregard the contribution of these latent variables since they essentially function as random noise. Conversely, if each instance is only trained for few steps, the latent variables would become significant and might cause decision-making to approach a random policy. It would be helpful if the authors could elaborate on this issue.
- Experimentally, some key ablation studies are missed in my opinion, e.g., 1. It would be interesting to see individual contributions of training with latent variables and sampling, namely, MP-ASIL with few sampling v.s. random policy with the same number of sampling adopted in the experiment. 2. How its performance changes as the sampling times change from a small number to a larger one.  3. How different generation policies of $z$ affect the performance?

---

> ### Author Rebuttal · Authors · 2025-07-30
>
> We thank the reviewer for their comments and positive feedback! Please find our responses to the main concerns and questions below:
> > **W1.1**: Random noise
>
> Thank you for your valuable comment. We sincerely apologize for any confusion regarding our method. We clarify why latent variables are not acting as random noise.
> In our training procedure, sampling different latent variables for the same instance across epochs is not a weakness but a key design choice. Because, MP-ASIL does not seek to find a single “optimal” latent variable for each instance. Instead, we aim to learn a multimodal latent space for diverse policies, since multiple regions of the latent space can generate high-quality solutions for the same instance. This is crucial because Combinatorial Optimization (CO) problems often have multiple distinct solution strategies that yield the same objective value. Therefore, sampling new latent variables each time is an essential algorithmic component for learning a general multimodal policy latent space.
> In this learning process, policy specialization is achieved through MP-ASIL's distinctive model update mechanism. At each training step,
> - We sample $k$ latent variables and generate $k$ solutions for instance $s$
> - Only the best-performing policy $\pi(·|s, z^*)$ receives gradient updates
> - This mechanism establishes a self-reinforcing dynamic in which successful pairings are strengthened.
>
> This joint learning process ensures that latent variables become specialized experts for specific instance sub-distributions—the latent variables evolve from random perturbations into specialized decision-making factors. This conditioning mechanism has been empirically validated in recent work [1] to produce meaningful, structured policy latent spaces rather than noise. Additionally, our ablation studies (Table 4, MP ablation) demonstrate that the latent variables contribute meaningfully to performance improvements. Furthermore, beyond performance , when we tracked the loss during the training process of the version with latent variables removed, the multi-policy approach showed lower loss values.
> In SIL, the loss can be interpreted as the model’s confidence in its best trajectories. If the latent variables were functioning as random noise, such results would be impossible to achieve.
>
> >**W1.2**: Conversely, if each instance is only trained for few steps…
>
> As mentioned in our response for W1.1, MP-ASIL is designed to learn a general policy latent space. That is, even if each instance is only trained for a few steps, we are not learning a one-to-one mapping, so we can share general properties from the learning of instance distributions. Therefore, it is not problematic that specific instances are only trained for a few steps. What matters is that the learned latent space encodes diverse, high-quality strategies that generalize across the problem distribution. This capability stems from deep learning's inherent strength in capturing general mappings, much like how trained neural solvers can still perform well on test instances that were not experienced during the training process. While latent variables may act like random noise in the early stages of training, we assume the standard deep learning setting where the model is fully trained until convergence. Hence, the trained model is not a random policy.
>
> >**W2.1**: Ablation study 1
>
> Could you please elaborate on what you mean by "random policy with the same number of sampling adopted in the experiment"? Based on our understanding, you are suggesting that we conduct the same model training process, but generate candidate solutions for training using stochastic sampling from a separate random policy rather than from the MP-ASIL-trained model.
> Due to the limited rebuttal time, we apply this approach only to PFSP and confirmed that no performance improvement was achieved compared to MP-ASIL validation performance. If our understanding of your suggestion is incorrect, we would appreciate further clarification!
>
> >**W2.2**: Ablation study 2
>
> We analyze the effect of the number of policies ($k$) on model performance during training. We train our model under varying choices of $k$, with $k$ ∈ {32, 64, 128} for PFSP and $k$ ∈ {64, 128, 256} for JSSP. Tables 1 and 2 (below) show the results of our analysis on the TA benchmark. From the tables, we can observe that training with larger $k$ generates stronger models. This result aligns with our expectation that a larger $k$ produces a greater number of specialized behavior policies, enabling more extensive exploration of the solution space and improving the chance of finding better solutions, albeit at the cost of increased memory usage.
>
> Table 1. The effect of $k$ on PFSP.
>
> |     | PFSP 20x5 | PFSP 50x5 | PFSP 100x5 |
> |-----|-----------|-----------|------------|
> | k=32 | 0.64\% | 0.31\% | 0.20\% |
> | k=64 | 0.44\% | 0.25\% | 0.19\% |
> | k=128 | 0.37\% | 0.22\% | 0.16\% |
>
> Table 2. The effect of $k$ on JSSP.
>
> |     | JSSP 15x15 | JSSP 20x15 | JSSP 20x20 |
> |-----|------------|------------|------------|
> | k=64 | 7.25\% | 9.01\% | 8.96\% |
> | k=128 | 6.89\% | 8.95\% | 8.73\% |
> | k=256 | 6.76\% | 8.50\% | 8.65\% |
>
> >**W2.3**: Ablation study 3
>
> In Appendix E.2, we evaluated models trained under three different latent distributions. In summary, although we used the distribution that performed best on the TA benchmark, all distributions demonstrated superior performance compared to the single policy approach. This finding provides evidence for the robustness and generalizability of our approach. Our method's effectiveness is not dependent on fine-tuning the latent space configuration, but rather stems from the fundamental multi-policy optimization mechanism. Please let us know if you would like any other clarification. We would be happy to provide them.
>
> >**Q1**: Page 3, line 135
>
> Thank you for your valuable comment. A solution of scheduling problem can be autoregressively constructed by sequentially assigning each operation to a compatible machine and appending it to the end of that machine's operation sequence. Since we build a solution by selecting one operation per decision step, the total number of decisions equals the number of operations. In single machine scheduling problems, each job consists of exactly one operation, and our task is to determine the sequence of these operations on the machine. Since we build this sequence by selecting one operation per decision step, the total number of decisions equals the number of operations. However, in parallel machine scheduling, our task includes assigning each operation to a machine. So, the total number of decisions equals the number of operations, but, as you correctly pointed out, the expression in line 135 is imprecise because the range from 1 to $ \mathcal{|O|}$ does not include any machine assignments. We will remove this expression in the revised manuscript to improve clarity. Thank you for your valuable feedback! This revision will help eliminate potential confusion for readers and make our problem formulation more precise.
>
> [**Removed**] Line135: where $\tau_t \in \{1, \dots, \mathcal{|O|}\}$ and $\tau_t \neq \tau_{t'}, \forall t \neq t'$.
>
> >**Q2**: Compared to GRPO loss function
>
> Thank you for your insightful question. Both GRPO and MP-ASIL compute normalized advantage values similarly, using the average reward over multiple sampled outputs for each instance as the baseline. However, their loss functions differ in the following key aspects:
> - GRPO is an RL-based method (PPO variant), while our method is Self-Imitation Learning (SIL)-based method. Therefore, no reward function is required.
> - GRPO's loss function focuses on single policy optimization, while MP-ASIL focuses on multi-policy optimization.
> - GRPO updates the model for all trajectories generated for the same instance. However, MP-ASIL only update the model for the best trajectory, encouraging each policy to learn specialized problem-solving strategies.
> - While both methods calculate normalized advantage values similarly, GRPO's method represents a novel approach to computing advantage values for standard PPO loss functions, whereas MP-ASIL's advantage value is introduced to dynamically control imitation intensity for the best trajectory—a novel training mechanism not utilized in existing self-imitation learning for NCO.
>
> >**Q3**: Exploration-exploitation control
>
> Thank you for your informative question. In SIL, we can characterize exploration and exploitation as follows:
> - Exploration (Sampling phase): generating diverse candidate solutions
> - Exploitation (Imitation phase): Imitating the best solution found during exploration
>
> MP-ASIL extends this SIL framework with multiple policies and adaptive mechanism.
>
> - Exploration phase: $k$ diverse policies with different latent variables explore different regions of the solution space. This multi-policy approach significantly enhances exploration beyond standard single policy approaches.
> - Exploitation phase: self-imitation on the best trajectory found during exploration, with adaptive imitation intensity control that prevents over-exploitation of marginally better solutions.
>
> In this context, exploration is controlled by the number of policies sampled during the learning process, which involves a trade-off with training time. Additionally, exploitation control is achieved through dynamic control of imitation intensity based on the quality of pseudo-labels. While we don't employ explicit ε-greedy mechanisms or entropy bonuses, the MP-ASIL framework inherently provides an exploration-exploitation balance that is well-suited for self-imitation learning.
>
> [1] Chalumeau, Felix, et al. "Combinatorial optimization with policy adaptation using latent space search." Advances in Neural Information Processing Systems 36 (2023): 7947-7959.

---

> > ### Comment · Reviewer_NWcr · 2025-08-05
> >
> > Thank you to the authors for the rebuttal, which resolves most of my concerns. However, I'm still confused about the point: **This joint learning process ensures that latent variables become specialized experts for specific instance sub-distributions—the latent variables evolve from random perturbations into specialized decision-making factors.** If I understand correctly, I assume that during inference, the latent variables are randomly sampled. If these latent variables become specialized decision-making factors, how can we ensure that the randomly sampled ones are those we desire? (Of course, we can increase the number of samples, but I would like to understand the mechanism more deeply.)

---

> ### Author Response · Authors · 2025-08-05
>
> We sincerely thank you for your time and effort in reviewing our work. Please find our response to your new question below.
> > How can we ensure that the randomly sampled ones are those we desire?
>
> Our main contribution lies in how to effectively train Multiple Policies (MP) represented by latent variables, and how to efficiently utilize latent variables during the inference was outside the scope of our research. Therefore, as you correctly understand, randomly sampling latent variables at inference time does not guarantee that promising latent variables will be included (although increasing the number of samples can improve the likelihood). **However, at inference time, we can apply a principled search procedure to find the most performant regions.** Recently, COMPASS [1] introduced a methodology that utilizes the Covariance Matrix Adaptation (CMA) [2] evolutionary algorithm to search the policy latent space. To verify whether latent conditioned policies trained with MP-ASIL combine well with CMA, we performed latent space search using CMA at inference time, as mentioned in Appendix E.7. In this experiment, we employ three independent CMA components in parallel with 1,600 attempts, following the original paper settings, and report the results in Table 1 below (MP-ASIL + CMA). For comparison, we report results Poppy + CMA obtained by combining CMA with latent conditioned policies trained using Poppy [3], a state-of-the-art RL-based MP optimization method, instead of MP-ASIL. This baseline exactly matches the original COMPASS training setup. We also include results from performing the same number of samplings as the search attempts (MP-ASIL + Sampling). From the table, we can see that MP-ASIL + Sampling already outperforms Poppy + CMA. Furthermore, MP-ASIL + CMA achieves significant relative performance improvements of 25.1% on Permutation Flow Shop scheduling Problem and 9.6% on Job Shop Scheduling Problem compared to MP-ASIL + Sampling. These results demonstrate the effectiveness of MP-ASIL in optimizing MP compared to the RL-based method and show that performance can be significantly improved through procedures that search promising latent regions. Beyond CMA at inference time, future work could develop more efficient methods by learning instance-conditioned priors $p_{\theta}\left(z|s\right)$ that predict which latent variables are likely to perform well for a given instance. Such priors could enable targeted sampling or Bayesian optimization approaches, effectively bridging the gap between random sampling and full search by leveraging the relationship between instance features and successful latent variables learned during training.
>
> **Table1**. Performance evaluation results combined with CMA on the TA benchmark
>
> | | PFSP 20x10 | JSSP 20x15 |
> |---|---|---|
> | | Gap ↓ | Gap ↓ |
> | Poppy + CMA | 2.27% | 7.35% |
> | MP-ASIL + Sampling | 2.04% | 6.98% |
> | MP-ASIL + CMA | **1.63%** | **6.37%** |
>
> ---
> [1] Chalumeau, Felix, et al. "Combinatorial optimization with policy adaptation using latent space search." Advances in Neural Information Processing Systems 36 (2023): 7947-7959.
>
> [2] Hansen, Nikolaus, and Andreas Ostermeier. "Completely derandomized self-adaptation in evolution strategies." Evolutionary computation 9.2 (2001): 159-195.
>
> [3] Grinsztajn, Nathan, et al. "Winner takes it all: Training performant rl populations for combinatorial optimization." Advances in Neural Information Processing Systems 36 (2023): 48485-48509.

---

> > ### Comment · Reviewer_NWcr · 2025-08-05
> >
> > Thank you to the authors for the clarification, which partially resolves my main concern about the inference algorithm. However, it raises another question: does the current algorithm only work for deterministic optimization problems? For instance, what if there is some stochasticity in the JSPs, such as variable durations of jobs? Can the algorithm still work in this scenario?

---

> ### Author Response · Authors · 2025-08-05
>
> > Stochastic optimization problems
>
> MP-ASIL can also be applied to stochastic scheduling problems. For stochastic problems, we can evaluate the solution quality through expected objective value or Value-at-Risk ($\textit{VaR}_{\alpha}$) values of scenarios by applying the model's decisions identically to multiple scenarios sampled from probability distributions. Therefore, in MP-ASIL, we can evaluate each policy's decisions through the scenario set without any modifications, allowing us to apply MP-ASIL in the same way as deterministic problems.
>
> To demonstrate this, we conducted experiments on the stochastic Flexible Job Shop Scheduling Problem (FJSSP), where durations are random variable, one of the most complex stochastic scheduling problems. We applied MP-ASIL to the Scenario Processing Module (SPM)-DAN [1], which proposes an attention-based SPM to solve stochastic FJSSP. This research is a representative and recent stochastic JSP paper in the NCO field. Specifically, we implemented latent conditioned policies by concatenating latent variables to the input of the decision-making layer (MLP) as mentioned in Appendix B. We applied the same MP-ASIL training procedure introduced in the main text for model training. Training hyperparameters and test setups followed the original paper, and we conducted training on 10×5 size instances.  For evaluation, we use the First-In-First-Out (FIFO), Most Operations Remaining (MOR), Shortest Processing Time (SPT), Most Work Remaining (MWKR), and SPM-DAN as baselines.
>
> Tables 1 and 2 report the Gap relative to SPM-DAN. From the Table 1, we can see that MP-ASIL shows clear performance improvement over SPM-DAN and significantly outperforms all dispatching rules. Also, for different objective function ($\textit{VaR}_{\alpha}$), as shown in the Table 2, MP-ASIL still demonstrates the best performance.
> Through these experimental results, we demonstrate that MP-ASIL can be applied to stochastic scheduling problems as well as deterministic problems. Particularly, in stochastic problems where computing the reward for each decision is very challenging, our framework has the advantage of not requiring separate reward functions. Moreover, in uncertain situations, our framework enables more robust learning by allowing more diverse scheduling experiences through specialized multiple policies.
>
> **Table 1**. Experiment results on stochastic FJSSP using the expected makespan objective.
> | | 10x5 | 20x5 | 15x10 | 20x10 |
> |---|---|---|---|---|
> | | | Gap | | |
> | FIFO | 13.00\% | 6.98\% | 14.46\% | 11.41\% |
> | MOPNR | 12.41\% | 8.43\% | 11.40\% | 10.37\% |
> | SPT | 22.43\% | 16.71\% | 23.32\% | 14.22\% |
> | MWKR | 10.66\% | 7.66\% | 8.82\% | 9.12\% |
> | SPM-DAN (S=100) | 0.00\% | 0.00\% | 0.00\% | 0.00\% |
> | **MP-ASIL** (k=100) | -2.35\% | -2.39\% | -3.28\% | -5.08\% |
>
>
> **Table 2**. Experiment results on stochastic FJSSP using the $\textit{VaR}_{\alpha}$ objective. We set $\alpha=95$%, following the original paper.
> | | 10x5 | 20x5 | 15x10 | 20x10 |
> |---|---|---|---|---|
> | | | Gap | | |
> | FIFO | 10.18\% | 4.98\% | 12.75\% | 9.16\% |
> | MOPNR | 9.52\% | 6.31\% | 9.63\% | 8.07\% |
> | SPT | 20.45\% | 14.73\% | 22.52\% | 21.64\% |
> | MWKR | 7.97\% | 5.71\% | 7.15\% | 7.03\% |
> | SPM-DAN (S=100) | 0.00\% | 0.00\%| 0.00\%| 0.00\%|
> | **MP-ASIL** (k=100) | -2.60\% | -2.24\% | -2.74\% | -4.38\% |
>
> ---
> [1] Smit, Igor G., et al. "Neural Combinatorial Optimization for Stochastic Flexible Job Shop Scheduling Problems." Proceedings of the AAAI Conference on Artificial Intelligence. Vol. 39. No. 25. 2025.

---

> > ### Comment · Reviewer_NWcr · 2025-08-06
> >
> > Thank the authors for their responses, which largely address my concerns. Based on this, I have decided to raise the score to 4.

---

> > > ### Author Response · Authors · 2025-08-06
> > >
> > > Thank you again for your time and effort in reviewing our work.

---

### Official Review · Reviewer_aRja · 2025-07-02

**Clarity:** 3
**Significance:** 3
**Originality:** 3
**Rating:** 4
**Confidence:** 3

**Summary:**

The paper studies the job scheduling problem, a classic optimization challenge in networking and machine learning, using reinforcement learning (RL). Unlike prior work that trains a single RL policy, this paper proposes an ensemble of RL policies that share the same optimization objective but capture diverse and complementary solution patterns. These policies are modeled by conditioning a single neural network on distinct latent variables, enabling the approach to explore a variety of solutions and select the best one for each problem instance. Experimental evaluation on different variants of the JSP, as well as the VRP, demonstrates the effectiveness of the proposed method.

**Questions:**

Please check the weaknesses section above.

**Ethical Concerns:**

["NO or VERY MINOR ethics concerns only"]

**Limitations:**

No discussion is provided in the paper. For some suggestions, please check the weaknesses section.

**Paper Formatting Concerns:**

NA.

**Quality:**

3

**Strengths And Weaknesses:**

Strengths:

- JSP is a classic optimization problem. While extensively studied, there remains significant room for improvement.

- The paper is well written and easy to follow.

- The idea of training multiple policies that encode diverse and complementary solution patterns using a single neural network is practically relevant.

- The performance evaluation is rigorous, and including results on the VRP setting is a nice added bonus.

Weaknesses:

- The JSP problem formulation overlooks communication aspects. It considers only processing time, ignoring communication latency. Additionally, each machine serves only one job at a time (no resource sharing), processing times are fixed and known a priori, and there is no dynamicity in the environment—no link congestion, failures, or server capacity fluctuations. These assumptions make the formulation overly abstract and less practical (for a more realistic setting, please see [1]). Could you elaborate on why these assumptions were made, and how they affect the practical applicability of your solution? Specifically, have you considered modeling communication costs? What are the challenges?

- Another concern is the use of a centralized solution for a problem that is inherently distributed (as servers/machines operate across different clusters). A distributed, multi-agent approach might be more appropriate, potentially addressing the same challenges—improving exploration, generalization, and scalability. (Again, [1] serves as an example of such a distributed solution). Could you elaborate why a centralized solution is used? Any thoughts on multi-agent alternatives?

[1]. https://ieeexplore.ieee.org/abstract/document/10621125

---

> ### Author Rebuttal · Authors · 2025-07-30
>
> We thank the reviewer for their comments and positive feedback! Please find our responses to the main concerns and questions below:
>
> >**W1**: More realistic constraints
>
> As you correctly pointed out, the problems addressed in this study often overlook realistic settings such as dynamicity and communication latency, which may limit their direct practical applicability in real-world scenarios. We acknowledge this limitation of our current work. However, our primary goal is to develop a foundational framework that can be universally applied across various Job Scheduling Problems (JSP). Therefore, we initially focused our research on standard static JSPs [1], which represent the most widely studied benchmark problems in the scheduling community and serve as fundamental building blocks for the field. In this work, we have developed a novel and generalizable learning algorithm and demonstrated its superior performance, significantly outperforming existing state-of-the-art methods across various benchmarks. We strongly believe that our work establishes a solid foundation that can serve as a stepping stone toward more complex, realistic scenarios. Our future work will indeed focus on extending our approach to JSPs under realistic assumptions, including machine breakdown, communication latency, stochastic processing times, and other practical constraints. We will update the limitation section to include this perspective. Thank you for your valuable comment!
>
> [**Added**] Line 930: Additionally, we acknowledge that in this work, we only consider standard and deterministic JSPs. We need to consider more realistic constraints and dynamicity in future work to improve real-world applicability.
>
> >**W2**: Centralized method
>
> We chose centralized architecture for the following reasons:
> As mentioned in our response to W1, we focus on standard and static JSPs. Therefore, we adopted the problem setting that is widely used in static scheduling research from an optimization perspective. Our goal is to retain the problem setting and solution framework of prior work while developing a stronger learning method. In this setting, we assume that the scheduler (agent) has full observability over the global state, including job/machine states, and system constraints. This allows a centralized agent to make globally informed decisions. Moreover, we do not account for communication latency; accordingly, we assume that the centralized agent can perceive changes and act instantaneously.
> Finally, because our setting does not involve partial observability or asynchronous machine updates, introducing multiple agents could complicate the learning process, as such systems often suffer from coordination and communication challenges.
> However, we fully agree with your observation that real-world environments are inherently distributed and that a centralized agent may be unable to make real‑time decisions due to communication latency. Consequently, our current problem setting may be challenging to maintain under such conditions. Extending our work to decentralized, multi‑agent systems is an important and promising future research direction that could significantly enhance the practical applicability of our approach. We will update the limitation section to include this perspective. Thank you for your valuable comment. We would be happy to collaborate if the opportunity arises.
>
> [**Added**] Line 930: Also, given that real-world environments are inherently distributed and a centralized agent may be unable to make real time decisions due to communication latency [2], extending our work to decentralized, multi-agent systems represent an important and promising future research direction.
>
> [1] Pinedo, Michael L. Scheduling. Vol. 29. New York: Springer, 2012.
>
> [2] Blöcher, Marcel, et al. "Train Once Apply Anywhere: Effective Scheduling for Network Function Chains Running on FUMES." IEEE INFOCOM 2024-IEEE Conference on Computer Communications. IEEE, 2024.

---

> ### Author Response · Authors · 2025-08-06
>
> Thank you again for your time and effort in reviewing our work.

---

### Official Review · Reviewer_SjhF · 2025-07-04

**Clarity:** 4
**Significance:** 3
**Originality:** 3
**Rating:** 4
**Confidence:** 4

**Summary:**

This article proposes MP-ASIL, a multi-policy adaptive self-imitation learning framework for solving combinatorial optimization problems, with a focus on the job-shop scheduling problem (JSP). The method constructs an ensemble of policy networks trained through self-imitation learning (SIL), where the imitation intensity is modulated based on a learned diversity score. This adaptive mechanism aims to promote exploration while ensuring performance improvement. The framework is evaluated on several JSP benchmarks and demonstrates state-of-the-art performance compared to both classical heuristics and learning-based baselines. The authors argue for the generality, robustness, and efficiency of MP-ASIL and emphasize its potential as a scalable and adaptive approach to combinatorial scheduling.

**Questions:**

The self-imitation process only reinforces the best trajectory found so far. Does this not introduce the risk of overfitting or premature convergence? Have the authors considered mechanisms to maintain diversity over time, such as memory buffers of high-quality trajectories, entropy regularization, or stochastic selection?

MP-ASIL’s inference time is compared against the total execution time of classical and hybrid solvers, but training time is not included. Furthermore, some runtime values are drawn from previous works using different hardware. How do the authors justify the fairness of this comparison? Would it be possible to provide normalized runtime metrics or at least clearly distinguish training and deployment costs?

How does MP-ASIL relate to the well-established literature on hyper-heuristics and automated algorithm design [1]? These approaches also aim to generalize across problem instances and domains, and may offer relevant baselines or complementary perspectives.

While the paper includes examples of MP-ASIL applied to TSP and VRP in the appendix, these applications are limited in scale and not thoroughly evaluated. Could the authors elaborate on how MP-ASIL generalizes across different combinatorial problem classes? Are modifications to the architecture, policy representation, or diversity mechanism required when extending to problems with different structures and constraints?

**Ethical Concerns:**

["NO or VERY MINOR ethics concerns only"]

**Final Justification:**

I have no further remarks or questions. While I will maintain the overall score based on the level of contribution, I have increased the clarity score to reflect the enhancements made during the rebuttal phase.

**Limitations:**

The authors provide a dedicated discussion of limitations in Appendix G. Key points include the reliance on handcrafted rewards, the lack of theoretical guarantees, and scalability concerns. However, the comparison with classical methods still omits training cost, and the generalization beyond JSP is only briefly explored through small-scale TSP and VRP examples.

**Paper Formatting Concerns:**

Main concern: The reference section is unusually long (77 entries); some consolidation may be possible to allocate space to more content in the main text.

Additional minor comments:
- Tables 2 and 3 are dense and use small font sizes, which hinders readability.
- Some terms (e.g., “diversity score”, “surprising trajectory”) are informally introduced and would benefit from clearer formalization.

**Quality:**

3

**Strengths And Weaknesses:**

Strengths:
- The paper addresses a well-established and practically important problem using a novel, learning-based formulation that builds on the strengths of policy ensembles and adaptive imitation.
- MP-ASIL introduces a principled mechanism for adjusting imitation intensity based on trajectory diversity, which helps mitigate common SIL limitations such as overfitting to early successes or reduced exploration.
- The method is clearly described and supported by comprehensive experiments on multiple JSP standard benchmarks, with strong empirical results in terms of solution quality and inference efficiency.
- The evaluation includes comparisons to exact methods (CPLEX, Gurobi, OR-Tools), classical heuristics (e.g., ACO, ILS, IGA) and recent learning-based methods, showing consistent improvements and even some new state-of-the-art results.
- The appendices provide useful implementation and training details, along with further ablation studies and case analyses supporting the method’s robustness and scalability.

Weaknesses:
- The paper does not consider other relevant literature like hyper-heuristics and automated algorithm design, which share the objective of instance-general learning and optimization. These paradigms could serve as additional baselines or conceptual anchors [1].
- The description of the proposed adaptive imitation intensity control approach includes informal terms such as “unexpected or surprising”. More formal terms are expected.
- The diversity metric used in the experiments is very simple. Why not relying on existing ones from the literature, such as in [2] which proposed a more robust population-level diversity measure.
- While MP-ASIL achieves impressive inference-time results, the comparison with other methods does not always consider total compute effort. Specifically, MP-ASIL’s inference time is compared to the full execution time of classical or hybrid solvers, omitting training time and possibly relying on disparate hardware setups from prior works. This limits the fairness and interpretability of the runtime comparison.
- The use of only the best trajectory for self-imitation introduces the risk of overfitting or premature convergence. It remains unclear whether the model retains any stochasticity or memory of alternative high-quality solutions to avoid narrow policy convergence.
- The comparison of the different policies performance on batches of performance is not always very significant and lacks a more precise and quantified analysis.
- Tables 2 and 3 present extensive results but suffer from readability issues due to dense formatting and small font sizes.

[1] Burke, E. K., Gendreau, M., Hyde, M., Kendall, G., Ochoa, G., Özcan, E., & Qu, R. (2013). A survey of hyper-heuristics. ACM Computing Surveys (CSUR), 45(3), 1–35. https://doi.org/10.1145/2480741.2480752
[2] Halim, A. H., Ismail, I., & Das, S. (2021). Performance assessment of the metaheuristic optimization algorithms: an exhaustive review. Artificial Intelligence Review, 54, 2323–2409. https://doi.org/10.1007/s10462-020-09906-6

---

> ### Author Rebuttal · Authors · 2025-07-30
>
> We thank the reviewer for their comments and positive feedback. Please find our responses to the main concerns and questions below:
>
> >**W1&Q3**: Hyper-Heuristics (HH) and automated algorithm design
>
> Our approach shares the same spirit as the primary objective of HH in automatically deriving heuristics. Also, Neural Combinatorial Optimization (NCO) can be regarded as a variant of HH, wherein neural architectures and solution pipelines define a heuristic space, and training algorithms search within it [1]. We will update the paper to include a discussion on HH.
>
> [**Revised**] Line 27: Beyond expert-designed heuristics, Neural Combinatorial Optimization (NCO) methods, as a variant of Hyper-Heuristics (HH) [2], have recently emerged to automate the heuristic design process
>
> Additionally, we conducted further experiments on the Flexible Job Shop Scheduling Problem (FJSSP) as requested by another reviewer. We included Genetic Programming (GP) [3], a representative HH, as a baseline. From the Table1 (below), we can see that MP-ASIL significantly outperforms all baselines. Thank you once again for providing valuable insight!
>
> Table 1. Experiment results on FJSSP.
>
> | Method                 | Type        | Brandimarte |
> |------------------------|-------------|-------------|
> | MWKR                   | Heuristics  | 28.91%      |
> | GP (Single-tree)       | HH          | 12.71%      |
> | GP (multi-tree)        | HH          | 12.13%      |
> | HGNN (S=100)           | NCH         | 18.56%      |
> | RS (S=100)             | NCH         | 15.40%      |
> | MCGN-PPO (S=100)       | NCH         | 18.67%      |
> | DAN (S=100)            | NCH         | 9.53%       |
> | MP-ASIL (k=100)        | NCH         | **7.32%**      |
>
> >**W2**: Informal terms
>
> Thank you for the helpful comment. We will remove informal terms from the manuscript.
>
> [**Removed**] Line 198: This weight measures how unexpected or surprising the best solution is relative to others.
>
> >**W3**: Diversity metric
>
> As you pointed out, we could employ more robust population-level diversity measures such as distance-based metrics or Shannon entropy. However, our main purpose was to measure how many non-duplicate solutions our method generates. This is because sampling from single policy approaches often produces many duplicate samples due to the relatively deterministic probability distribution, as has been pointed out in the literature [4,5,6]. In this context, we aimed to verify how much our multi-policy approach reduces the generation of duplicate solutions compared to single policy approaches. Therefore, we used the percentage of unique solutions as our diversity measure. We believe that the percentage of unique solutions is a sufficient and intuitive metric to achieve our evaluation purpose. Additionally, recent works in NCO literature [4,7] have employed the same percentage of unique solutions as their diversity measure.
>
> > **W4&Q2**: Omitting training time and runtime comparison
>
> - **Training time**: Training time is indeed a critical metric, which we have reported in Table 6 of Appendix D. However, we would like to clarify the distinction between training time and inference time in our experiments. In our experimental tables, the Time metric specifically reports the inference time required to solve the test datasets, which is the standard metric commonly adopted across NCO literature. Learning-based methods, including MP-ASIL, once trained offline, enable real-time decision making for new problems without retraining. This represents a significant computational advantage of learning-based methods over classical or heuristic solvers, which must spend considerable computational effort to solve each new instance from scratch. This is precisely why recent NCO methods have garnered significant attention in the optimization community.
>
> - **Runtime**: We acknowledge that direct runtime comparisons can indeed vary significantly due to implementation and hardware, as you point out. We appreciate your comment and will update the paper to provide clearer context regarding the limitations of runtime comparisons.
>
> [**Added**] Line 223: Importantly, Time may not be directly comparable due to different hardware and other factors. So, for clarity, we mark * on the results reported from the original papers.
>
> >**W5&Q1**: Overfitting or premature convergence
>
> - Naive self-imitation learning only imitates the best solution using single policy, which can lead to the issues you mentioned. In contrast, our method mitigates these problems through a policy specialization process in which different policies learn different high-quality solutions, thereby promoting solution diversity across the population. This design enables multiple policies to learn diverse and complementary problem-solving strategies based on latent variables. We also demonstrate empirically that our model consistently improves validation performance during training in Figure 10 of Appendix E.3.
>
> - As you mentioned in Q1, several components can be considered in our framework. Actually, in our framework, each policy employs stochastic selection rather than greedy selection. Also, we could apply entropy bonus. However, computing the entropy over the entire trajectory remains computationally intractable [5,7]. Finally, memory buffers of high-quality solutions represent a promising avenue for reducing overfitting risks and enabling more stable training. We appreciate these constructive suggestions for future research directions. We would be happy to collaborate if there is the opportunity.
>
> >**W6**: The comparison of the different policies performance
>
> Could you please clarify whether you are referring to the results presented in Table 8, which evaluates the performance across different latent variable distributions?
> If you meant a different result of our policy comparison analysis, we would appreciate your guidance to ensure we address your concern accurately.
>
> >**W7**: Readability
>
> We sincerely apologize for readability issues. For improved readability, we will update the paper to move the Time from Tables 2 and 3 to a separate table in the appendix. This will allow us to increase the font size for each metric.
>
> >**Q4**: Generalizes across different CO classes
>
> - Our main contribution and focus are on scheduling problems. However, to demonstrate the broader applicability of our approach to other CO problems, we conducted evaluations on TSP and 10 CVRP variants, which are widely used benchmarks in the NCO literature. While we acknowledge that 200 node problems may be considered relatively small, we scaled to the same instance sizes as previous works for fair comparison [8,9]. Our experimental results demonstrate that our method consistently outperforms previous state-of-the-art methods on routing problems by a large margin. We recognize that evaluating the applicability to much larger-scale routing problems would require additional research and experimentation. However, since our primary focus is on scheduling, we did not pursue this direction in the current study.
>
> - Our method can be easily applied to other CO problems with minimal modifications
>     - **Architecture**: As described in Appendix B, multi-policy can be readily implemented by concatenating latent variables to the input of the decision-making layer, regardless of problem-specific architectures. This design choice ensures that our approach is model-agnostic and can be integrated with existing neural architectures for various CO problems.
>     - **Policy Representation**: As mentioned in the Architecture section, since we only need to condition on latent variables, there is no need to modify the policy representation.
>     - **Diversity Mechanism**: Since all CO problems allow for ranking candidate solutions generated for an instance based on their objective values, MP-ASIL can be applied without any modifications. Therefore, MP-ASIL specialization mechanism ensures that different policies can learn diverse problem-solving strategies, eliminating the need for additional diversity mechanisms.
>
> > **P**: Reference section
>
> We appreciate your feedback! By consolidating some references without any ethical issues, we will add more content (e.g., HH) in the main text.
>
>
> [1] Ye, Haoran, et al. "Reevo: Large language models as hyper-heuristics with reflective evolution." Advances in neural information processing systems 37 (2024): 43571-43608.
>
> [2] Burke, Edmund K., et al. "A survey of hyper-heuristics." Computer Science Technical Report No. NOTTCS-TR-SUB-0906241418-2747, School of Computer Science and Information Technology, University of Nottingham (2009).
>
> [3] Braune, Roland, et al. "A genetic programming learning approach to generate dispatching rules for flexible shop scheduling problems." International Journal of Production Economics 243 (2022): 108342.
>
> [4] Shi, Kensen, David Bieber, and Charles Sutton. "Incremental sampling without replacement for sequence models." International Conference on Machine Learning. PMLR, 2020.
>
> [5] Xin, Liang, et al. "Multi-decoder attention model with embedding glimpse for solving vehicle routing problems." Proceedings of the AAAI Conference on Artificial Intelligence. Vol. 35. No. 13. 2021.
>
> [6] Pirnay, Jonathan, and Dominik G. Grimm. "Self-improvement for neural combinatorial optimization: Sample without replacement, but improvement." arXiv preprint arXiv:2403.15180 (2024).
>
> [7] Hottung, André, Mridul Mahajan, and Kevin Tierney. "PolyNet: Learning diverse solution strategies for neural combinatorial optimization." arXiv preprint arXiv:2402.14048 (2024).
>
> [8] Grinsztajn, Nathan, et al. "Winner takes it all: Training performant rl populations for combinatorial optimization." Advances in Neural Information Processing Systems 36 (2023): 48485-48509.
>
> [9] Kwon, Yeong-Dae, et al. "Pomo: Policy optimization with multiple optima for reinforcement learning." Advances in Neural Information Processing Systems 33 (2020): 21188-21198.

---

> > ### Comment · Reviewer_SjhF · 2025-08-05
> >
> > Most of my comments have been properly addressed either in your responses or through the modifications in the manuscript. Please find below my follow-up remarks:
> >
> > - W1 & Q3: The added sentence is a good improvement, and the inclusion of GP is certainly a plus. However, is it made explicit in the text that GP is used here as a representative of HH? This point may not be obvious to readers unfamiliar with HH concepts.
> >
> > - W2: Addressed satisfactorily.
> >
> > - W3: Thank you for the clarification. To avoid similar questions from other readers, it would be beneficial to motivate the choice of this metric directly in the article, for example by citing the references [4,7] you mention in your response.
> >
> > - W4 & Q2: Regarding training time, it would be worthwhile to explicitly mention in the paper this advantage over classical or heuristic methods. I am satisfied with the small but meaningful update you made regarding runtime.
> >
> > - W5 & Q1: Could this aspect also be reflected in an updated future work section?
> >
> > - W6: Correct, this was indeed referring to Table 8 and its corresponding analysis, apologies for the earlier lack of precision.
> >
> > - W7: Addressed satisfactorily.
> >
> > - Q4: Thank you for the clarifications. Do you plan to make this point more explicit in the revised manuscript? As it stands, it is somewhat scattered across a couple of sections.
> >
> > - References: The additional references are a valuable inclusion and will help strengthen several of your claims.

---

> ### Author Response · Authors · 2025-08-05
>
> We sincerely thank you for your time and effort in reviewing our work. We also appreciate your thoughtful feedback and dedication to improving the quality of our paper. Please find our response to your new remarks below.
>
> >**W1&Q3**:
>
> We acknowledge that a sufficient explanation of GP should be provided for readers who are not familiar with HH. We have added content about a flexible job shop in a new appendix section. We will provide an explanation here that GP is a representative of HH. Thank you for your valuable comment.
>
> [**Added**] We also include Genetic Programming (GP) [1], a representative Hyper-Heuristics (HH), as a baseline. GP is the widely used methodology among dispatching rule generation HH, evolving populations of individual tree structures to automatically discover effective problem-solving strategies [2]. This shares the same spirit as our method from the perspective of "heuristics to generate heuristics."
>
> >**W3**:
>
> We have added relevant references and the reasoning behind our choice (as mentioned in our original response to W3) to the main text to provide further context. Thank you for your helpful feedback.
>
> >**W4&Q2**:
>
> We will add the computation time advantages of learning-based methods over classical or heuristic methods to the related work section.
>
> [**Added**] Line 76: These methods, once trained offline, enable real-time decision making for new problems without retraining. This represents a significant computational advantage over classical or heuristic solvers, which must spend considerable computational effort to solve each new instance from scratch.
>
> >**W5&Q1**:
>
> Adding this content to the main paper will help clarify the strengths of our methodology and future research directions. We will add the advantages of our methodology over naive SIL methods regarding overfitting or premature convergence issues to the related work section.
>
> [**Added**] Line 96: These features can lead to overfitting or premature convergence issues. In contrast, MP-ASIL addresses these challenges via policy specialization process and adaptive self-imitation mechanism, demonstrating its effectiveness (performance and exploration capability) across various JSPs.
>
> Also, we will add content about the potential of memory buffers to the future work section (We have revised the title of Appendix G from "Limitations" to "Limitations and Future Works"). Thank you once again for suggesting this valuable research direction!
>
> [**Added**] Line 920: Finally, to enhance the training stability and sample utilization of MP-ASIL, memory buffers of high-quality solutions, similar to experience replay buffer [3], can be introduced, which represents a promising avenue for reducing overfitting risks and enabling more stable training.
>
> >**W6**:
>
> Thank you for confirming this! As you mentioned, it may be difficult to consider the differences between different distributions as significant. Nevertheless, as shown in Table 1, all distributions demonstrated superior performance compared to the state-of-the-art single policy approaches. This finding provides evidence for the robustness and generalizability of our approach, because our method's effectiveness is not dependent on fine-tuning the latent space design, but rather stems from the fundamental multi-policy optimization mechanism. We will add this analysis to Appendix E.2.
>
> **Table1**. The effect of latent distributions. Notations follow the main paper.
> | | PFSP | JSSP |
> |---|---|---|
> | SLIM [4] | 2.98\% | 7.80\% |
> | REINFORCE [5] | 2.60\% | 9.36\% |
> | $\mathcal{Z}_1$| 1.79\% | 7.53\% |
> | $\mathcal{Z}_2$| 1.74\% | 7.47\% |
> | $\mathcal{Z}_3$| 1.68\% | 7.41\% |
>
>
> >**Q4**:
>
> To enhance clarity, we will add the content from our response to Q4 to Appendix B to explicitly explain the versatility of MP-ASIL across various COPs. We will also modify the title of Appendix B to "Implementation of MP-ASIL to other COPs" to better emphasize that MP-ASIL can be universally applied to other COPs without algorithm modifications. Thank you for your informative comment.
>
> ---
> [1] Braune, Roland, et al. "A genetic programming learning approach to generate dispatching rules for flexible shop scheduling problems." International Journal of Production Economics 243 (2022): 108342.
>
> [2] Burke, Edmund K., et al. "Exploring hyper-heuristic methodologies with genetic programming." Computational intelligence: Collaboration, fusion and emergence. Berlin, Heidelberg: Springer Berlin Heidelberg, 2009. 177-201.
>
> [3] Mnih, Volodymyr, et al. "Human-level control through deep reinforcement learning." nature 518.7540 (2015): 529-533.
>
> [4] Corsini, Andrea, et al. "Self-labeling the job shop scheduling problem." Advances in Neural Information Processing Systems 37 (2024): 105528-105551.
>
> [5] Kwon, Yeong-Dae, et al. "Pomo: Policy optimization with multiple optima for reinforcement learning." Advances in Neural Information Processing Systems 33 (2020): 21188-21198.

---

> > ### Comment · Reviewer_SjhF · 2025-08-07
> >
> > Dear Authors,
> > Thank you for the constructive exchange and the improvements made to the manuscript.  I have no further remarks or questions. While I will maintain the overall score based on the level of contribution, I have increased the clarity score to reflect the enhancements made during the rebuttal phase.

---

> ### Author Response · Authors · 2025-08-08
>
> Thank you again for your time and effort in reviewing our work. We are happy to know your concerns are all well addressed!

---

### Note · Authors · 2025-08-12

Dear AC and Reviewers,

We sincerely thank you for your time and effort in reviewing our work.
We are encouraged that reviewers found our paper to be well-written (**SjhF**, **aRja**, **9EdW**), addressing practically important problems (**SjhF**), and that our method and idea were recognized as principled (**SjhF**), novel (**9EdW**),  interesting (**9EdW**), and practically relevant (**aRja**). We also appreciate the acknowledgement of our extensive experimental evaluation (**SjhF**, **aRja**, **NWcr**, **9EdW**), which includes vehicle routing problems and demonstrates significant performance improvements (**SjhF**, **NWcr**, **9EdW**).

We are pleased to have addressed all reviewers' concerns and questions. Following the rebuttal, all reviewers have recommended a borderline accept.
We will carefully incorporate the feedback received during the rebuttal period and additional experimental results into the revised version of the paper. Specifically:
- We will revise the main text in line with all comments received, enhancing clarity and precision.
- We will include additional experimental results, such as for (stochastic-) flexible job shop scheduling problems, further ablation studies, and the application of MP-ASIL to BQ [1] for solving large-scale TSP.
- To ensure a comprehensive comparison, we will add results for additional baseline methods, including genetic programming (a well-known hyper-heuristic) and SI GD (another self-imitation learning approach) [2] to the experiment section.

Once again, we greatly appreciate your thoughtful feedback and constructive suggestions, which have been invaluable in improving the quality of our work.

Best Regards,

Paper 9960 Authors

---
[1] Drakulic, Darko, et al. "Bq-nco: Bisimulation quotienting for efficient neural combinatorial optimization." Advances in Neural Information Processing Systems 36 (2023): 77416-77429.

[2] Pirnay, Jonathan, and Dominik G. Grimm. "Self-improvement for neural combinatorial optimization: Sample without replacement, but improvement." arXiv preprint arXiv:2403.15180 (2024).

---

### Decision · Program_Chairs · 2025-09-17

**Decision:**

Accept (poster)

**Comment:**

This paper introduces a multi-policy optimization method that that uses an adaptive self-imitation learning mechanism for job scheduling problems.

The reviewers generally found that the paper paper is well-written, addresses a practical problem, and presents an interesting approach to multi-policy optimization. They also appreciated the extensive experimental results. Although there were a few concerns regarding comparisons to other methods, the practicality of the problem formulation, and requests for further ablation studies and more diverse benchmarks. The authors addressed them with additional experiments during the rebuttal period, which includes adding a Genetic Programming baseline, evaluating their method on the complex stochastic and deterministic Flexible Job Shop Scheduling Problem, and providing new analysis on inference-time search strategies. As a result, all of the reviewers converged on a positive recommendation for the paper.  Thus, I recommend acceptance.